# PD-L1 and ICOSL discriminate human Secretory and Helper dendritic cells in cancer, allergy and autoimmunity

Caroline Hoffmann [1,2,3✉], Floriane Noel [4], Maximilien Grandclaudon[1,3,20], Lucile Massenet-Regad [4,5,20], Paula Michea[6,7,20], Philemon Sirven[1,3], Lilith Faucheux [4,8], Aurore Surun[9], Olivier Lantz [1,3,10], Mylene Bohec[3,11], Jian Ye[12], Weihua Guo[12], Juliette Rochefort[13], Jerzy Klijanienko[3,14], Sylvain Baulande [3,11], Charlotte Lecerf[3,15], Maud Kamal[3,15], Christophe Le Tourneau [5,15,16], Maude Guillot-Delost[1,3,10] & Vassili Soumelis [1,3,4,17,18,19✉]

Dendritic cells (DC) are traditionally classified according to their ontogeny and their ability to induce T cell response to antigens, however, the phenotypic and functional state of these cells in cancer does not necessarily align to the conventional categories. Here we show, by using 16 different stimuli in vitro that activated DCs in human blood are phenotypically and functionally dichotomous, and pure cultures of type 2 conventional dendritic cells acquire these states (termed Secretory and Helper) upon appropriate stimuli. PD-L1highICOSLlow Secretory DCs produce large amounts of inflammatory cytokines and chemokines but induce very low levels of T helper (Th) cytokines following co-culturing with T cells. Conversely, PD-L1lowICOSLhigh Helper DCs produce low levels of secreted factors but induce high levels and a broad range of Th cytokines. Secretory DCs bear a single-cell transcriptomic signature indicative of mature migratory LAMP3+ DCs associated with cancer and inflammation. Secretory DCs are linked to good prognosis in head and neck squamous cell carcinoma, and to response to checkpoint blockade in Melanoma. Hence, the functional dichotomy of DCs we describe has both fundamental and translational implications in inflammation and immunotherapy.

[1] Institut Curie, INSERM U932, Immunity and Cancer, Paris, France. [2] Institut Curie, Department of Surgical Oncology, Paris & Saint-Cloud, France. [3] Université Paris Sciences Lettres (PSL), Paris, France. [4] Université de Paris, Institut de Recherche Saint-Louis, INSERM U976, Hôpital Saint-Louis, 75010 Paris, France. [5] Université Paris-Saclay, Orsay, France. [6] Institut Paoli Calmette, INSERM U1068—CNRS UMR7258—AMU UM105, Marseille, France. [7] Université Aix-Marseille, Marseille, France. [8] Statistic and Epidemiologic Research Center Sorbonne Paris Cité, INSERM UMR-1153, ECSTRRA team, Paris, France. [9] Institut Curie, SIREDO Cancer Center, Paris, France. [10] CIC IGR-Curie 1428, Center of Clinical Investigation, Paris, France. [11] Institut Curie, NGS platform, Paris, France. [12] City of Hope Comprehensive Cancer Center, Department of Immuno-Oncology, Duarte, CA, USA. [13] Cimi Paris, INSERM U1135, and Hospital Pitié Salpêtrière, Odontology department, Université de Paris, Paris, France. [14] Institut Curie, Department of pathology, Paris, France. [15] Institut Curie, Department of Drug Development, and Innovation (D3i), Paris & Saint-Cloud, France. [16] Institut Curie, INSERM U900, Saint-Cloud, France. [17] Institut Curie, Clinical immunology department, Paris, France. [18] Assistance Publique-Hôpitaux de Paris (AP-HP), Hôpital Saint-Louis, Laboratoire d'Immunologie, F-75010 Paris, France. [19] Present address: Université de Paris, Institut de Recherche Saint-Louis, INSERM U976, Hôpital Saint-Louis, 75010 Paris, France. [20] These authors contributed equally: Maximilien Grandclaudon, Lucile Massenet-Regad, Paula Michea. ✉email: caroline.hoffmann@curie.fr; vassili.soumelis@aphp.fr

Dendritic cells (DC) play a key role in inflammation by initiating and polarizing immune responses and are considered as the most important cell type bridging the innate and adaptive immune systems. Immature DC patrol tissues, sense, and capture antigens, which induces their maturation. Matured DC secrete inflammatory mediators, acquire a strong T cell stimulatory capacity and antigen presentation properties[1]. Functionally, DC have been classified as immunogenic when they promoted the immune response including inducing CD4 T helper (Th) and CD8 cytotoxic T cell responses, or tolerogenic when they induced tolerance and regulatory T cells (TReg)[2–4].

In cancer, DC are critical for the antitumor immune response and essential for immunotherapy efficacy, whether they are directly being used or targeted[5], and/or are indirectly needed like for T-cell-negative checkpoint blockade[6,7]. Studies have suggested that factors derived from the tumor microenvironment induce tolerogenic DC with various degrees of maturation[8–12]. Hence, the phenotypic and functional state of DC in cancer remains controversial. This complicates the choice and development of DC-targeting strategies. Many different compounds are currently being tested to induce DC recruitment and maturation that have been recently reviewed[13]. When used as single agents, the clinical efficacy has been limited, and novel approaches integrate DC stimuli combined with other treatments, such as checkpoint blockade, chemotherapy, or radiotherapy[5].

In autoimmunity, DC are considered immunogenic and may contribute to disease physiopathology by breaking tolerance to self-antigens[14]. Similarly in allergy, DC become immunogenic after stimulation by allergens[15] and by epithelial derived factors called alarmins[16,17]. The modulation of DC maturation is a promising therapeutic approach in allergy and autoimmunity, with the objective of reducing inflammation and promoting immune tolerance[18,19].

In this study, we use a systematic translational research approach to connect three major components of DC contribution to disease: (1) diversity of activating stimuli, representative of various physiopathological conditions, and immune-modulating drugs, (2) DC phenotype and function, and (3) disease implication through ex vivo analyses. We uncover a novel functional dichotomy of human DC maturation in vitro that has physiopathological relevance in several cancers, atopic dermatitis, and lupus. For immunotherapy, this dichotomy may guide the selection of DC activating compounds.

## Results

**T-cell-inflamed head and neck squamous cell carcinoma are enriched in DC expressing PD-L1$^{high}$ and ICOSL$^{low/neg}$.** To decipher the phenotypic heterogeneity of human cancer-infiltrating DC and its relation to T cell infiltration and other immune cell types, we analyzed by flow cytometry 22 fresh head and neck squamous cell carcinoma (HNSCC) samples. We used two different antibody panels analyzing T cell subsets (Supplementary Fig. 1) and myeloid cells subsets (Fig. 1A). In the myeloid panel, CD45$^+$Lineage$^-$ (CD3, CD19, CD56) cells were analyzed by their expression of CD11c and HLA-DR (Fig. 1B, Supplementary Fig. 2A). The double-positive population was separated into four populations based on CD14 and CD1c, and included the CD14$^+$ monocytes and macrophages (MMAC), the CD14$^+$DC, the cDC2 (CD1c$^+$CD14$^-$) and the double negative population named "DN DC/MMAC" containing cDC1 (Supplementary Fig. 2B) and other myeloid CD14$^-$CD1c$^-$ cells[20]. Plasmacytoid DC (pDC) were gated as CD11c$^-$, HLA-DR$^+$, CD123$^+$. We extracted a total of 434 parameters, and we used a sublist of 81 non-redundant parameters in which each population was expressed only in percentage of its parental population

(Supplementary Table 1) to avoid bias in the subsequent analyses. We found a large variation of CD3 infiltration across tumors ranging from 1 to 61% of live cells (Fig. 1C). Using correlation analysis, we observed that tumors high in CD3 were enriched in cDC2, DN DC/MMAC, pDC, PD-L1$^+$ cDC2, PD-L1$^+$CD14$^+$DC, PD-L1$^+$ DN DC/MMAC and PD-L1$^+$ MMAC, and in CD8 T cells (Fig. 1D; full correlation matrix in Supplementary Data 1). Conversely, tumors low in CD3 were enriched in Lin$^-$DR$^-$ cells and neutrophils (Supplementary Fig. 2C), MMAC, and ICOSL$^{int}$-expressing DC and MMAC (all subsets) (Fig. 1D). DC and MMAC subsets in CD3 low tumors expressed PD-L1$^{int}$ and ICOSL$^{int}$ and were closer to the expression observed on their blood counterparts than the same subsets in CD3 high tumors, which were PD-L1$^{high}$ and ICOSL$^{low/neg}$ (Fig. 1E, Supplementary Fig. 2D). We then performed an elastic net model including all the 434 phenotypic parameters and 14 clinical parameters (Supplementary Table 2) to determine if any was associated with CD3 infiltration levels; that model required a classification of the tumors by tertile as "CD3 High" ($n = 8$), "CD3 Int" ($n = 6$), and "CD3 Low" ($n = 8$) (Supplementary Fig. 3A). We confirmed that ICOSL$^{int}$ expression on CD11c+ HLA-DR- cells was highly characteristic of the CD3$^{low}$ group (Supplementary Fig. 3B). Only parameters causally linked to T cell infiltration (percentages of T cell subsets in live cells) were found associated to the CD3$^{high}$ group. In summary, we showed that CD3-inflamed tumors were more infiltrated by DC subsets that expressed higher levels of PD-L1 than in non-inflamed tumors, and that PD-L1 expression was opposed to ICOSL expression on DC and MMAC.

**PD-L1 and ICOSL expression on matured blood cDC are mutually exclusive.** To identify candidate stimuli that could be responsible for the opposed PD-L1/ICOSL expression pattern on DC and to further understand the subsequent functional implications, we took advantage of a large, protein level, human DC–T cell dataset generated in our laboratory[21]. We used the existing data on primary blood cDC composed of a majority of cDC2 and a minority of cDC1 (we excluded monocyte-derived DC and pDC), and generated novel experiments and analyses. Briefly, blood cDC were activated for 24 h by 16 different types of "perturbators" (innate stimuli) (Supplementary Table 3) and analyzed for their protein expression of 29 surface markers listed in Supplementary Table 4 ($n = 154$ data points), and their secretion of 32 chemokines and cytokines listed in Supplementary Table 5 ($n = 130$ data points). The remaining cells were co-cultured with allogeneic naïve CD4 T cells for 6 days and we measured the expansion fold. After 24 h of restimulation by anti CD3/CD28, we measured the secretion of 17 Th cytokines listed in Supplementary Table 6 (Fig. 2A).

First, we assessed the relation between PD-L1 and ICOSL expression in this unbiased dataset, representative of a broad array of innate stimulatory conditions. This revealed three main expression groups: (i) PD-L1$^{high}$ and ICOSL$^{low/neg}$, like ex vivo cDC2 from inflamed tumors; (ii) PD-L1$^{low}$ and ICOSL$^{high}$, and (iii) PD-L1$^{low}$ and ICOSL$^{low}$, like culture medium alone, representative of unstimulated blood cDC and ex vivo cDC2 in non-inflamed tumors (Fig. 2B). Co-expression of both PD-L1$^{high}$ and ICOSL$^{high}$ (upper right quadrant) was a rare profile and was never observed for the highest expression levels (Fig. 2B). Specific ICOSL expression was null when PD-L1 expression reached its highest levels. This confirmed the anticorrelation of PD-L1 and ICOSL expression observed in the tumor dataset, and further established that high expression levels of PD-L1 and ICOSL were mutually exclusive.

We performed an unsupervised analysis of the 29 surface markers to verify that PD-L1 and ICOSL were relevant markers in

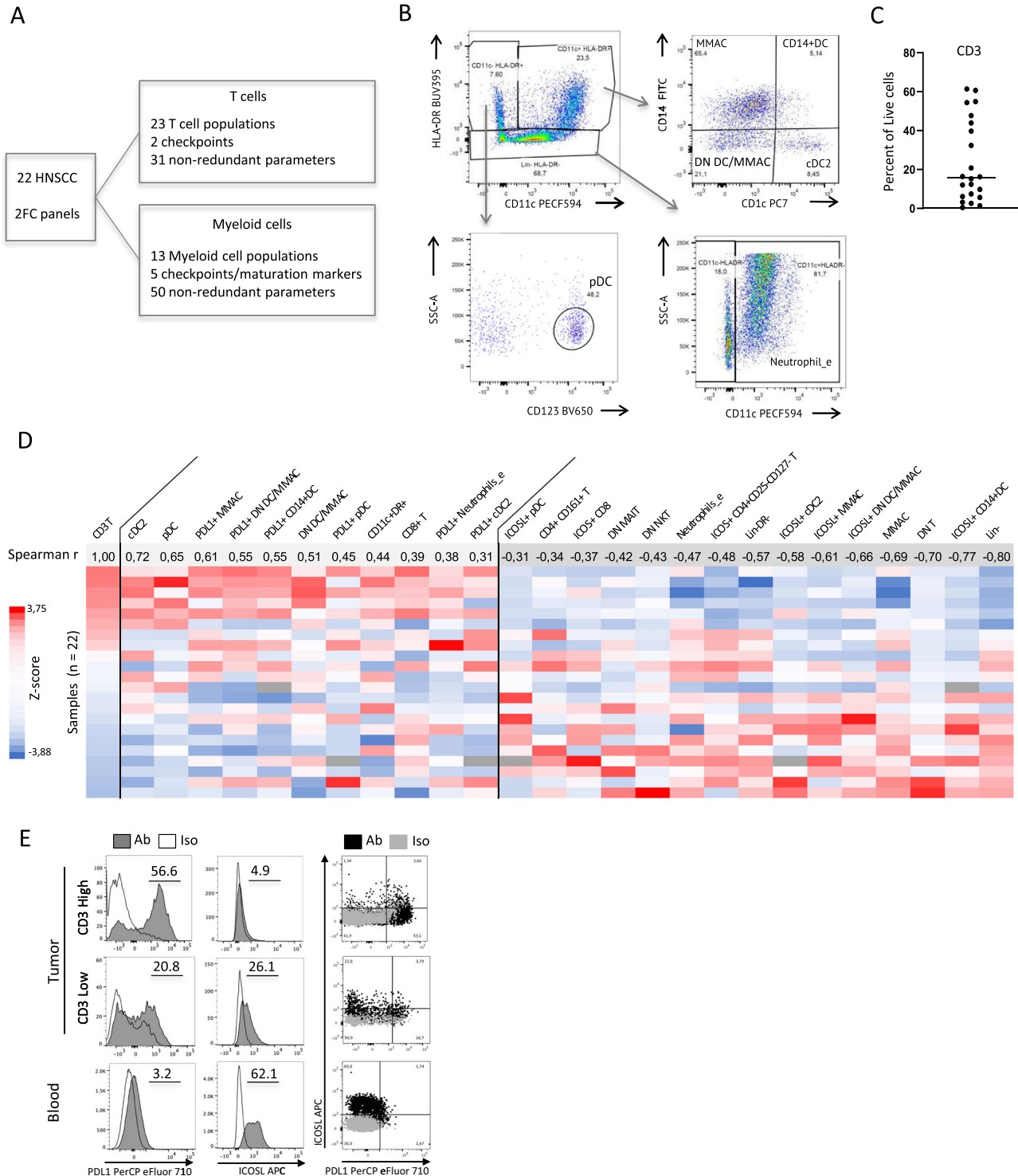

**Fig. 1 T-cell-inflamed head and neck squamous cell carcinoma are enriched in cDC expressing PD-L1$^{high}$ and ICOSL$^{low/neg}$.** Phenotypic characterization of 22 human head and neck squamous cell carcinoma (HNSCC) primary tumor-infiltrating cells. **A** Multicolor flow cytometry (FC) analysis scheme. **B** Myeloid cell panel gating strategy for the CD45$^+$CD3$^-$CD16$^-$CD19$^-$ (Lin) compartment (initial gates in Supplementary Fig. 2A). MMAC monocytes and macrophages, DN DC/MMAC double negative population. **C** Percentage of CD3-positive cells among live cells, bar represents median. **D** Heatmap representing the normalized values of the parameters (columns) with a Spearman correlation coefficient (r) over 0.3 (left) or under −0.3 (right) for the 22 tumors (rows) ordered from top to bottom by decreasing CD3/Live normalized value. Dark gray cells represent missing values. T T cells, pDC plasmacytoid DC, DR HLA-DR, Neutrophils_e neutrophils enriched, DN MAIT, NKT, T CD4$^-$CD8$^-$ MAIT, Natural Killer T, and T cells, respectively. **E** Representative staining of PD-L1 (left), ICOSL (center), and PD-L1 versus ICOSL (right) in CD11c$^+$DR$^+$ cells in a CD3 high tumor (top), CD3 low tumor (middle) from the cohort in **D**, and blood from a healthy donor (bottom); the percentage of positive cells was measured based on antibody (Ab) as compared to isotype (Iso).

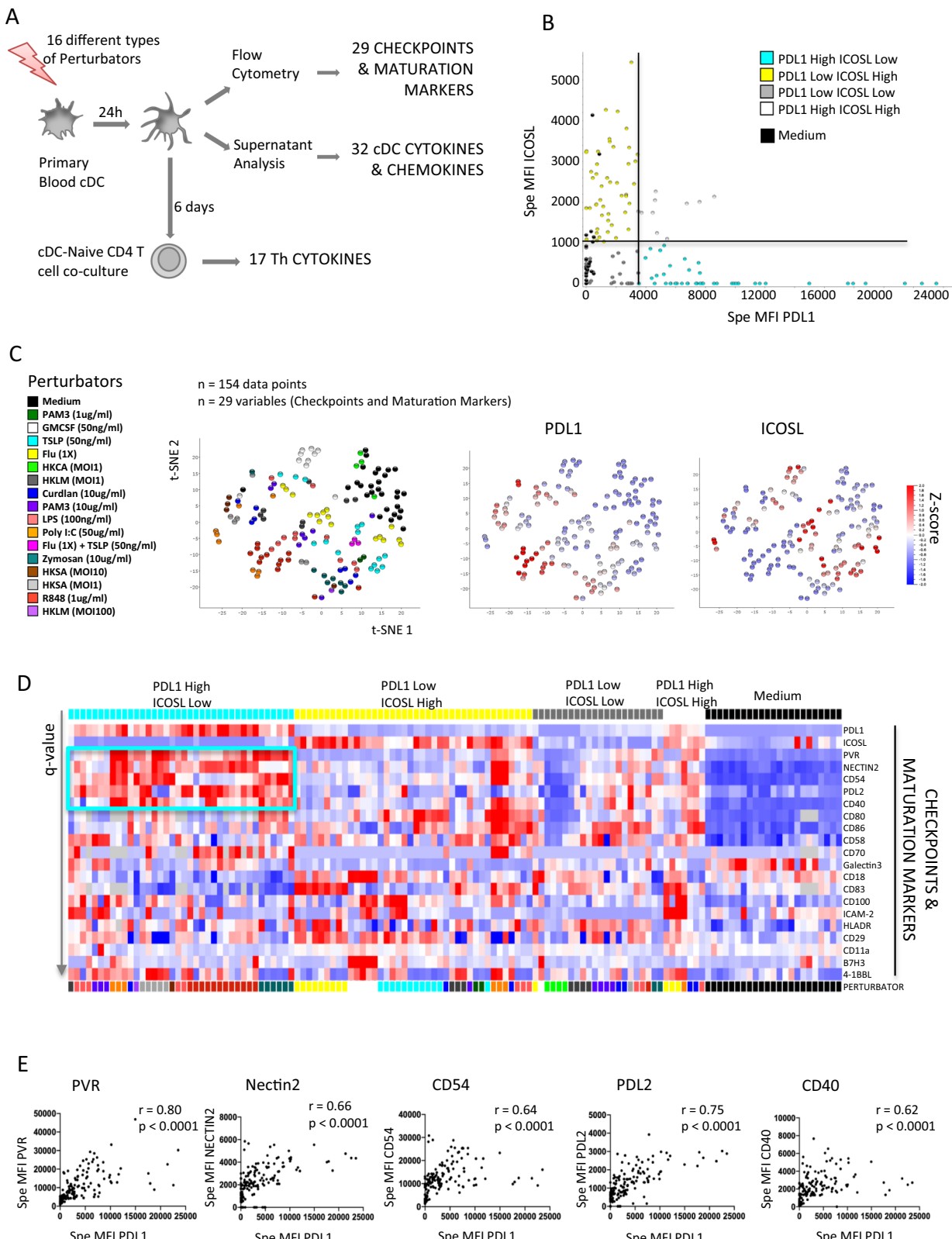

discriminating the various cDC phenotypes observed in vitro (Fig. 2C). We observed that PD-L1$^{high}$ cells clustered together, and were distinct from ICOSL$^{high}$ cell clusters, and from PD-L1$^{low}$ ICOSL$^{low}$ cluster, the latter including most Medium conditions (Fig. 2C). The cDC perturbators inducing a majority of PD-L1$^{high}$ ICOSL$^{low}$ cDC were R848, Zymosan, heat-killed Staphylococcus aureus (HKSA), and heat-killed Listeria

monocytogenes (HKLM), while the ones inducing a majority of PD-L1$^{low}$ ICOSL$^{high}$ cDC were Thymic stromal lymphopoietin (TSLP), GM-CSF, and influenza (Flu) (Fig. 2C, Supplementary Table 3).

We defined four groups of activated cDC based on their PD-L1 and ICOSL expression (Fig. 2B) and a 5th group for Medium-cDC and analyzed the 29 surface markers. PD-L1$^{high}$ ICOSL$^{low}$

**Fig. 2 PD-L1 and ICOSL expression on mature blood cDC are exclusive and PD-L1$^{high}$ DC overexpress PVR, Nectin2, CD54, CD40, and PD-L2.**
**A** Methods for the in vitro analysis of primary blood cDC. **B** Expression of PD-L1(x) vs ICOSL(y) on cDC at H24. 154 individual tests were annotated according to their expression of PD-L1 as high/low and ICOSL high/low with the thresholds of specific mean fluorescence intensity (MFI) at 3500 and 1000, respectively. **C** T-SNE of the 29 surface markers colored by stimuli (left), PD-L1 specific MFI (center), and ICOSL-specific MFI (right) using Qlucore software, $n = 154$. **D** Heatmap representing the expression of the 29 surface markers in the four groups defined by PD-L1 and ICOSL in "B", and in Medium condition. Multigroup comparison by Kruskal–Wallis test and Tukey post-hoc test. Only the variables significant at a $p$-value < 0.05 are represented and ordered by increasing $q$-value (max $q$-value = 0.046), among 130 individual experiments, ordered as in Fig. 3A. **E** Correlation of PD-L1 (x) with PVR, Nectin2, CD54, PD-L2, and CD40 (y). «r» values are Spearman correlation coefficients, $p$-value are for two-sided statistical analyses, $n = 154$. TSLP thymic stromal lymphopoietin, Flu influenza, HKCA heat-killed Candida albicans, HKLM heat-killed Listeria monocytogenes, HKSA heat-killed Staphylococcus aureus, LPS lipopolysaccharide.

cDC co-expressed PVR, PD-L2, Nectin2, CD54, and CD40 (Fig. 2D), and PD-L1 had Spearman correlation coefficients of 0.8, 0.75, 0.66, 0.64, and 0.62 with these molecules, respectively (Fig. 2E). PD-L1$^{low}$ ICOSL$^{high}$ cDC (Fig. 2D, yellow group) did not have systematically co-expressed molecules, and none of the measured molecules were associated with ICOSL with a Spearman correlation coefficient over 0.5.

**PD-L1 and ICOSL expression patterns characterize Secretory and Helper cDC.** Next, we jointly analyzed in these 5 cDC groups the secretion of 32 cDC-derived cytokines and chemokines (Fig. 3A top), and the induction of 17 Th cytokines secreted by naïve CD4 T cells after 6 days of co-culture (Fig. 3A bottom). PD-L1$^{high}$ ICOSL$^{low}$ cDC secreted the largest amount of most cytokines measured, including TNFa, IL-1a, IL-1b, IL-1RA, IL-6, IL-10, IL-12p40, IL-23, IL-27, CCL19, BCA1, MIP1a, as compared to the four other cDC groups (Fig. 3A top), and as compared to both PD-L1$^{low}$ ICOSL$^{high}$ cDC and to Medium-cDC (Fig. 3B top and Supplementary Fig. 4A, all $p$-values < 0.01)). Conversely, it was the PD-L1$^{low}$ ICOSL$^{high}$ cDC that induced the highest and broadest Th cytokine secretion after cDC-T co-culture, as compared to the four other cDC groups (Fig. 3A bottom). PD-L1$^{low}$ ICOSL$^{high}$ cDC induced more IL-3, IL-4, IL-5, IL-6, IL-9, IL-10, IL-13, IL-31, and GM-CSF as compared to both PD-L1$^{high}$ ICOSL$^{low}$ cDC and to Medium-cDC (Fig. 3B bottom, Supplementary Fig. 4B, all $p$-values < 0.05). IFNg and IL-2 were also preferentially induced by PD-L1$^{low}$ ICOSL$^{high}$ cDC as compared to PD-L1$^{high}$ ICOSL$^{low}$ cDC, but the difference was not significant with Medium-cDC. However, T cell expansion was neither significantly different between the PD-L1$^{high}$ ICOSL$^{low}$ and the PD-L1$^{low}$ ICOSL$^{high}$ cDC conditions (Supplementary Fig. 4C), nor was the percentages of live cDC (Supplementary Fig. 4D). Given their opposing functional profile, we named PD-L1$^{high}$ ICOSL$^{low}$ cDC "Secretory cDC" because of their ability to secrete cytokines and chemokines themselves, and PD-L1$^{low}$ ICOSL$^{high}$ cDC "Helper cDC" because of their ability to induce broad cytokine secretion by Th cells. This dichotomy was defined with blood cDC, and therefore largely refers to cDC2 that are the main subset of this compartment, as opposed to the minority of cDC1. However, although most stimuli fell into the category of Secretory or Helper inducers, regardless of their receptor and signaling pathway, Poly I:C had a unique profile undoubtedly via its effect on TLR3 on cDC1. Furthermore, we validated this functional dichotomy on sorted pure cDC2 (Supplementary Fig. 5A) similarly co-cultured with naïve CD4 T cells using R848-cDC2 as Secretory cDC and TSLP-cDC2 as Helper cDC (Supplementary Fig. 5B).

Next, we wanted to determine if the absence of stimulation of CD4 cytokine secretion by Secretory cDC was induced by the main immunosuppressive molecules they expressed. We performed a multiple blocking of the PD-1/PD-L1/2; IL-10/IL-10 receptor (IL-10R); IL-4/IL-4 receptor (IL-4R); TIGIT/PVR axes in a naïve CD4 T cell/Secretory R848-cDC2 co-culture. IL-4 was not

among the Secretory cDC molecules identified above, but its inhibitory role had been previously shown[10]. This multiple blocking increased the CD4 cytokine production of IL-2, and to a minor extend but not significantly of IL-3, IL-5, IL-13, and IFNg (Fig. 3C, Supplementary Fig. 6A), but the levels remained low as compared to the Helper TSLP-cDC2 condition. No change was observed for the other cytokines measured (IL-4, IL-6, IL-9, IL-10, GM-CSF, and TNFa). No individual blocking was able to induce the increased CD4 cytokine production observed with the multiple blocking. Therefore, the expression of PD-L1, PD-L2, PVR, and secretion of IL-10 and IL-4 by Secretory cDC2 played a significant but limited role in the inhibition of CD4 Th production of IL-2.

Given the importance of CD8 T cells in onco-immunology, we tested the effect of Secretory vs Helper cDC2 on naïve CD8 T cells. Again, Helper cDC2 and not Secretory cDC2 increased cytokine secretion by CD8 T cells, namely IL-3, IL-5, IL-13, GM-CSF, IFNg and TNFa (Supplementary Fig. 6B). The results were not significant for IL-2, IL-6, and IL-10 and no secretion of IL-4 and IL-9 was measured in any condition.

In summary, we have shown that cDC2 have two main maturation pathways with opposed functional profiles named Secretory and Helper, and that the main inhibitory molecules expressed by Secretory DC are only weakly inhibiting CD4 Th cytokine production.

**RNAseq of tumor and blood cDC2 confirms that T-cell-inflamed HNSCC are infiltrated by Secretory cDC.** In our phenotypic study of HNSCC-infiltrating DC, we identified PD-L1$^{high}$ICOSL$^{low}$ cDC2 in T-cell-inflamed tumors (Fig. 1), suggestive of Secretory cDC, whereas PD-L1$^{int}$ICOSL$^{int}$ cDC in non-inflamed tumors resembled unstimulated blood cDC (Fig. 1E, Fig. 2B). To validate that cDC2 in inflamed tumors harbored a Secretory cDC signature, we performed RNA sequencing (RNAseq) of cDC2 sorted from HNSCC and blood. Due to the minimum number of cells required for this experiment, inflamed tumors highly infiltrated by DC were necessarily selected (Supplementary Fig. 7). We identified 882 differentially expressed genes (DEG): 639 genes increased in tumor cDC2 (HNSCC-DC signature), and 243 genes increased in blood cDC2 (Fig. 4A, Supplementary Table 7 for donors' characteristics, and Supplementary Data 2 for DEG list). In order to define a transcriptomic signature characteristic of Secretory and Helper cDC, we made use of public transcriptomics data of cDC2 activated with two stimuli: pRNA, a TLR7/8 ligand expected to induce Secretory cDC, and GM-CSF a Helper cDC inducer (Fig. 4B)[22]. We compared each stimulus to unstimulated blood cDC2 and the two stimuli together to identify upregulated genes in both pRNA and GM-CSF conditions. Gene lists intersections were used to define the pRNA Secretory and GM-CSF Helper signatures including 1473 and 1277 genes, respectively (Fig. 4C, Supplementary Data 3 for gene lists). We compared those two signatures with the HNSCC-cDC2 signature: already, among the 639 genes

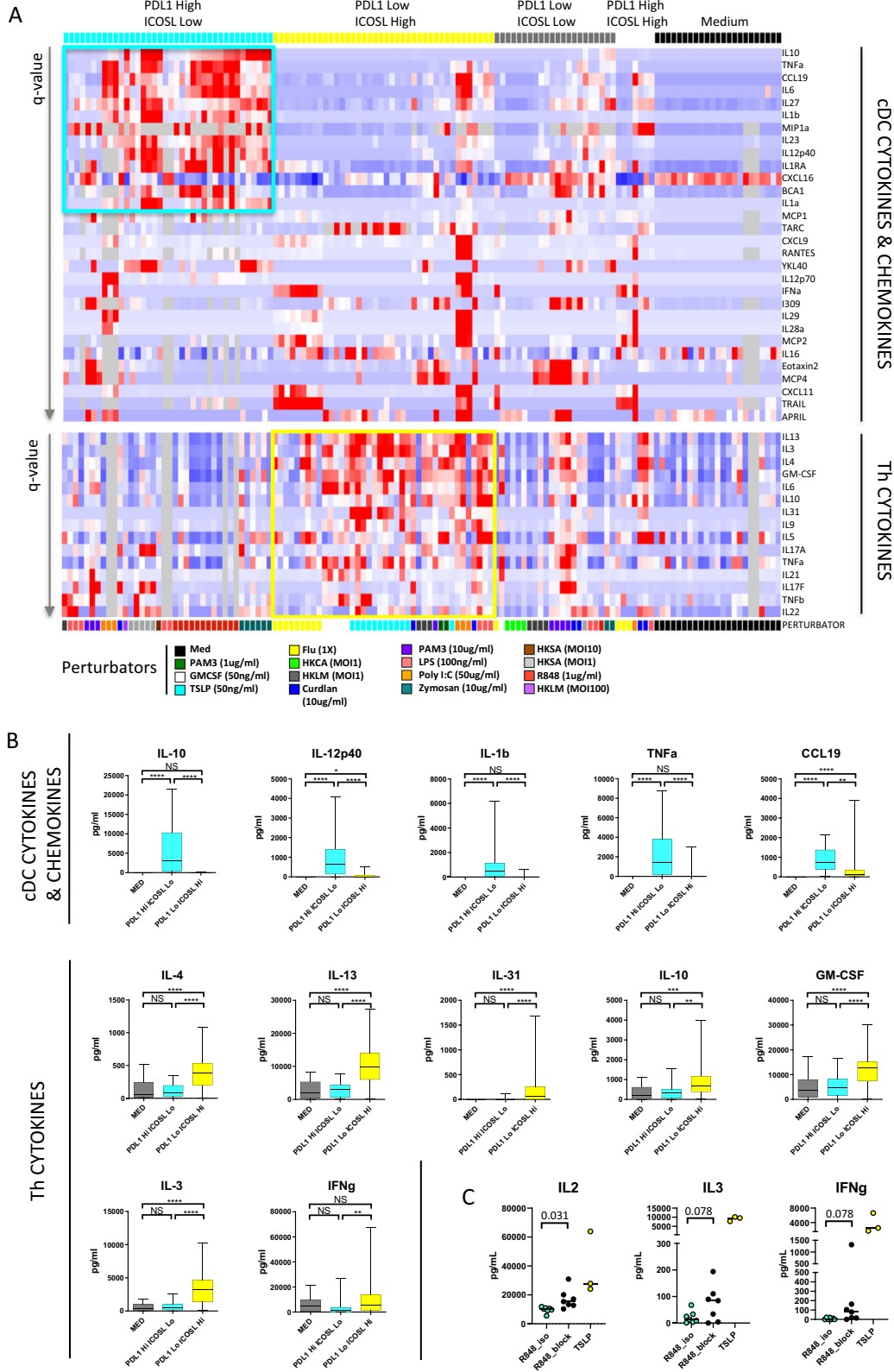

upregulated during tumor-induced maturation, 135 (21%) were shared with the Secretory cDC signature and only 64 (10%) with the Helper cDC signature, showing that tumor-induced maturation was biased towards Secretory cDC (Fig. 4D). More importantly, in order to compare with our proteomic data and to further define HNSCC-cDC2 phenotype, we defined by literature-

search large supervised lists of genes coding for checkpoints and maturation markers (148 genes, Supplementary Data 4), cytokines and chemokines (117 and 52 genes, respectively, Supplementary Data 5–6), and of the NFkB pathway, known for its importance in cDC activation[23] (100 genes, Supplementary Data 7), and identified in total 83 genes in common with the

**Fig. 3 PD-L1 and ICOSL expression pattern characterize Secretory and Helper cDC. A** Heatmaps representing the cytokines and chemokines secreted by the cDC measured in H24 supernatants (top), and the CD4 Th cell cytokines measured after co-culture (bottom) in the 4 groups defined by PD-L1 and ICOSL expression and Medium condition. Only the variables significant at a p-value < 0.05 after Kruskal–Wallis test on the 5 groups and Tukey post-hoc tests are represented and ordered by increasing q-value (max q-value = 0.035 (top) and 0.055 (bottom)), among 130 individual experiments, ordered as in Fig. 2D. Cells in gray are missing values. Abbreviations for the perturbators: see Fig. 2. **B** Quantification of cytokines and chemokines secreted by the DC (top row) and of the CD4 Th cell cytokines (2 bottom rows) in the Medium ($n = 23$), PD-L1^high ICOSL^low ($n = 38$), and PD-L1^low ICOSL^high ($n = 40$) conditions (two-sided Kruskal–Wallis test on the 3 groups and Dunn's multiple comparison test). Central line represents median, box represents quartiles, whiskers represent min to max. p values are represented by range: *<0.05, **<0.01, ***<0.001, ****<0.0001. NS not significant. MED medium, Hi high, Lo low. **C** Quantification of cytokines secreted by the CD4 Th cytokines after co-culture with pure cDC2 sorted as in Supplementary Fig. 5A and treated with R848 (Secretory) or Thymic stromal lymphopoietin (TSLP) (Helper, $n = 3$). For the R848-cDC2 the co-culture was performed in the presence of PD-1; IL-10; IL-10R; IL-4R; TIGIT multiple blockings "R848_block" or the corresponding isotypes "R848_Iso" ($n = 7$, two-sided Wilcoxon test). Bar represents median. Source data are provided as a Source Data file.

HNSCC-cDC2 overexpressed genes (Fig. 4E). This supervised approach highlighted that HNSCC-cDC2 overexpressed several Secretory cDC-specific markers identified previously at the protein level, such as *CD274*/PD-L1, *PDCD1LG2*/PD-L2, *PVR, CD40* (Fig. 2D, E), *IL1B, IL12B, IL23A, TNF,* and *CCL19* (Fig. 3B, Supplementary Fig. 6), and other negative checkpoints such as *IDO1, IDO2,* and *HAVCR2*/TIM3, the migration marker *CCR7,* the maturation marker *LAMP3,* and *NFKB1* (Fig. 4F). Altogether, transcriptomic analysis confirmed that HNSCC-cDC2 shared many similarities with in vitro defined Secretory cDC. This provides an ex vivo validation at the transcriptomic level for the existence of Secretory cDC and suggests physiopathological relevance to cancer.

**Tumor Secretory cDC2 are associated with good prognosis and response to checkpoint blockade.** We have shown that inflamed tumors were enriched in matured cDC2 with a Secretory phenotype and hypothesized that this inflamed immune archetype reveals the existence of a spontaneous antitumor response that may benefit patients. To analyze the prognostic and predictive impact of tumor Secretory cDC2, we defined their specific gene signature. First, the genes coding for the proteins overexpressed by Secretory cDC in vitro (Fig. 2, Fig. 3A top) defined the 17-gene Secretory signature (Supplementary Fig. 8A). Second, we defined a 21-gene tumor cDC2 subset-specific signature by comparison with blood cDC2 (Fig. 4A) and tumor MMAC, CD14+ DC and pDC (Supplementary Fig. 8B–E, Supplementary Data 8). We merged the two signatures to obtain a final 36-gene signature of Tumor Secretory cDC2 (Supplementary Table 8).

We took advantage of publicly available datasets to analyze the prognostic and predictive value of this signature. We observed that a high expression of the Tumor Secretory cDC2 signature was associated to good prognosis in the HNSCC cohort of The Cancer Genome Atlas (TCGA) ($p = 0.007$, $n = 500$ patients) (Fig. 5A). To validate the inflamed immune archetype associating T cells and mature Secretory cDC2, we used a five-gene T cell signature and observed that it also had a similar prognostic impact and was highly correlated to the tumor Secretory cDC2 signature (Fig. 5A). In order to extend these findings to another type of tumor—and one type not directly in contact with the microbiota—, we exploited the METABRIC dataset[24]. The tumor Secretory cDC2 and the T cell signatures were of good prognosis in triple negative breast cancer (TNBC) ($p = 0.001$, $n = 318$ patients) (Fig. 5B left, Supplementary Fig. 9A), another type of inflamed tumor[25,26], but not in luminal breast cancer (LumBC) ($n = 1407$ patients) (Fig. 5B right, Supplementary Fig. 9B), known to be mostly a non-inflamed tumor type[25,26]. The signatures were highly correlated in both tumor types (Supplementary Fig. 9A, B).

Then we analyzed the expression of these signatures in public transcriptomic datasets of patients treated with negative immune checkpoint blockers. In the absence of public data available for HNSCC, we selected two independent datasets including 103[27] and 91 patients[28] presenting advanced melanoma and having received checkpoint blockers in monotherapy or combination. In both, we observed an increased expression of the Tumor Secretory cDC2 signature among responders as compared to non-responders (Wilcoxon[27] $p = 0.006$, Fig. 5C left[28] $p = 0.003$, Fig. 5C right). Again, the same results were obtained with the T cell signature, and both signatures were highly correlated (Supplementary Fig. 9C, D).

Altogether, tumor Secretory cDC2 were associated to an inflamed favorable microenvironment for antitumor immune response, spontaneously or in association with immunotherapy, in at least three cancer types.

**Tumor Secretory cDC2 align with the mature migratory cDC in HNSCC.** To determine whether this Secretory phenotype affected all cDC2 or only a subset of them in HNSCC, we performed single-cell RNA sequencing (ScRNAseq) of two tumor samples and one juxtatumor sample (Supplementary Table 9). We performed an asymmetric enrichment towards DC by flow cytometry sorting to increase their proportion in the final sample, but all cell types of the tumor microenvironment (TME) were sequenced (Supplementary Fig. 10). Unsupervised clustering found 24 clusters (Fig. 6A) that were named according to the DEG (Supplementary Data 9–10) and the expression of the 10 proteins measured by antibody-derived tags (ADT) (Fig. 6B right, Supplementary Fig. 11A, B; Supplementary Data 11). We identified cDC (#7-20-23); MMAC (#0-2-12); pDC (#1) and pDC_TCL1A (#16), previously described in blood;[29] CD8 T cells (#15: Exhausted, with the highest expression of ADT CD103, and two *TCF7* clusters #5: Cytotoxic, and #10: Memory); Conventional CD4 T cells (#3; #4: memory); CD4 TReg (#6: typical TReg and #13: TReg_PD1 that expressed *PDCD1* and lower levels of TReg genes and of ADT CD25); Cycling T cells (#21); B cells (#9); Plasma cells (#8); NK cells (#17); Mast cells (#14); Neutrophils that expressed ADT CD15 (#11); and non-immune cells including cancer cells (#18), erythrocytes (#19), and fibroblasts (#22) (Fig. 6A).

Next, we focused on cDC and MMAC (Fig. 6B, C). The three MMAC clusters corresponded to: MMAC_PDL1 (#0) that expressed ADT PD-L1, *CD274, PTGS2* (COX2), *IL1B, ISG15, INHBA,* and many chemokines; MMAC_NLRP3 (#2) that also expressed *S100A8-9, IL1B* and *VCAN;* and MMAC_C1Q (#12) that expressed Complement genes, *APOE, CD163, MSR1, SPP1,* and several metalloproteinases (*MMP*). These subsets seemed consistent with those observed in esophageal carcinoma[30]. Among the three cDC clusters, we identified cDC1 (#23), cDC2 (#7), and a cluster of mature migratory cDC (mmDC) expressing *CCR7* and *LAMP3* (#20) (Fig. 6B). The Tumor Secretory cDC2 signature was selectively expressed by the mmDC #20,

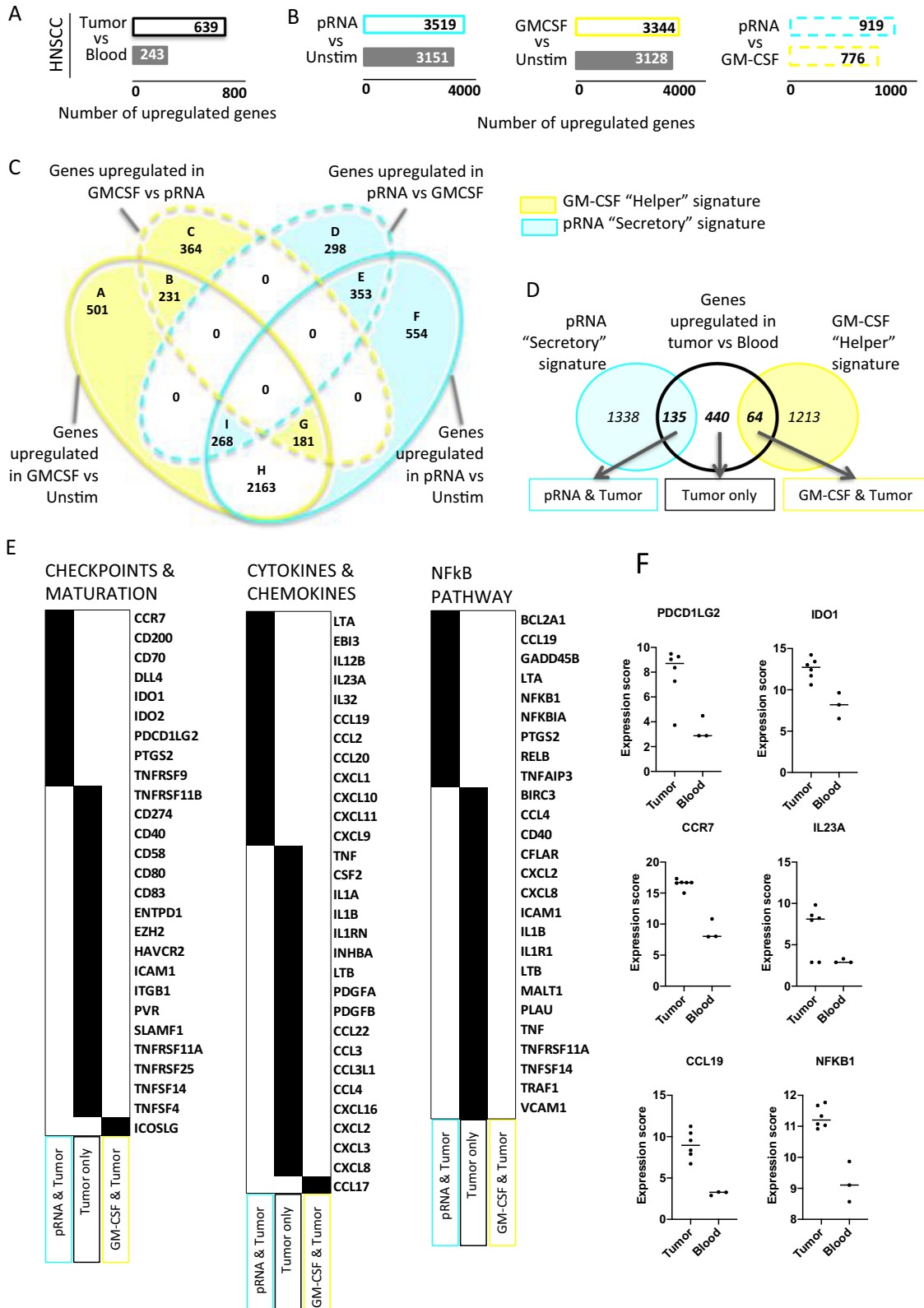

showing that a fraction of tumor cDC2 underwent a maturation that aligned with the in vitro defined Secretory maturation at the transcriptomic level (Fig. 6C, Supplementary Fig. 12A). ADT confirmed that mmDC expressed the highest levels of PD-L1 and did not express ICOSL (Fig. 6B, Supplementary Fig. 11A, B). ADT CD1c confirmed that these cells are at least in part cDC2 (Fig. 6B). cDC1 may also contribute to this population of mmDC

as previously shown[10,30]. None of the cDC clusters significantly expressed ADT CD103, as opposed to CD8 T cells and Mast cells (Supplementary Fig. 11A, B).

All immune cell clusters were identified in both the tumor and juxtatumor samples, but the relative proportions of mmDC, pDC, B cells, Plasma cells, Neutrophils, and Mast cells were increased in the tumor samples whereas MMAC_PDL1 and MMAC_NLRP3

**Fig. 4 RNAseq of tumor and blood cDC2 confirms that T-cell-inflamed HNSCC are infiltrated by Secretory DC. A** Analysis of differentially expressed genes (DEG) by DESeq2 between HNSCC tumor ($n = 6$) and blood cDC2 ($n = 3$). **B** Analysis of DEG from dataset GS87442 by DESeq2 between unstimulated cell and pRNA, a TLR7/8 ligand (left) or GM-CSF (center) and pRNA vs GM-CSF (right). **C** Venn diagram of upregulated genes identified in "**B**". The blue and the yellow-colored area contain the genes of the Secretory and Helper signatures, respectively. **D** Venn diagram of the 639 tumor cDC2 upregulated genes with the Secretory and Helper signatures defined in "**C**". **E** Supervised analysis of the 135 genes shared between tumor and pRNA "secretory" signature (light blue), 440 tumor specific genes (black), and the 64 genes shared between tumor and GM-CSF (yellow), using 3 gene lists: checkpoint and maturation markers (left, 148 genes), cytokines and chemokines (center, 169 genes), NFkB pathway (right, 100 genes). **F** Expression of selected genes in cDC2 from tumors and blood of HNSCC patients, central lines represent mean.

were increased in the juxtatumor sample (Supplementary Fig. 12B, Supplementary Table 9).

We used supervised gene lists to determine the specific function of each cluster (#). As expected, mmDC #20 expressed higher levels of chemokines and cytokines as compared to cDC2 #7 and cDC1 #23, such as *CCL19* and *CCL22*, *IL27*, *EBI3* (IL27B), and *IL32*, although *CXCL9* was also expressed by cDC1 and cDC2 (Supplementary Fig. 13A, B). Among the other cytokines of the Secretory phenotype, *IL12B* was detected in very few cells, but predominantly in mmDC (Supplementary Fig. 12C). *IL10*, *TNF*, and *IL1B* were mostly detected among MMAC (notably MMAC_PDL1), so that cDC were not the main source of those cytokines in the TME. mmDC expressed the checkpoints and maturation markers of Secretory cDC such as *CD274*, *PVR*, *CD40*, and *IDO1*, *CD200*, and *TNFRSF9* (4-1BB), consistent with our findings in bulk RNAseq (Supplementary Fig. 13C). The analysis of innate receptors expression (Supplementary Data 12) showed that mmDC specifically expressed *LY75* (DEC-205), a gene also identified in the bulk RNAseq cDC2 signature (Supplementary Data 8), whereas cDC1 expressed *CLEC9A* (Supplementary Fig. 13D). cDC2 expressed *CLEC10A*, similarly to the cDC2B subset previously observed in human by ScRNAseq in blood and one spleen sample[31], but we did not identify a *CLEC10A*neg cDC2A cluster. At the protein level, CLEC10A is known to be expressed in over 80% of cDC2 in blood, spleen, and thymus[32], and downregulated upon TLR7/8 maturation[33], consistent with the absence of expression on mmDC #20. Regarding transcription factors, mmDC selectively expressed *ZBTB10*, *IRF4*, *NFKB1*, *REL*, *ETV3*, *ARNTL2*. They also expressed *BATF3* although at a lower level than cDC1. cDC1 expressed *IRF8* and cDC2 expressed *SPI1*, *RUNX3*, and *ATF5* (Supplementary Fig. 13E). Altogether, mmDC aligned with the in vitro defined Secretory DC at the transcriptomic level.

**HNSCC mature Secretory cDC have an increased cell–cell communication network**. We used ICELLNET, a novel cell–cell communication tool adapted to single-cell transcriptomics, to compare the immature cDC2 #7 to the mmDC #20[34]. Communications were defined as IN when the central cell (#7 or #20) harbored the receptors and potentially received signals from peripheral cells expressing the ligands (Supplementary Data 13), and vice-versa for OUT communications (Supplementary Data 14) with each individual interaction scoring from 0 to 100. First, we used the sum of these scores to obtain the total scores and identify the main cell types interacting with cDC2 #7 and mmDC #20. For both, the highest OUT total scores were observed with Fibroblasts #22; CD8_Exhausted #15; CD8_Cytotoxic #5, and with TReg #6 for mmDC #20. OUT total scores were increased for #20 as compared to #7 with all the other cell types of the TME, with a global increase of +52% (+11% to +103%) (mean (min–max)) (Fig. 7A, B), which is in line with the Secretory phenotype. The highest increases in total OUT scores were observed with Fibroblasts #22; CD4 subsets #3, #4; TReg #6 and B #9 and Plasma cells # 8. To the contrary, there was little changes in total IN scores of cDC2 #7 and mmDC #20

(−2% (−20% to +18%) (mean (min–max)). For both, the highest IN scores were observed with Fibroblasts #22; mmDC #20; MMAC_PDL1 #0, and CD8_Exhausted #15.

Second, we used individual scores (Supplementary Data 13–14) to identify the specific and major receptor–ligand interaction changes occurring during maturation from #7 to #20, with some detailed below (Fig. 7C). mmDC expressed simultaneously several ligand-receptor pairs, such as *CCL19-CCR7*; *TNFSF9/TNFRSF9*; *SLAMF1/SLAMF1* allowing self-activation. Among mmDC receptors, *CCR7*, essential for DC migration to lymph node, was also predicted to interact with *CCL21* mainly expressed by Fibroblasts #22; *FLT3* with *FLT3LG* from CD4 T cells including TReg (#3-4-6) and CD8_Mem #10, an interaction important for DC differentiation; CD40 with CD40LG from CD4_Mem #4 and CD8_Exhausted #15; *TNFRSF11B* (osteoprotegerin) with *TNFSF10* from MMAC (#0-12), Neutrophils (#11) and Mast cells #14. Among mmDC ligands, predicted interactions had both inhibitory and stimulatory functions: *CD274* and *PDCD1LG2* (PD-L2) not only with *PDCD1* (PD-1) on CD8_cytotoxic #5 and CD8_Exhausted #15, but also with *CD80* on mmDC;[35] *PVR* with the negative checkpoints *TIGIT* on CD8 T cells and TReg and with the co-stimulatory *CD96* on all CD8 T cell subsets and NK cells (#5-10-15-17)[36,37]. As expected, mmDC #20 were predicted to induce immune cell recruitment via the expression of chemokines, such as MMAC (*CCL19/CCRL2*), CD4 (#3,4), TReg (#6) (*CCL22/CCR4*), and CD8_Cytotoxic (#5) (*CXCL9* and *CXCL10/CXCR3*). Among the cytokines expressed by mmDC (#20) and the most increased upon cDC2 maturation, *IL12B* and *IL23A* and were predicted to interact only with TReg (#6) and mmDC (#20), whereas *IL27* and *EBI3* were predicted to interact with various immune cell types. As observed by bulk RNAseq (Fig. 4), *IL10* was not among the genes increased upon cDC2 maturation within tumors with similar scores observed for cDC2 (#7) and mmDC (#20), a confirmed discrepancy with in vitro Secretory cDC. Finally, mmDC-derived *WNT5B* was predicted to interact with 10 receptors of the *FZD+LRP* family on fibroblasts with a score of 100, as opposed to 0 for the cDC2 #7, interactions described as promoting fibroblast activation[38].

In summary, a subset of cDC infiltrating HNSCC underwent a maturation that provided cDC with migratory, co-stimulatory, and inhibitory functions. These mmDC aligned with the in vitro inducible Secretory maturation. These mmDC had increased potential OUT interactions with most immune and non-immune cell types of the TME including numerous with fibroblasts, which is corresponding to increased ligands expression and is in line with the Secretory phenotype.

**Secretory cDC but not Helper cDC infiltrate tissues in cancer and autoimmunity**. We have identified PD-L1high ICOSLlow Secretory cDC in vivo in inflamed HNSCC, and PD-L1int ICOSLint immature DC in blood and as rare cells infiltrating non-inflamed HNSCC (Fig. 1D, E). To further explore the different maturation types according to the tissue and context, and to determine if a PD-L1low ICOSLhigh Helper cDC maturation could occur in vivo especially in contexts enriched in Helper cDC

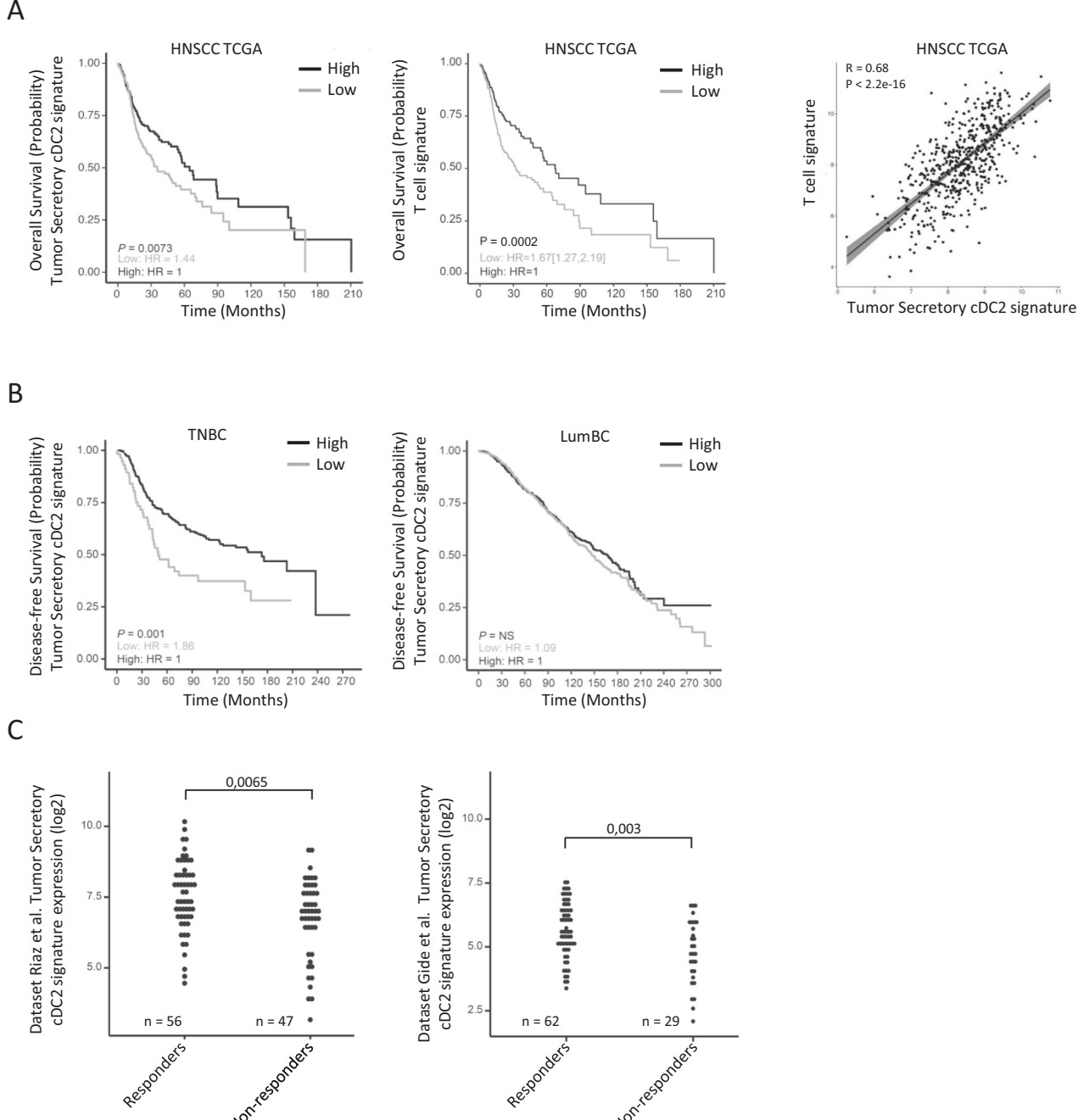

**Fig. 5 Tumor secretory cDC2 are associated to good prognosis in inflamed cancers and to response to immunotherapy. A** Survival analysis among patients expressing high (black) and low (gray) levels of the 36-gene tumor secretory cDC2 signature (left) and T cell signature (center) (cutoff at median, log-rank test), among 500 non-metastatic HNSCC patients from The Cancer Genome Atlas (TCGA). Right: Pearson correlation between the 2 signatures in the same dataset. Line represents linear regression; grey zone represents 95% confidence interval. *p*-value is for two-sided statistical analysis. **B** Survival analysis as in A left for 318 triple negative breast cancer (TNBC) patients (left) and 1407 luminal breast cancer (LumBC) patients (right) from METABRIC dataset[24]. **A, B**: HR hazard ratio, NS not significant. **C** Tumor secretory cDC2 signature expression among responders and non-responders melanoma patients treated by immune checkpoint blockade from previous studies[27] (left) and[28] (right), two-sided Mann–Whitney tests.

inducers such as TSLP in allergy[39], we merged single-cell transcriptomic datasets from different tissues and contexts (Supplementary Data 15): cancer (HNSCC (present study[40] Supplementary Fig. 14A, Supplementary Data 16); Luminal breast cancer (Supplementary Fig. 14B, Supplementary Data 17); Lung cancer[41]); atopic dermatitis (AD)[42] (Supplementary Fig. 14C; Supplementary Data 18); lupus nephritis[43]; blood from cancer patients[40,41]; and blood[40], skin[42], and tonsil[40] from healthy patients, with a total of 111 samples. We identified 13 clusters

(Fig. 8A) that were annotated according to their DEG (Supplementary Fig. 15–16; Supplementary Data 19–20), maturation markers expression (Fig. 8B), and tissue-specificity (Fig. 8C; Supplementary Data 21). Cluster #4 presented with the most abundant expression of maturation markers including *LAMP3, CD80, CD40* (Fig. 8B), the migration marker *CCR7*, and overexpressed the tumor Secretory cDC2 signature (Fig. 8D left), thus #4 was labeled mmDC. These cells were mostly abundant in cancer tissues, but also present in healthy and inflamed skin,

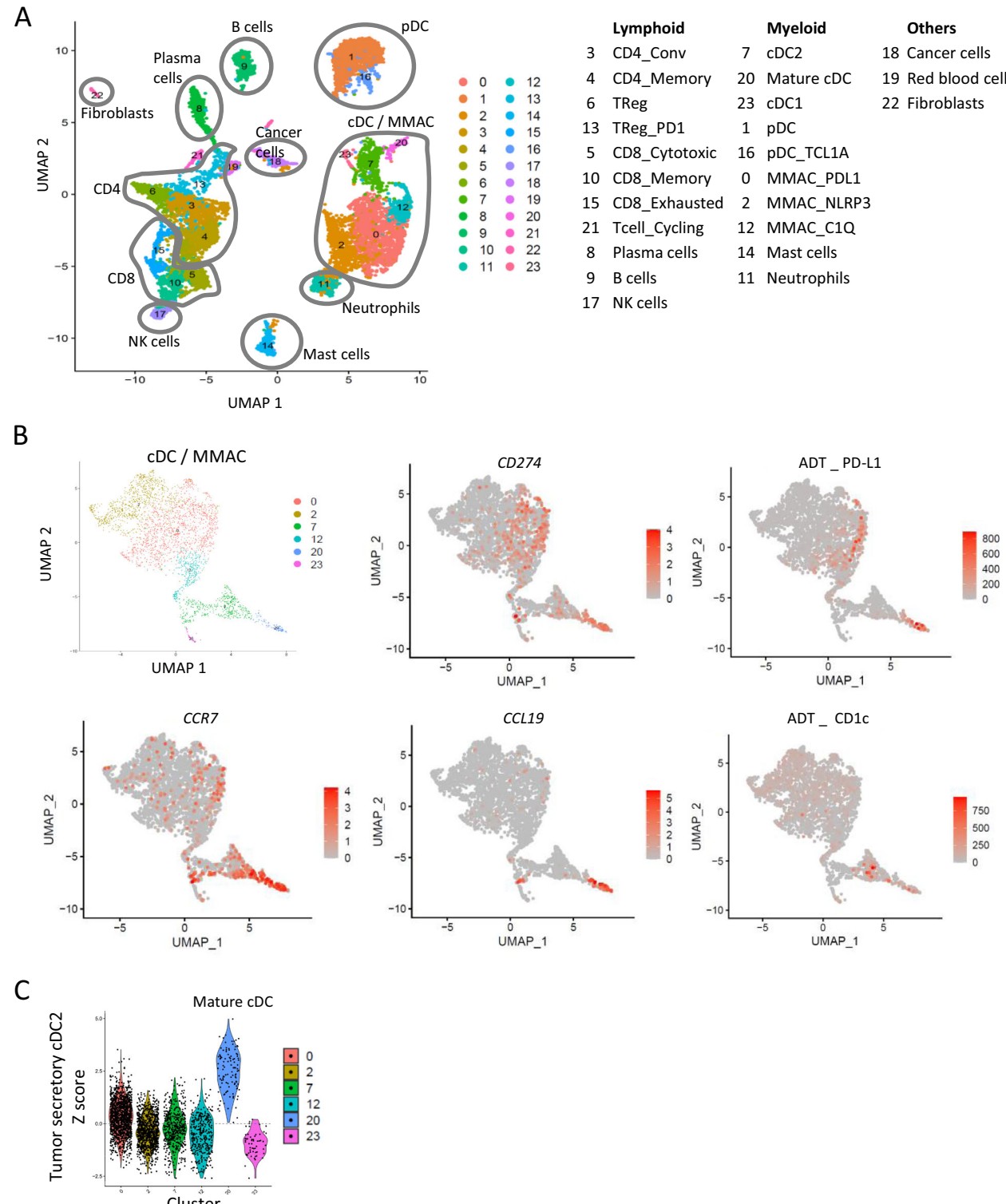

**Fig. 6 Tumor secretory cDC align with the mature migratory cDC subset in HNSCC. A** UMAP of the 10,503 cells analyzed by single-cell RNA sequencing displayed in 24 clusters. **B, C** Analysis of the 6 cDC/MMAC clusters as per 6A. **B** UMAP per cluster and expression of selected genes and antibody-derived tags (ADT). ADT staining was not present in all samples, see Methods for details. **C** Expression of the Tumor Secretory cDC2 signature. CD4 CD4 T cells, Conv conventional, TReg regulatory T cells, CD8 CD8 T cells, NK natural killer cells, pDC plasmacytoid DC, MMAC monocytes and macrophages.

in kidney/urine samples from Lupus patients, and in healthy tonsil. Cluster #1 and #8 also expressed maturation genes but not *CCR7* and were labeled mature non migratory tissue cDC2. Cluster #1 expressed *KLF4* as previously described in cDC2B, but not *RORC*[31], and *CD14* and *SOD2* as previously described DC3[44],

but not *C5AR1* nor *CD163*. Cluster #8 overexpressed the T-cell-attracting chemokine *CXCL10*, whereas mmDC #4 overexpressed *CXCL9*. Cluster #7 expressed *CD209* (DC-SIGN) but not *CCR7* and was preferentially from the AD skin samples. Cluster #5 expressed *CD40* and aligned to cDC1. Main immature cDC2

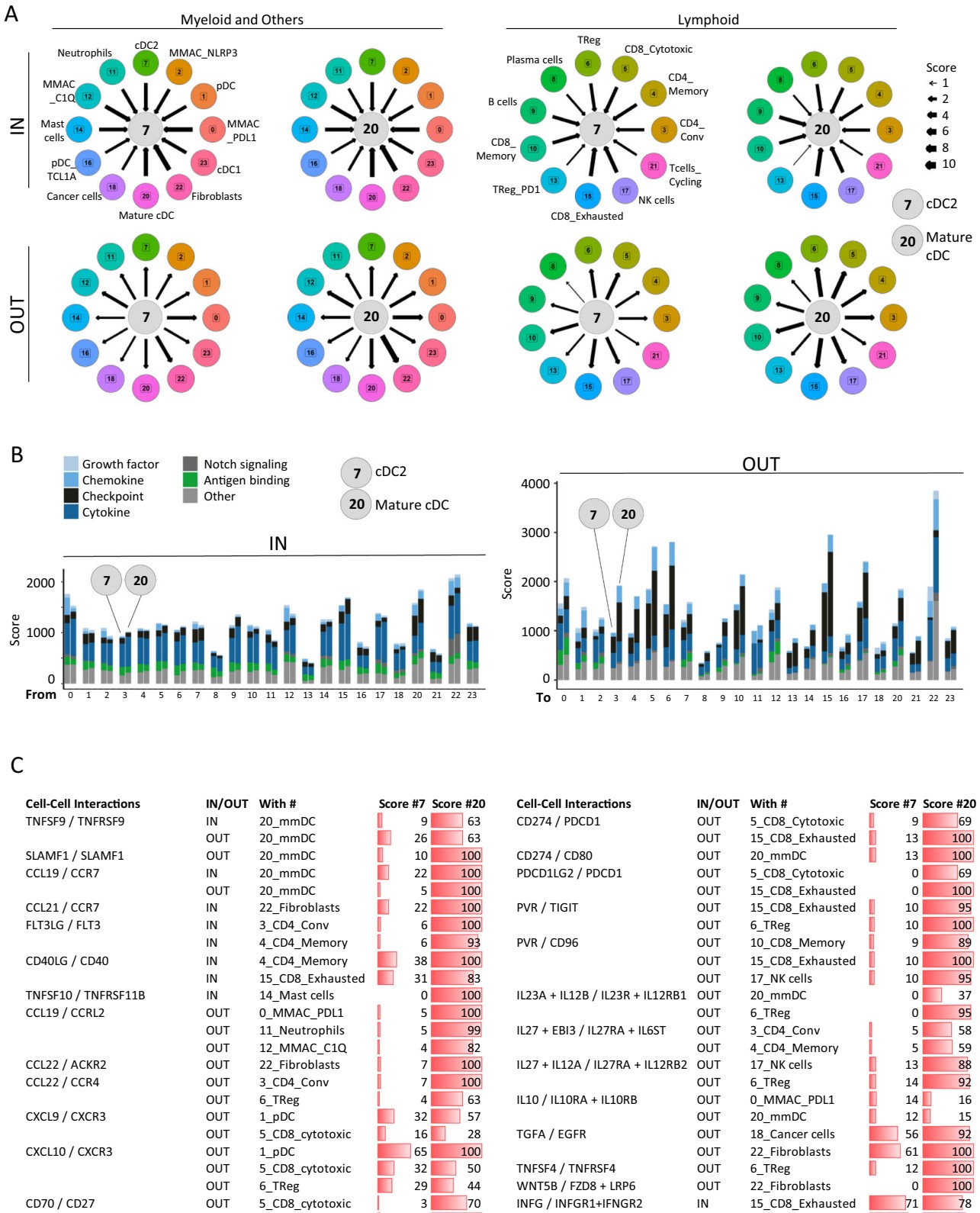

**Fig. 7 HNSCC secretory cDC have an increased cell–cell communication network.** Cell–cell communication analysis using ICELLNET[34] of cluster #7 and #20 as central cells in relation to other cell subsets of the tumor microenvironment. Cluster #19 of red blood cell was excluded from the analysis. **A** Networks. Interactions scores were normalized in a 0–10 scale represented by arrows. **B** Bar plots per interaction types. Bars are paired with the left bar being cluster #7 and the right cluster #20 that interact with the clusters numbered on the x-axis. Scores represent the sum of each individual interaction for each cluster–cluster interaction. **C** Table of selected interactions. Abbreviations: same as in Fig. 6.

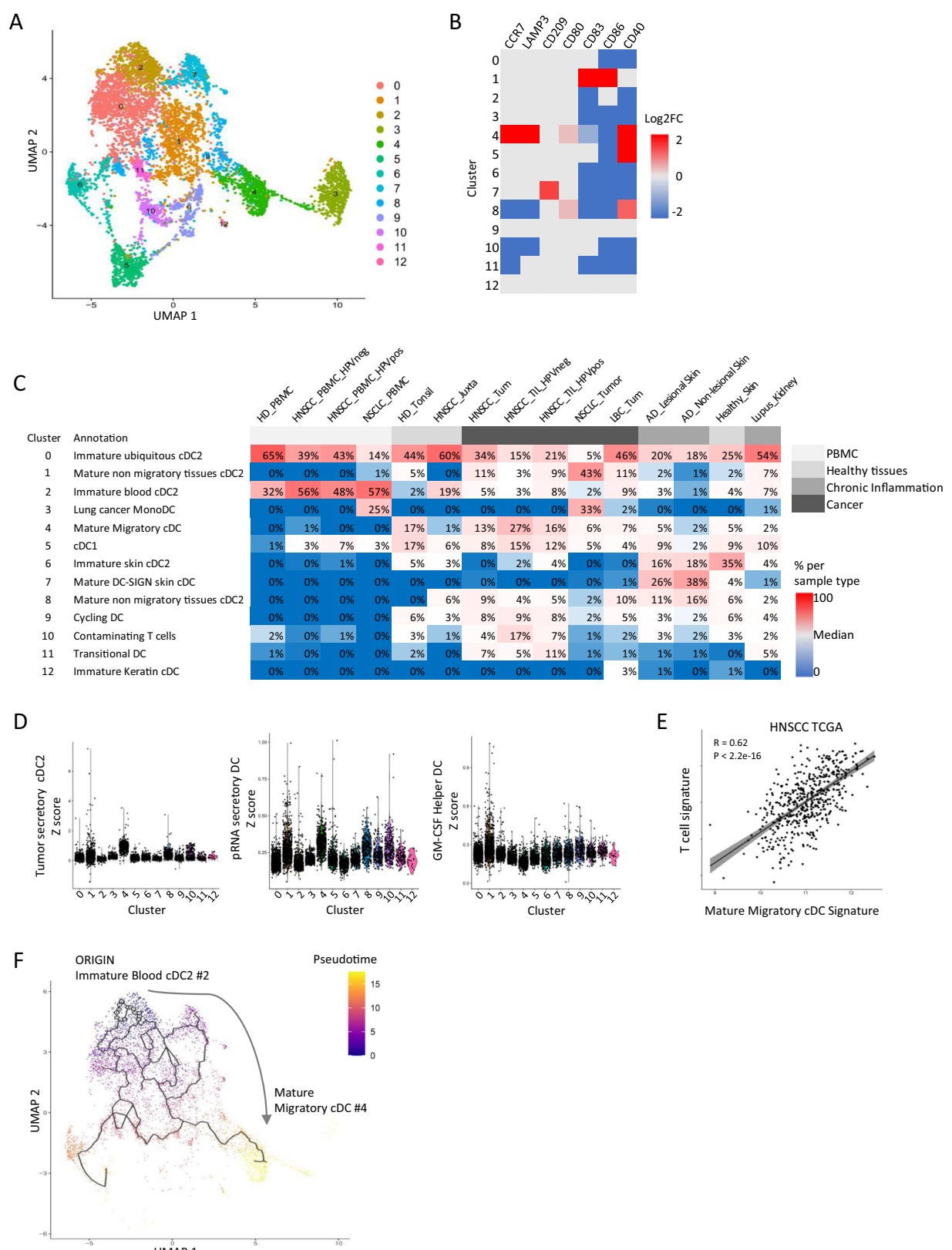

clusters were #2, mostly restricted to blood (Fig. 8C, Supplementary Fig. 17); the ubiquitous #0; and #6 preferentially found in skin. The minor cluster #11 expressed *AXL*, *IRF4*, *IRF7*, and *CD5* genes previously found in AS-DC, Pre-DC, or pDC[45,46], but was absent from blood samples; #11 better aligned to transitional DC[47] and were mostly observed in HNSCC and in Lupus.

The downregulation of TLR upon maturation was previously described in lung cancer[10]. Here, only *TLR2* was restricted to immature blood cDC2 #2 suggesting a possible downregulation upon maturation (Supplementary Fig. 15D).

As expected, #4 overexpressed the in vitro defined pRNA Secretory cDC signature as compared to all other clusters

**Fig. 8 Secretory cDC but not helper cDC infiltrate tissues in cancer and autoimmunity. A** UMAP of 6504 cDC merged from 6 datasets[40-43], (see Supplementary Data 15). **B** Heatmap of selected maturation genes per log2FC values. **C** Cluster annotation and distribution per sample type. PBMC peripheral blood mononuclear cells, HD healthy donor, HNSCC head and neck squamous cell carcinoma, Juxta juxtatumor, Tum tumor, NSCLC non-small cell lung cancer, LBC luminal breast cancer, AD atopic dermatitis. **D** Expression of the Tumor Secretory cDC2 (left), pRNA Secretory DC (middle), and GM-CSF Helper DC (right) signatures (n = 6504 cDC; ANOVA results are in Supplementary Data 22). **E** Pearson correlation between the Mature Migratory cDC signature derived from single-cell data and the T cell signature, among 500 non-metastatic HNSCC patients from the The Cancer Genome Atlas (TCGA). Line represents linear regression; gray zone represents 95% confidence interval, p-value is for two-sided statistical analyses. **F** Pseudotime analysis using Monocle 3 plotted on UMAP from A, with cluster #2 as origin, and excluding clusters #3, 5, 9, 10.

($p < 10E-6$, ANOVA) (Fig. 8D, Supplementary Fig. 15A–D; Supplementary Data 22), and genes from the NFkB pathway (Supplementary Fig. 15E). Clusters #1, #8, and #10 also overexpressed the pRNA Secretory signature significantly as compared to all other subsets excepted #4, with no significant difference among them ($p < 10E-4$, ANOVA). The in vitro defined GM-CSF Helper cDC signature was overexpressed in #1 as compared to all other clusters ($p < 10E-6$, ANOVA), showing that #1 expressed a mixed Secretory/Helper signature (Fig. 8D; Supplementary Data 22). No DC cluster overexpressed only the Helper DC signature in the present datasets.

We derived a mmDC signature from the single-cell data (see Methods and Supplementary Table 10). Correlation analyses with the CD3 T cell signature confirmed the expected enrichment in mmDC in inflamed tumors (Fig. 8E, Supplementary Fig. 18A–D).

Finally, we performed pseudotime analysis using Monocle 3[48] with immature blood cDC2 #2 defined as the origin (Fig. 8E). The clusters #9 of cycling DC, #3 of Lung Mono_DC, #5 of cDC1, and #10 of contaminating T cells were excluded from this analysis. We confirmed that cDC2 maturation followed the trajectory of #2, #0 immature ubiquitous cDC2, #1 mature non migratory cDC2 with a mixed Secretory/Helper phenotype, #8 mature non migratory cDC2 with a mild Secretory phenotype, and eventually #4 mmcDC with the strongest Secretory phenotype.

## Discussion

Dendritic cells are often described as tolerogenic in cancer[8,10,49,50], whereas they are thought excessively immunogenic in chronic inflammation[14] in which immunotherapy aims at inducing their tolerogenic function[18]. Classically, immunogenic DC secreted IL-12, IL-1b, and other proinflammatory cytokines, while inducing T effector cells production of IFNg and various combination of other Th cytokines[2]. Conversely, tolerogenic DC secreted IL-10, and TGFb, or no cytokines, and could induce regulatory T cells[3,4,51,52]. However, such a classification remains controversial and does not reflect several physiological and pathological conditions. For example, TNF was shown to be produced both by immunogenic and tolerogenic DC[53,54]. IL-10 can be induced in DC by a number of microbial stimuli promoting strong immunity, for example bacterial LPS[55]. TSLP-activated DC were extensively shown to be immunogenic, and proinflammatory, while they neither produce IL-12 nor IL-1b[56]. Hence, the global link between DC-derived secreted factors, immunogenicity, and Th cytokine production remains unclear.

In this study, we uncovered a novel functional classification of human cDC that mature according to two distinct programs: Secretory DC that predominantly secreted chemokines and cytokines as opposed to Helper DC that predominantly induced Th cytokine secretion after co-culture. DC-derived and Th-derived cytokines did not match the classical tolerogenic and immunogenic classification. PD-L1 and ICOSL were simple markers to identify respectively Secretory and Helper DC at the protein level. Secretory cDC were transcriptionally identified in cancer and chronic inflammation and phenotypically in HNSCC and corresponded to the mmDC LAMP3+subset. Here we show

that mmDC share a similar transcriptomic program in cancer, lupus, and atopic dermatitis, which sheds a new light on the tolerogenic/immunogenic dichotomy, raises doubts on its existence in vivo, and has therapeutic implications.

Regarding the classification of in vitro cDC maturation, each Secretory or Helper phenotype was induced by specific stimuli, but not restricted to a single receptor pathway. The main Secretory cDC inducers were R848, a TLR7/8 ligand; Zymosan, a TLR2 and Dectin-1 ligand; HKSA and HKLM, TLR2 ligands, that signal through the MyD88/MAPK/NFkB and the CARD9/NFkB (Dectin-1) pathways. The main Helper cDC inducers were Pam3 at low dose, also a TLR2 ligand; Flu that activates TLR7 and cytosolic sensors and signals through the MyD88/MAPK/NFkB pathway and the Caspase-1; IPS1/TRAF3/IRF3 pathways; and the cytokines GM-CSF and TSLP that bind their respective receptors[57]. The exact mechanisms by which a cell will present with different outputs after stimulation by two different ligands binding the same receptor or activating the same pathway, such as Myd88/NFkB, remains to be fully elucidated. Besides, although we tested a large panel of innate stimuli, we raise the question of whether other stimuli, dose, duration, would induce cDC falling in similar categories, which would propose that the Secretory versus Helper functional dichotomy may be used as a universal classification system for activated cDC. This classification is instrumental for the comparison of different cDC activation states in physiopathology and immunotherapy. Indeed, several of the receptors listed above are targeted by pharmaceutical compounds that are approved, such as TLR7/8 ligand Imiquimod, or being tested such as TLR7/8 ligand reformulated Resiquimod[58], TLR1:2 ligand Amplivant[59] and Dectin-1 ligand Imprime PGG[60], or used for DC maturation prior to adoptive cell therapy such as GM-CSF[5]. Zymosan has shown some preclinical efficacy in a mice model of melanoma, but no clinical trial is ongoing with this Secretory cDC inducer[61]. Our data may guide the selection of such compounds and we would recommend controlling for the differential effects of various doses. Regarding the impact of cDC subsets, our in vitro Secretory/Helper classification was defined on primary blood cDC encompassing mostly cDC2 and some cDC1. Both subsets diverge on their expression of receptors for the stimuli tested. For cDC2, we confirmed our conclusions and the physiopathological relevance by RNAseq and ScRNAseq and by in vitro validation with sorted pure cDC2. For cDC1, our data does not allow to conclude—nor exclude—that such dichotomy also applies to cDC1. However, we and others[10,30,62], observed that the transcriptome of cDC1 and cDC2 converged upon cancer-induced maturation. cDC1- and cDC2-derived LAMP3 + DC may express different surface receptors, but these two cell types cannot be distinguished at the transcriptomic level[30] and staining with CD141 and CD1c ADT is required to identify the subset of origin[10]. This transcriptomic convergence raises the question of the existence of distinct functions of these two cell types once they have reached this level of migratory maturation.

In vivo in cancer and chronic inflammation, DC maturation is induced by damage and pathogen associated molecular patterns,

inflammatory cytokines, cytosolic sensor ligands such as cyclic GMP-AMP Synthase (cGAS)[57], antibodies, immune complexes[14], and by allergens[63]. These stimuli act simultaneously to activate DC but the exact contribution of each stimulus in largely unknown. Here we show that, beyond in vitro cDC maturation states, our classification is relevant in vivo: Secretory cDC aligned with mmDC from various tissues and anatomical sites, and independently of the context of cancer, allergy, or autoimmunity. This observation has several implications for the understanding of physiopathology. First, Secretory cDC inducers listed above induce a maturation state more closely mimicking the in vivo state than Helper cDC inducers, which may guide stimuli selection for therapy depending on the context. Put differently, we may conclude that the signals inducing spontaneous cDC maturation in cancer or other types of chronic inflammation converge with the chemical Secretory DC inducers. Second, Secretory cDC inducers seem to dominate Helper inducers when co-existing. We have not tested such combination in vitro, but in cancer, high levels of GM-CSF[64] co-exist with other cancer-derived activating signals and the final output was a Secretory cDC phenotype. The same was true for allergy and TSLP[39].

We may summarize the Secretory cDC signature as the combination of maturation markers (*CD40, CD70, CD80, LAMP3*), migratory marker *CCR7*, chemokines (*CCL19, CXCL9*), cytokines (*IL12B, IL23A, EBI3*), the receptor *LY75* (DEC-205), checkpoints *CD274* (PD-L1), *PDCD1LG2* (PD-L2), *PVR, IDO, CD200, TNFRSF9* (4-1BB), and the transcription factors *ETV3/ETV3L* and *REL/RELB*. The obvious co-expression of stimulatory and inhibitory molecules raised the question of the output of DC–T cell interaction. In vitro, Secretory cDC had a limited potential to induce CD4 Th cytokine production mostly like unstimulated medium-cDC and strongly reduced as compared to Helper cDC. Helper cDC did not induce a bias in Th cytokine production towards a Th1, Th2, or TReg phenotype, but rather a global increase in the secretion of the different cytokines. However, the capacity to induce T cell proliferation of Secretory and Helper cDC were close. The essential role of CD4 T cells in the antitumor response is now clear[65]. In vivo in HNSCC, Secretory cDC had predicted interactions being both stimulatory and inhibitory with CD4 T cells, such as *CD80/CD28* and *PVR/TIGIT*. Th cytokine genes were not identified among CD4 T cells DEG. Whether Helper cDC could be chemically induced in vivo and would increase CD4 T cell cytokine production remains to be shown. Cytotoxic CD8 T cells are also essential for the antitumor immune response[66], and we confirmed that the concept of Secretory cDC also applied to CD8 T cells in vitro. Besides, the transcriptomic data in HNSCC showed that Secretory cDC had predicted interactions also being both stimulatory and inhibitory with CD8 T cells. Even for well-known molecules such as PD-L1, it is difficult to conclude if it has a pro- or antitumor role in the tumor microenvironment: PD-L1 on cDC may bind PD-1 on cytotoxic and exhausted CD8 T cells and have an inhibitory effect, and simultaneously bind in cis to CD80 and have a sti-mulatory effect while avoiding T cell inhibition[35]. A recent study in mouse using PD-L1$^{KO}$DC that were also deficient in PD-L2 showed that the PD-L1/PD-1 inhibitory effect dominated the cis PD-L1/CD80 activating effect[67]. These predicted interactions were identified using transcriptomic data and would need to be validated in vivo.

A classical view of cDC maturation includes the sequential performance of innate and adaptive functions. Some studies have suggested that a strong cDC activation would induce both innate and adaptive functions at high levels. For example, microbial stimuli induce large amounts of IL-12 secretion by cDC, and subsequently large amounts of interferon gamma by T cells[55]. We now provide systematic and definite evidence that innate and

adaptive cDC functions can often be dissociated. IL-12 plays an important role in CD8 T cells and NK activation and stimulation of IFNg production[68], but it is notable that, in the present HNSCC dataset, *IL12B* was predicted to mainly bind to mmDC themselves and to TReg, in association with *IL23*, but not on CD8 T cells. Here, the main mmDC cytokines predicted to interact with CD8 T cells were *IL27* and *IL15*, both known for their capacity to induce IFNg production[69,70]. One discrepancy between the in vitro and the in vivo Secretory cDC involved IL-10: it was highly produced at the protein level in vitro and expressed by the pRNA-DC at the transcriptomic level but not by tumor cDC2 in RNAseq data. In ScRNAseq from HNSCC, *IL10* was only weakly detected by in cDC as compared to the expression by MMAC. In the ScRNAseq merged datasets analyzed in other cancers and chronic inflammatory diseases, *IL10* was not among the DEG. However, *IL10/IL10R* binding was predicted to occur with a moderate score as an autoloop on mmDC and as an IN signal from MMAC, so that it may still contribute the induction of some tolerogenic features of Secretory cDC in vivo[3]. A recent study in lung cancer proposed that IL-12 secretion was in part inhibited by IL-4 signaling in cDC1[10]. Therefore, we tested in vitro the effect of a multiblocking of the IL-10 autoloop, the IL-4 signaling, the PD-1/PD-L1 and PDL2 axis and the TIGIT/PVR axis of Secretory cDC. This multiblocking only partially and weakly restored the CD4 Th cytokine secretion suggesting the existence of other mechanisms of inhibition.

We showed that Secretory cDC2 were associated to a T-cell-inflamed TME in HNSCC, to a good prognosis in two types of inflamed cancers, HNSCC and TNBC[25,26,71], but not in a non-inflamed cancer, LumBC[25,26], and to response to checkpoint blockade in melanoma. A previous study identified a 18-gene signature that was used as a surrogate signature for TME inflammation and was predictive of the response to checkpoint blockade[72]. It is notable that several genes characteristic of Secretory cDC2 (*CXCL9, IDO1, CD274,* and *PDCD1LG2*) were found in this signature. *CCL19, CCR7, LAMP3,* and *CD86* were also found in the signature of tertiary lymphoid structures (TLS) associated to response to immunotherapy in melanoma[73]. Therefore, high infiltration in CD3, mature Secretory cDC2 and TLS appear as linked key features of a favorable TME for clinical outcomes in several cancer types. This association between tumor Secretory cDC2 and T cell inflammation suggests that T-cell-attracting chemokines, such as CXCL9, are not only overexpressed by Secretory cDC2 at the transcriptomic level but also at the protein level in cancer.

Moreover, our Secretory cDC2 signature and analysis of cell–cell communication networks in the TME may guide the selection of targets for immunotherapy. Targeting PD-L1 and PD-L2 has already proved beneficial, and this is at least partially by the effect on cDC[67], but most other targets such as PVR / TIGIT are still under investigation. It is still unknown whether targeting PVR or PVRIG/TIGIT would be preferable, and one may want to preserve the activating PVR/CD96 interaction[36,37]. Secretory cDC2 also overexpressed CD40 that is targetable with CD40 agonists and has shown to be a promising treatment, acting independently of the type I IFN, TLR, and cGAS-Sting pathways[74], which offers opportunities for combination therapies.

Aside of immune checkpoints and pattern recognition receptor targeting, the differentiation molecule FLT3 ligand is known to promote antitumor immunity[7] and currently tested in multi-combinations strategies[5]. Here we showed that *FLT3/FLT3LG* binding was predicted to occur spontaneously in the TME, and that *FLT3LG* was expressed at the highest levels by CD4 T cells, TReg and CD8_mem, and not predominantly by NK cells as previously observed[75]. Beyond DC–T cell interactions detailed in

HNSCC in our study and previously described in hepato-cellular carcinoma[76], we revealed unexpected numerous interactions between mmDC and other immune and non-immune cells. mmDC had the highest predicted interactions with Fibroblasts. Among them, CCL21/CCR7 favored mmDC migration, similarly to a previous observation in atopic dermatitis (CCL19/CCR7)[42]. We also unraveled new mmDC-Fibroblast interactions such as WNTB5/FZD + LRP. These predicted interactions pave the way for further studies on the mmDC-Fibroblast interplay in cancer.

Altogether our study shows that two main maturation types may be obtained in vitro that have dissociated innate and adaptive functions and were named Secretory and Helper cDC, respectively. This dichotomy in cDC maturation may be considered as an equivalent of the Th1/Th2 dichotomy for CD4 T cells. cDC maturation modulation is a promising therapeutic approach in many diseases with chronic inflammation. With the number of pharmaceutical compounds available and the excessive number of possibilities for combinations, a strong rational is needed to select a priori the best treatment options to test in clinical trials. Our data reveal many targets and interactions that may guide selection of optimized combination therapies. cDC maturation in vivo largely converged across tissues and diseases and matched the in vitro Secretory cDC phenotype. cDC-based therapeutic approaches in cancer and non-cancer chronic inflammation therefore have similar targets, although they may be used differently to induce activation or inhibition on demand.

## Methods

**Human samples and patient characteristics**. Fresh samples of HNSCC tumor tissues and blood of untreated patients with head and neck cancers were obtained from the pathology departments of the Institut Curie and the Pitié-Salpêtrière hospital. Patient characteristics for the flow cytometry cohort (Fig. 1) and RNAseq cohort (Fig. 4) are summarized in Supplementary Tables 2 and 7, respectively. Fourteen of the 22 patients of the flow cytometry cohort were included in the observational clinical trial SCANDARE NCT03017573. For the ScRNAseq samples, both HNSCC patients presented with an HPV negative (PCR) HNSCC of the oral cavity. Patient 1 was aged 81, non-smoker, non-drinker, and had a T4aN0M0 tumor; Patient 2 was aged 44, smoker, non-drinker, and had a T3N0M0 tumor. For the ScRNAseq analysis of DC maturation merging various datasets, we added a sample obtained from a 44-year-old female that underwent surgery for an untreated luminal breast cancer, and similarly provided by the pathology department of the Institut Curie. All samples were obtained in accordance with the ethical guidelines, with the principles of Good Clinical Practice and the Declaration of Helsinki, and with patients' consent. This study was approved by the Internal Review Board and Clinical Research Committee of the Institut Curie.

**Single-cell suspensions**. Tissue samples were mechanically and enzymatically digested in $CO_2$-independent medium (Gibco) containing 5% FBS (HyClone). Enzymatic digestion consisted of three rounds of 15 min of incubation with agitation at 37 °C, separated by pipetting, with 2 mg/ml collagenase I (C0130, Sigma), 2 mg/ml hyaluronidase (H3506, Sigma), and 25 µg/ml DNAse (Roche). The samples were filtered on a 40 µm cell strainer (Fischer Scientific) and were diluted in PBS 1× (Gibco) supplemented with EDTA 2 mM (Gibco) and 1% decomplemented human serum (BioWest). After two washes, cells were suspended in PBS before being stained for flow cytometry or flow sorting. PBMC were isolated from blood samples using FICOLL (GE Healthcare) gradient centrifugation.

**Antibodies, flow cytometry, and cell sorting**. Single-cell suspensions from digested tissue samples and from blood were stained with antibodies (listed in Supplementary Data 11) for 15 min at 4 °C. After washing step, cells were analyzed or sorted directly, immediately after having added DAPI (Miltenyi Biotec) for dead cells exclusion. Flow cytometry phenotyping was performed on BD LSR Fortessa Analyzer. Cell sorting for the bulk and ScRNAseq experiments were performed on BD FACS Aria III using the purity and low-pressure mode, and a 100 µm nozzle. For the bulk RNAseq experiment, DC subsets and MMAC were sorted in Eppendorf tubes containing TCL buffer (Qiagen) supplemented with 1% β−mercaptoethanol (SIGMA) before RNA extraction, as described in Michea P, Noël F et al.[20]. For the ScRNAseq experiment, cells were sorted in Eppendorf tubes containing RPMI 1640 Medium Glutamax (Life Technologies) enriched with 10% Fetal Calf Serum (Hyclone), 100 U/ml Penicillin/Streptomycin (Gibco), 1% MEM Non-Essential Amino Acids (Gibco), and 1% pyruvate (Gibco).

**Analysis of flow cytometry data**. We measured a total of 434 parameters including 52 cell/cell ratios analyzed by FlowJo V10. We established a sublist of 81 non-redundant parameters, in which each population was expressed only in percentage of its parental population. The list of 81 parameters was used in Fig. 1D, and the list of 434 parameters enriched with 14 clinical parameters was used for the elastic net model in Supplementary Fig. 3B. The elastic net model was performed using R software, a Lambda at 1SE, and an alpha of 0,5.

**In vitro analysis**. For Fig. 2 and Fig. 3A, B, material and methods were described in details in the resource paper from Grandclaudon et al.[21]. We used only the primary blood cDC (referred to as bDC) in this database and excluded monocyte-derived DC and plasmacytoid DC. These cells were sorted as CD3−, CD14−, CD16−, CD19−, HLA-DR+, CD4+, CD11c+ and were composed of around 80% of cDC2, and less than 10% of cDC1[77]. As compared to the resource paper containing 118 data points for primary blood DC, we generated supplementary experiments and analysis to specifically address our question. We added 36 data points for the analysis of surface markers (leading to a total of 154 data points) among which 12 for the analysis of cDC secreted cytokines and chemokines and of the Th cytokines (leading to a total of 130 data points). Extra data points included: Curdlan 10 ug/ml ($n=1$), Flu (1X) ($n=3$), Flu(1X) + TSLP (50 ng/ml) ($n=3$), HKSA (MOI10) ($n=3$), GM-CSF 50 ng/ml ($n=4$), LPS ($n=3$), Medium ($n=9$), Poly I:C 50 ug/ml ($n=4$), R848 1 ug/ml ($n=3$), TSLP 50 ng/ml ($n=3$), for a total of 29 blood donors. The antibodies used for the checkpoints and maturation markers analyzed by flow cytometry are listed in Supplementary Table 4. For the cDC secreted cytokines and chemokines, we measured 24 supplementary cytokines and chemokines. IL1a, IL1b, IL6, IL10, TNFa, and IL12p70 were measured by cytometry bead assay flex set (CBA) and we added the measure of IFNa. IL23 and IL28a were measured by Luminex and we added the measure of APRIL, BCA1, CCL19, CXCL11, CXCL16, CXCL9, Eotaxin2, I309, IFNb, IL12p40, IL16, IL1RA, IL27, IL29, IP10, MCP1, MCP2, MCP4, MIP1a, RANTES, TARC, TRAIL, YKL40 (Supplementary Table 5). The 17 Th cytokines were analyzed by CBA or Luminex (Millipore) (Supplementary Table 6), similarly to the resource paper.

For validation, we performed new similar experiments using pure cDC2 sorted as shown in Supplementary Fig 5A: CD3− CD14− CD16− CD19− CD20− CD56− CD123− CD11c+ CD1c+. Before the sorting, a step of DC pre-enrichment was performed on PBMCs using the EasySep™ Human Pan-DC Pre-Enrichment Kit (StemCell Technologies). cDC2 were activated by R848 1 ug/ml or TSLP 50 ng/ml or not activated (Medium: cultured in RPMI—GlutaMax supplemented with 10% FBS, 1% MEM NEAA (Gibco), 1% Sodium Pyruvate (Gibco)). cDC2 secreted cytokines were measured by CBA in cDC2 supernatant after 24 h. cDC2 were then co-cultured with naïve CD4 T cells (Fig. 3C, Supplementary Fig. 5B, 6A) or naïve CD8 T cells (Supplementary Fig. 6B) at a 1:5 ratio in Xvivo15 medium (Lonza) for 6 days. At day 6, T cells were restimulated 2.5 ul of anti CD3/CD28 beads (Thermofisher) for 100,000 T cells. At day 7, CD4 or CD8 T cell secreted cytokines were measured by CBA in the supernatant. CD4 or CD8 T cells were purified from healthy donors PBMC using EasySep Human Naive CD4 + T cells Isolation Kit or EasySep™ Human Naïve CD8 + T Cell Isolation Kit II (StemCell Technologies) respectively.

For blocking experiments (Fig. 3C, Supplementary Fig. 6A), cDC2 (when use of aIL10) and T cells (for other antibodies) were cultured separately for 1 h with blocking antibodies or the corresponding isotypes before starting the co-culture. Blocking was performed using anti-PD1 (10 µg/ml), anti-TIGIT (10 µg/ml), or anti-IL4R (20 ug/ml) individually or combined together and with anti-IL10 (10 µg/ml) and anti-IL10R (10 µg/ml) for the multiblocking. The IL-10/IL-10R only blocking experiment (Supplementary Fig. 6B bottom row) was performed independently. All material details are provided in the Supplementary Data 11.

**Statistical analysis**. Statistical analyses were performed using the following softwares: Excel (Microsoft); Qlucore version 3.5; GraphPad Prism 8 (GraphPad Software Inc.); R versions 3.5.3, 3.6.1, and 3.6.3 softwares. Correlations analyses were Pearson (Fig. 5A right, Supplementary Fig. 9A–D right; R) or Spearman (Figs. 1D, 2E; Prism; Excel) correlations for parametric and non-parametric data, respectively. Kruskal–Wallis tests were used for multiple comparisons of unpaired non-parametric data, combined with a Tukey post-hoc test (Figs. 2D, 3A; Qlucore) or a Dunn's post-hoc (Fig. 3B, Supplementary Fig. 4A–D; Prism). ANOVA was used for multiple comparisons of unpaired parametric data and comparison of signatures expression (Fig. 8D, Supplementary Fig. 8C; R). Wilcoxon tests were used for two-group comparisons of paired non-parametric data (Fig. 3C, Supplementary Fig. 5B middle-bottom, Supplementary Fig. 6A, B; Prism). Mann–Whitney tests were used for two-group comparisons of unpaired non-parametric data (Fig. 5C, Supplementary Fig. 5B top, 9C, D left; Prism; R). T-sne with a perplexity of 15 was used for unsupervised analysis of flow cytometry data in Fig. 2C (Qlucore). UMAP was used for unsupervised clustering of single-cell transcriptomic data, with the parameters mentioned in the corresponding paragraph (Fig. 6A, B, 8A, Supplementary Fig. 14A–C; R). Survival analyses used log-rank tests (Fig. 5A, B, Supplementary Fig. 9A, B left; R). Data were considered significant for p-values superior to 0.05. When not provided as values, p-values are represented by range: * < 0.05, ** < 0.01, *** < 0.001, **** < 0.0001.

**Bulk RNA extraction, sequencing, and data processing**. Material and methods are described in details in the resource paper[20]. Briefly, single Cell RNA Purification Kit (Norgen Bioteck) was used for RNA extraction, including on-column DNase digestion (Qiagen), as described by the manufacturer's protocol. RNA integrity was controlled with Agilent RNA 6000 Pico Kit (Agilent Technologies) in BioAnalyzer. cDNA was generated with SMARTer Ultra Low input RNA for Illumina Sequencing-HV (Clontech), following manufacturer's protocol with 14 cycles for amplification. Quality controls were performed with Qubit dsDNA high sensitivity (Thermofisher) and an Agilent RNA 6000 Nano Kit (Agilent Technologies). Multiplexed pair-end libraries 50nt in length were obtained using Nextera XT kit (Clontech). Sequencing was performed in a single batch with Illumina HiSeq 2500 using an average depth of 15 million reads. Library, sequencing, and quality controls were performed by the NGS facility at the Institut Curie. Reads were mapped to the human genome reference (hg19/GRCh37) using Tophat2 version 2.0.14. Gene expression values were quantified as read counts using HTSeq-count version 0.6.1. Genes with less than one read count in at least one sample were filtered out. The remaining raw data were normalized and analyzed using DESeq2 R package version 1.26.0 (Fig. 4) or by ANOVA (Supplementary Fig. 8). Differentially expressed genes were obtained with an adjusted *p*-value of 0,10 (Fig. 4) or 0.05 (Supplementary Fig. 8). The supervised list of genes used in Fig. 4E were established by including all markers analyzed at the protein level in the in vitro analysis and by adding other known checkpoints and maturation markers, cytokines, and chemokines from literature search. The NFkB pathway genes list was established by literature search.

**Analysis of bulk transcriptomic public data**. Mathan et al. dataset[22] used in Fig. 4 was downloaded from Gene Expression Omnibus (GSE89442) as annotated count data and was analyzed by DESeq2 R package. This dataset included three samples per condition: unstimulated; pRNA; GM-CSF. TCGA data were downloaded from https://portal.gdc.cancer.gov and included 502 patients. Two patients with metastasis at the time of diagnosis were excluded, so that the survival analysis presented in Fig. 5A includes 500 patients. Counts were normalized using R package DESeq2. METABRIC data[24] used in Fig. 5B and Supplementary Fig. 9A, B was provided by Dr Anne-Sophie Hamy-Petit, Institut Curie, Paris, FRANCE. Patients without complete survival data were excluded. The two genes *ANKRD33B* and *WFDC21P* from our 36 genes of tumor secretory cDC2 were missing in this dataset. RNAseq data from melanoma patients treated by immunotherapy from Riaz et al.[27] (Fig. 5C left, Supplementary Fig. 9C left) were downloaded as annotated count data from https://github.com/riazn/bms038_analysis/tree/master/data. The gene *ABP1* from our 36 genes of tumor secretory cDC2 was missing in this dataset. RNAseq data from melanoma patients treated by immunotherapy from Gide et al.[28] (Fig. 5C right, Supplementary Fig. 9 right) were downloaded as annotated count data from https://github.com/miabioinformatics/Gide_CancerCell2019. The gene *NECTIN2* from our 36 genes of tumor secretory cDC2 was missing in this dataset. For both datasets of immunotherapy-treated patients, responders included patients with complete and partial responses and stable disease, and non-responders included patients with progressive disease. In Fig. 5 and Supplementary Fig. 9, the comparison and Pearson correlation of the Tumor Secretory cDC2 signature with T cell infiltration was performed using a 5-gene T cell signature including *CD3D*, *CD3E*, *CD3G*, *ZAP70*, and *PTPRC*.

**Processing of samples for ScRNAseq**. For HNSCC patient 1 (Pt1), there were single-cell suspensions from a juxtatumor sample and from a tumor sample. The two samples were stained with antibodies for cell sorting for 15 min at 4 °C (Supplementary Data 11). The samples were washed, and DAPI was added just before sorting. After sort, the different cell populations sorted were merged in defined proportions detailed below for each of the two samples. The samples were stained with CITE-seq antibodies (Supplementary Data 11) at 4 °C for 30 min according to manufacturer's recommendations (Biolegend). Cells were washed, and the tumor sample was split in two. In one of the samples, the primer specific for ADT was not added, to avoid the expansion by PCR of ADTs. This separation of the tumor sample in two was done to allow for comparison of tumor samples with or without ADT, because this was a new technology. We did not find any relevant difference between the samples with and without ADT, validating the absence of bias induced by this technological approach. Samples were then processed as recommended in 10X Genomics protocols. For HNSCC patient 2 (Pt2), there was only one tumor sample available. Staining with antibodies for sort and with CITE-seq antibodies were done simultaneously for 30 min at 4 °C, before wash, DAPI staining and sorting. Sorting was performed using the purity and low-pressure mode, and a 100 μm nozzle on a BD FACS Aria III flow sorter for all samples.

The gating strategy is displayed in Supplementary Fig. 10. We excluded debris and doublets by FSC, SSC gating; dead cells by DAPI gating; red blood cells by CD235a staining (Pt2 only). All remaining cells were sorted in one of the five final population groups. Group 1 included CD45+CD3+ T cells; Group 2 included non-immune CD45− cells; Group 3 included MMAC and CD14+DC gated as CD45+CD3−CD19−CD56−CD11c+HLA-DR+CD14;+ Group 4 included cDC1 and cDC2 gated as CD45+CD3−CD19−CD56−CD11c+HLA-DR+CD14− and was merged with pDC gated as CD45+CD3−CD19−CD56−CD11c−HLA-DR+ CD123;+ Group 5 included all remaining cells: CD45+CD3−CD19+ and/or

CD56+, CD45+CD3−CD19−CD56−CD11c−HLA-DR+CD123−, and CD45+CD3−CD19−CD56−HLA-DR−.

The count provided by the flow sorter was used for subsequent processing. The group with the highest number of cells was also counted by trypan blue to ensure accuracy of the flow sorter counts. The final sample used for chip loading ScRNAseq contained 18,000 (Pt1) or 16,000 (Pt2) cells obtained by combining 3600 (Pt1) or 3200 (Pt2) cells from each of the five groups. The only exception was for the juxtatumor sample of Pt1, in which only 857 cells were available for the Group 4 of cells.

For the luminal breast cancer sample, the single-cell suspension from the tumor sample was stained with antibodies for cell sorting for 15 min at 4 °C (Supplementary Data 11). The sample was washed, and DAPI was added just before sorting. The gating strategy for DC included the following steps: (i) exclusion of debris and doublets by FSC, SSC gating; (ii) exclusion of dead cells by DAPI gating; (iii) selection of cells positive in CD45 BV570; (iv) exclusion of T cells, identified as CD3+ APC and CD19− Alexa700 and CD56− Alexa700; (v) exclusion of CD19+ Alexa700 and CD56+ Alexa700 cells; (vi) selection of CD11c− PECy5 cells, among which cells CD123+ BV650 and HLA-DR+ eFluor760 double positive were sorted as pDC; besides, selection of CD11c+ PECy5 and HLA-DR+ eFluor760 double-positive cells, among which cells were classified according to CD1C PE and CD14 FITC: MMAC defined as CD1C− and CD14+ were excluded and all the other cells were sorted as cDC. Eventually, the DC sample used for ScRNAseq included the pDC and the cDC. The CD45− cells, CD3+ cells, the MMAC and a mixed group of CD45+CD3− cells were also sorted independently but were not analyzed in the present study.

Single-cell sequencing of final samples was performed using Single Cell 3′ kit V1 (Breast cancer sample), v2 (HNSCC Pt1) and v3 (HNSCC Pt2) (10X Genomics), according to the manufacturer protocol (10X Genomics), including the CITE-seq libraries (HNSCC samples). Sample quality was checked with Bioanalyzer Agilent 2100 using a HighSensitivity DNA chip (Agilent Genomics). Samples were sequenced on an Illumina HiSeq (Breast cancer sample) and Novaseq (HNSCC samples) with a depth of sequencing of 50,000 reads/cells (Breast cancer) and 100,000 reads/cells (HNSCC samples).

**ScRNAseq data processing and analysis**. FASTQ files were processed using the Cell Ranger pipeline (10X Genomics; V3 for HNSCC samples and V2 for Breast cancer sample), and reads were aligned to the human genome hg38. True cells are distinguished from empty droplets in the Cell Ranger pipeline; additionally, they were also excluded by Dropletutils R package version 1.6.1: the intersection of both was used. Cells with a high fraction of mitochondrial genes were excluded (>10% HNSCC: >20% Breast); potential doublets cells were excluded by filtering out cells with too high library complexity (cells that expressed more than 4000 genes (HNSCC) or 5000 genes (Breast): this threshold was based on the distribution of the number of cells per gene expression). For the HNSCC count matrix, the final number of cells is detailed in Supplementary Table 9. For the Breast cancer sample, the count matrix contained 333 cells, among which 20 pDC (cluster 4, Supplementary Fig. 14B) were excluded for the merged analysis. The filtered count matrices were then analyzed by Seurat R Package version 3.2.2. Data were normalized for library size and log2 transformed.

For the HNSCC, all samples (Supplementary Table 9) were merged: we selected canonical component analysis dimensions 1–50 for anchors selection for merging / integration. We selected the top PC based on the elbow plot representing the decay in explained variance per additional PC: HNSCC: 30PC; Breast: 75PC. We retained the reduced matrix for further downstream analysis. Graph-based clustering was performed with Seurat R package and represented by UMAP. The clustering resolution were 0.7 (HNSCC) and 0.8 (Breast). Differential gene expression analysis was performed using Seurat R package and based on Wilcoxon rank-sum test with and a minimal Log FC of 0.25. Clustering was based only on RNA data, whereas CITE-seq data was used for downstream analysis and cluster annotation. CITE-seq features were detected using CITE-seq Count version 1.4.3[78]. Cluster-specific gene expression and protein expression as detected by CITE-seq were used to annotate the cell clusters based on literature. For cell–cell interactions prediction, we used ICELLNET as described in ref. [34] with its ligand/receptor database now including 543 interactions, and enriched with a filtering of the genes expressed in less than 2% of the cells at the cluster level (https://github.com/soumelis-lab/ICELLNET). Those filtered genes were given a score of 0 for the corresponding cluster(s). Interactions including at least one gene expressed in less than 2% of the cells in all 24 clusters were removed from the tables, so that the final number of interactions in Supplementary Data 13 and 14 is 304. Analysis was performed on the matrix for which features expression values were normalized by library size for each cell.

**Analysis of ScRNAseq public data**. Datasets from four different publications were selected and are listed in Supplementary Data 15. cDC clusters were identified per author annotation or by performing de novo clustering with the technical parameters presented in Supplementary Data 15. Raw data from cDC of each dataset were merged in a meta-dataset, which also included the ScRNAseq data from the HNSCC samples of this study, and a tumor sample of a luminal breast cancer patient for which DC were sorted (see "Patients" and "Processing of samples for ScRNAseq" sections). We selected CCA dimensions 1–50 for anchors selection for merging / integration of the six datasets; 3000 features were selected as anchors

with a k-filter of 50. Eventually, we selected 50 PC and clustering resolution 1.2. Differential gene expression analysis was performed using Seurat R package and based on integrated data matrix and on Wilcoxon rank-sum test with and a minimal Log FC of 0.25. Pseudotime analysis was performed using Monocle 3 version 0.2.2[48] with cluster #2 of immature blood cDC2 defined as the origin. The clusters #9 of cycling DC, #3 of Lung Mono_DC, #5 of cDC1, and #10 of contaminating T cells were excluded from this analysis.

The mature migratory cDC signature presented in Supplementary Table 10 and used in Fig. 8E and Supplementary Fig. 18 is composed of the 29 genes in common between the signatures of cluster 20 in Fig. 6 and cluster 4 in Fig. 8, using the thresholds of log2 fold change > 1 and adjusted $p$-value < 0.05.

**Reporting summary**. Further information on research design is available in the Nature Research Reporting Summary linked to this article.

## Data availability

RNAseq and ScRNAseq data that support the findings of this study have been deposited in the NCBI under the identification numbers GSE169381 and GSE170673. Reads were aligned to the human genome hg19 (bulk RNAseq) and hg38 (single-cell RNAseq). Other datasets were used in this study: Mathan et al. (GSE89442); Cillo et al. (GSE139324); Zillionis et al. (GSE127465); He et al. (GSE147424); Arazi et al. (https://doi.org/10.1038/s41590-019-0398-x); Riaz et al. (https://github.com/riazn/bms038_analysis/tree/master/data); Gide et al. https://github.com/miabioinformatics/Gide_CancerCell2019); Grandclaudon et al. (DC–T database, source of Fig. 2, 3A, B; Supplementary Fig. 4A–D) is from Supplementary Table 2 in ref. [21]; Metabric data ref. [24]; TCGA (https://portal.gdc.cancer.gov). The source data file contains the raw data for Fig. 3C; Supplementary Fig. 5B; 6A, B. Source data are provided with this paper.

## Code availability

Code is available on demand by contacting the corresponding author. The count matrix and associated metadata of the merged cDC meta-dataset related to Fig. 8 is available in GSE170673.

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

## Acknowledgements

This work was supported by the Institut National de la Santé et de la Recherche Médicale under Grants BIO2012-02, BIO2014-08, and HTE2016; Agence Nationale de la Recherche under Grants ANR-10- IDEX-0001-02 PSL, ANR-11-LABX-0043 CIC IGR-Curie 1428, ANR-13-BSV1-0024-02 and ANR-16-CE15-0024-01; European Research Council under Grant IT-DC 281987; Institut National du Cancer under Grant EMERG-15-ICR-1, Cancéropole INCA PhD grant to C.H., and INCA PLBio Grants (INCA 2016-1-PL BIO-02-ICR-1; 2017-1-PLBIO-09-ICR-1) to V.S; Fondation ARC pour la Recherche sur le Cancer under Grants PJA 20131200436, ARCPGA12021020003101_3587, and DOC20160604230 to M.G.; Agence Nationale de Recherches sur le Sida et les hépatites virales to M.G.; Fondation pour la Recherche Médicale to M.G.; Ligue nationale contre le cancer (labellisation EL2016.LNCC/VaS) to V.S., and PhD grant to L.M.R.; and Institut Curie, in particular the PIC TME. We wish to thank for technical help: the INSERM U932, in particular Ares Rocañín-Arjó and Anne-Sophie Hamy-Petit; the Institut Curie Flow-Cytometry facility, in particular, Annick Viguier, Sophie Grondin, Lea Guyonnet, Coralie Guérin, the Institut Curie NGS platform, in particular Sonia Lameiras and Laura Baudrin; François Lemoine, Géraldine Lescaille, and Chloé Bertolus at the Hôpital Pitié Salpêtrière, Paris, France; Mickael Ménager and Marine Luka, Institute Imagine, Paris, France; Peter P Lee, City of Hope, CA, USA. We wish to thank for their feedback on the study: the INSERM U932, in particular Sebastian Amigorena and Nicolas Manel; Marc Dalod, CIML, Marseille, France; Marie-Caroline Dieu-Nosjean, Hôpital Pitié Salpêtrière, Paris, France; Pierre Saintigny and Christophe Caux, CLB, Lyon, France. This study makes use of data generated by: Mathan et al., Radboud Institute for Molecular Life Sciences, Nijmegen, Netherlands; TCGA and METABRIC consortia; Riaz et al., from the Memorial Sloan Kettering Cancer Center, New York, USA; Gide et al. from the Melanoma Institute Australia, Sydney, Australia, and we thank James Wilmott and Camelia Quek for providing us with the expression matrix in CPM; Cillo et al., University of Pittsburgh, PA, USA; Zilionis et al., Harvard Medical School, Boston, MA, USA; He et al., Icahn School of Medicine at Mt. Sinai, New York, NY, USA; Arazi et al., Broad Institute of MIT and Harvard, Cambridge, MA, USA.

## Author contributions

C.H. performed the flow cytometry and ScRNAseq experiments on human cancer samples, performed all analyses, identified the Secretory Helper dichotomy, and wrote the manuscript; F.N. performed all bioinformatical analyses and wrote the corresponding methods paragraphs; M.G. performed the in vitro assays with blood cDC; L.M.R. performed the ICELLNET analyses and the in vitro assays with purified cDC2; P.M. performed the RNAseq experiment; P.S. gave technical support and conceptual advice for the flow cytometry experiments and performed the ScRNAseq experiments with C.H.; L.F. performed the elastic net model and supervised the statistical analyses; A.S. and M.G.D. initiated in vitro assays and identified the PD-L1 and ICOSL mutual exclusion; O.L. designed the T cell flow cytometry experiment on human cancer samples; M.B. and S.B. sequenced the RNAseq and ScRNAseq data; J.Y. and W.G. provided the count matrix for the lung ScRNAseq dataset; J.R. provided samples and clinical data of patients from the hospital Pitié-Salpetrière, Paris; J.K. was referral pathologist for the human samples from the Institut Curie; C.L., M.K., and C.L.T. were in charge of the patients from the clinical trial SCANDARE NCT03017573; V.S. was the principal investigator of the study. All authors discussed the results and implications and commented on the manuscript at all stages, in particular C.H., F.N., O.L., M.G.D., and V.S.

## Competing interests

M.G. is currently employed by the company Generate Biomedicines. The remaining authors declare no competing interests.
