## [Peer Review File · Nature Communications]

PD-L1 and ICOSL discriminate human secretory and helper dendritic cells in cancer, allergy and autoimmunityEditorial Note: Parts of this Peer Review File have been redacted as indicated to remove third-party material where no permission to publish could be obtained.

REVIEWER COMMENTS

Reviewer #1 (Remarks to the Author):

In this manuscript the authors have claimed the discovery of functional dichotomy of human DC in different pathological conditions. They have identified "secretary DCs" and "helper DCs" that can be distinguished by the expression of PD-L1 and ICOSL. The formers are PD-L1^{hi}/ICOSL^{lo} and secrete large amount of cytokines including both inflammatory and suppressive cytokines. The latter are PD-L1^{lo}/ICOSL^{hi} that induce high level of the secretion of CD4 T cell cytokines. In order to identify the stimuli that are responsible for the induction of these 2 types of DCs, they have stimulated dendritic cells with a spectrum of stimuli and have identified that TLR-7 ligands could induce "secretary DCs" while GM-CSF and TSLP could induce "helper DCs". They have also address the issue of the interaction of the "secretary DCs" with other cells. The authors claim that the secretary DCs infiltrate in different pathological tissues and are even beneficial for cancer patients.

Despite these positives, however, there are several significant shortcomings in the experimental design and data interpretation that should be addressed before this manuscript could be considered for publication

Specific comments:

Antigen presenting cells are heterogenous and the roles of each subset of DCs have since long been debated. It has been shown in several studies that BATF3⁺ DC1 play important role in anti-tumor responses and the correlation between DC1 and favorable prognosis has been shown in different tumors, while the roles of the DC2 and other subsets are less clear. In this study the authors were attempting to touch this issue and call their "secretary DCs" DC2 (or DC2 derived), however, their experimental design and the results are quite confusing.

1. The authors should make it clear whether they are analyzing whole CD11c+HLA-DC⁺ cells or specifically focus on CD1c+CD14⁻ DC2, because they behave differently. The DCs dichotomy in this manuscript is established on cancer or blood CD11c+HLA-DR⁺ cells (Fig 1-3). The authors thereafter claimed that secretary DCs are DC2, based on the argument that "they are the major group of DCs", without providing detail analysis. This argument is problematic. Based on their gating strategy, it is very likely that CD14⁺ monocyte, monocyte-derived DC or macrophages are all included in the analysis. Therefore the secretary DC could come from multi sources. It can also be seen in their HNSCC DCs that the majority of CD11c+HLA-DR⁺ cells are MMAC (>60%, defined by the authors, and similar results can be seen in their previous publications) and in Fig1D that PD-L1⁺ DC1e and PD-L1⁺ MMAC but not PD-L1⁺ DC2 are dominant in CD3^{hi} tumor. As the consequence, the scRNAseq analysis (Fig 6B) shows that substantial proportion of MMAC express PD-L1 (another major source of PD-L1 in tumor), and the gene expression pattern of these PD-L1⁺ mmDC are quite different from DC2 (Fig S11), rather closer to DC1. Therefore it is confusing why the authors sorted only blood and tumor DC2 for their "secretary DCs signature" (Fig4). The authors should either include other subsets in their "secretary DC signature" or prove there is no difference between blood CD11c+HLA-DR⁺ cells from DC2 (what exactly the cells were used in Fig2-3 and how many of them are CD1c+CD14⁻ DC2?), or provide more evidences to prove that "secretary DCs" are indeed only derived from DC2.

2. The authors have demonstrated the presence of "secretary DCs" in different pathological tissues. However, the fundamental question remain to answer: what are the roles of "secretary DCs" and "helper DCs"? and are "secretary DCs" superior than "helper DCs" in terms of anti-tumor responses, for example? The authors have demonstrated in Fig5 that the infiltration of secretary DCs has a beneficial effect. One could therefore ask whether this is because they attract more CD8 or CD4 T cells or other immune cells to infiltrate to tumors? and is there OS or PFS difference between CD3^{hi} and CD3^{lo} patients? and how about patients with more "helper DCs" (in Fig1 for example)? Do they have worse prognosis?

3. The authors have demonstrate the difference between "secretary DCs" and "helper DCs" in term of CD4 activation. However, CD8 T cells might be more critical than CD4 T cells for anti-tumor responses. Is there any difference between these 2 types of DCs in term of CD8 T cells activation?

4. It is curious how could "secretary DC" that express suppressive molecule PD-L1 and do not induce activation of CD4 T cells (few cytokine secretion), do not induce more T cell proliferation than "helper DC", could be beneficial.

Minor comments:

1. Please verify carefully the manuscript. There are several typos at line 203, line 479, legend of Fig 1E (left vs right) and Fig 8 (C vs D).

2. The abbreviation should be mentioned when using for the first time: HKLM, HKSA, cGAS...etc.

Reviewer #2 (Remarks to the Author):

In their manuscript, Hoffmann et al define two different subsets of activated dendritic cells (DC) according to the expression of different markers identified initially in tumor samples cancer patients. By using a combination of in vivo and in vitro approaches and flow cytometry, protein secretion and bulk and single cell-RNAseq techniques, they confirm the data not only in several groups of cancer patients, but also in other inflammatory settings. According to this, they define secretory DC (PD-L1^{high}, ICOSL^{low}) and helper DC (PD-L1^{low}, ICOSL^{high}). Secretory DC with a mature migratory profile are found in cancer (where they are associated with a better prognosis) and in inflammatory diseases like atopic dermatitis and lupus. Analysis of the main features of these cells, together with the role they may play in inflammatory conditions, the characterization of suitable markers for their identification and its prognostic value, led authors to propose important implications of these data in immunotherapy.

The identification of DC features that characterize them as immunogenic or tolerogenic is an important goal to understand the biology of these cells and to apply this knowledge for immunotherapeutic purposes. In this regard, their data are of relevance, since they define precise markers to identify DC subsets with different functional properties, which are associated with several inflammatory conditions.

There are several comments regarding the manuscript:

- A general comment relates to the particular DC subset used for most of the analysis. At the beginning of the manuscript, they refer to DC (in general), but later, they jump to cDC2. When authors stimulate blood DC in vitro and they analyze the resulting PD-L1/ICOSL phenotype they present a subset of "PD-L1^{high} and ICOSL^{low/neg}," indicating that these are "like ex vivo cDC2 from inflamed tumors". Although true, in this sense, they are like most DC subsets found on inflamed tumors, not only cDC2, which specially for cDC2 PD-L1^{high}, they are not predominant in the tumor. In the same line, when testing DC in vitro, they use blood DC, where they indicate a higher proportion cDC2 cells. However, according to Figure 1B, in the tumor cDC2 account for ~8% of CD11c+HLA-DR+ DC, whereas cDC1e correspond to 21%.

- In this regard, there are some issues that would need some clarification, such as they precise method of DC purification. If the DC purification protocol has been based on lin- CD11c+HLA-DR+ DC, results are more associated with cDC1e in the tumor. Moreover, these cells are not the predominant ones, but MMC1 cells. On the other side, in Methods, authors refer a previous paper (ref 21), where the purification protocol relies on Lin-, CD11c+, CD4+, presumably related to cDC2. These discrepancies need to be clarified, since depending on the purification method, the DC subset analyzed and the corresponding results may be biased. Finally, poly(I:C) is a ligand for cDC1 cells. How do they explain their effects on their DC, if they have purified cDC2? Are there contaminating cells responsible for what they define as a "unique profile"?

- For figure 1, they indicate "We extracted a total of 434 parameters" "To avoid bias in the subsequent analyses, we used a sub-list of 81 non-redundant parameters among the 434 measured" Please explain better this point in order to understand how this selections was carried out. Moreover, these parameters are checked in 3 subgroups of patients with low, medium and high T cell infiltration. It would be also helpful to indicate how this division has been carried out.

- In Figure 4, authors compare genes upregulated in tumor DC vs those upregulated in blood DC. In order to ascribe a secretory vs helper signature to those DC, they next define these signatures based on differences between DC stimulated with secretory- and helper-inducing stimuli, and finally they compare all lists. Although they show differences between tumor and blood DC, features of blood DC in a cancer patient may mask differences that could be observed when using blood DC from non-cancer individuals. This underestimation should be considered when

interpreting data.

- Moreover, when comparing tumor and blood DC they use DC from highly infiltrated tumors (due to logical methodological reasons). However, this may bias the type of DC found in their experiments, mainly considering initial data shown in figure 1, where infiltrated tumors have more secretory DC.

- Supplementary Figure 7B-D. Which are the comparisons carried out to extract the DEGs in Fig S7D, every DC subset against the others?

- Authors show that secretory DC are associated with a favorable microenvironment for antitumor immunity, as well as with response to immunotherapy based on immune checkpoint blockers. Considering that secretory DC are predominant in tumors with high CD3 infiltrate, to what extent can be these parameters (secretory DC and CD3 infiltrate) considered as independent factors or are they two inter-related factors?

- Experiments of scRNAseq show that, as oppose to bulk RNAseq, there are different profiles of cytokine secretion between mmDC, cDC1, cDC2 and MMAC. Therefore, profiles of cytokine secretion by DC shown in Figure 3 may represent the overall secretion by all DC groups. Authors should clarify this issue when interpreting their data, since "secretory" phenotype may encompass different secretion profiles leading to different outcomes. Indeed, CCL19 and IL27 are now shown to be produced by cDC2, IL12 by mmDC and IL10, TNF and IL1B by MMAC.

- cDC2 and Tregs in tumors have been correlated with the presence of differentiated CD4 T cells and with CD8 T cell responses (their ref#6). These results were obtained using the global cDC2 subset. Since according to the present manuscript mmcDC seem as those precise cells with the highest interaction capacity, would authors see a better association with Tregs and CD8 T cells when considering only mmcDC?

- In the Discussion, authors mention about the potential effect of "other stimuli, dose, duration, etc". Results from those perturbators that have been tested at different doses (e.g. PAM3, HKSA or HKLM) show that the chosen doses dictates important features. In some cases, increase/decrease in the dose may simply modify the "potency" of the stimulus (HKSA) without changing the profile of DC generated. In other cases (PAM3, HKLM) changes in the doses shifts the profile from a purely helper DC or purely secretory to a mixed one. Therefore, in addition to the corresponding signaling pathway, the dose should be considered when classifying these compounds, mainly for translational purposes.

- Finally, a secretory DC profile is associated with a T cell inflamed TME and with good prognosis in two of their cancer types. However, this DC subset does not promote cytokine secretion by CD4 cells in vitro, a property of CD4 T cells that would suggest a higher chance of activated T cells. Although in vivo vs in vitro differences may account for these discrepancies, there seem to be other factors involved which are not considered in this model? Would this classification apply to cDC1 cells (which, as they show, predominate in the tumor) and are responsible for activating CD8 T cells? Would it similarly affect to effector functions displayed by these CD8 T cells, with a proven role in antitumor immunity?

Minor comments

- Reference 21 should be updated in the reference list section

- Line 203: "IL-10, IL-13, IL-31 and GM-CSF as compared to both PD-L1low ICOSLhigh DC" Should this be PD-L1high ICOSL low DC?

- References shown in the Y-axis in figure 5C are 25 and 26 whereas in the main text (Results and Reference sections) correspond to 27 and 28.

Reviewer #3 (Remarks to the Author):

In this paper, the authors seek to correlate T-cell infiltration with dendritic cell phenotype and function in human head and neck squamous cell carcinoma (HNSCC) samples and also other diseased tissues. They conclude that PD-L1 expression on DCs positively correlates with CD3+ T cell infiltration, whereas ICOS expression correlates negatively. The myeloid cells were initially categorized as monocytes and macrophages (MMAC), CD14+ DC, cDC2 and cDC1e based on flow cytometry staining. Next, the classification of DCs by PD-L1 and ICOS expression was extended into blood DCs and a database generated previously, and the analysis led the authors to conclude that PD-L1 and ICOS expression are mutually exclusive. By coculturing the 5 blood DC populations matured in vitro and analyzing the cytokines secreted, the authors conclude that PD-L1 and ICOS define "secretory" and "helper" DCs, respectively. In separate studies, based on RNA-seq, they conclude that T- inflamed HNSCCs are enriched with secretory DC2s, and further that tumor secretory DC2s are associated with a better prognosis and response to immune checkpoint antibody therapy. Tumor secretory DC2s are characterized by mature migratory and cell-cell interaction associated genes. The secretory DC signature also exists in other chronic inflammatory and autoimmune disease sites.

Critique:

- 1) In Figure 1, the gating of DCs appears incorrect. DCs in humans are generally gated as CD45+ Lin neg (CD3-CD19-, CD20-, CD335-) and HLA-DR+CD14-CD16-. Among these cells, pDCs are BDCA2+ CD123+, while cDC1s are CD141+ and cDC2s are CD1c+. Unfortunately, none of the DC subsets shown in figure 1B appear to be correctly gated. Problems related to the gating of DCs and T cells are present throughout the paper, which make it difficult to interpret the data.
- 2) Unrelated to the gating issues, in Fig. 1c it is unclear why the authors decided to divide the tumor samples obtained from the 24 patients into groups of "equivalent size" based on relative T cell frequency. In the first place, a visual examination of the dot plot suggests that rather than 3 "clusters", there are only 2. However, there is no a priori reason to divide the samples into arbitrary groups when there are standard unbiased statistical methods that can be used to determine if there is a correlation between relative T cell frequency and other parameters. If the authors' objective is to correlate CD3+ T cell infiltration with DC subsets, they should consider performing RNA-seq on an adequate cell sample and classify DC subsets using an unbiased algorithm.
- 3) Different DCs preferentially stimulate CD4+ and CD8+ T cells, respectively, so it is difficult to understand the rationale for using percentages of total CD3+ T cells to correlate with DC subsets and function.
- 4) The authors do not explain why PD-L1 and ICOS expression are expressed in a mutually exclusive manner on the secretory and helper DCs, respectively. If this is truly the case, then shouldn't these molecules contribute to the different functions of the DCs? The details underlying Figures 1E and 1F are not provided. The data are based on analysis of DCs in tumors, but no studies are performed to directly investigate whether or how either DC population affects anti-tumor immunity?
- 5) Figure S3, since all the four DC subsets express CD141, it is difficult to understand the intent of the data.
- 6) The rationale for performing the extensive experiments described in Figure 3 on blood DCs using diverse stimuli, in vitro, is unclear. By definition, all of these conditions are distinct from those in HNSCC. Importantly, also, the studies described in reference 21 used moDCs, which are distinct from blood CD11c+ DCs.
- 7) The raw data for Figure S5 C and D are not provided, so it is not possible to match the data with the figures.
- 8) The rationale for the studies of DC2s described in Figures 4 and 5 is not well explained, and therefore difficult to connect to the initial and main focus of the report.
- 9) In the discussion, the authors describe the "Secretory DC signature as the combination of maturation markers (CD40, CD70, CD80, LAMP3), migratory marker CCR7, chemokines (CCL19, CXCL9), cytokines (IL12B, IL23A, EBI3), the receptor LY75 (DEC-205), checkpoints CD274 (PD-L1), PDCD1LG2 (PD-L2), PVR, IDO, CD200, TNFRSF9 (4-1BB), and the transcription factors ETV3/ETV3L and REL/RELB." Since DEC205 is a DC1 marker, while IDO, CD200, PD-L1 are generally expressed by regulatory DCs, the authors' definition is at odds with current understanding of DC biology.

REVIEWER COMMENTS

Reviewer #1 (Remarks to the Author):

In this manuscript the authors have claimed the discovery of functional dichotomy of human DC in different pathological conditions. They have identified “secretary DCs” and “helper DCs” that can be distinguished by the expression of PD-L1 and ICOSL. The formers are PD-L1hi/ICOSLlo and secrete large amount of cytokines including both inflammatory and suppressive cytokines. The latter are PD-L1lo/ICOSLhi that induce high level of the secretion of CD4 T cell cytokines. In order to identify the stimuli that are responsible for the induction of these 2 types of DCs, they have stimulated dendritic cells with a spectrum of stimuli and have identified that TLR-7 ligands could induce “secretary DCs” while GM-CSF and TSLP could induce “helper DCs”. They have also address the issue of the interaction of the “secretary DCs” with other cells. The authors claim that the secretary DCs infiltrate in different pathological tissues and are even beneficial for cancer patients.

Despite these positives, however, there are several significant shortcomings in the experimental design and data interpretation that should be addressed before this manuscript could be considered for publication.

Specific comments:

Antigen presenting cells are heterogenous and the roles of each subset of DCs have since long been debated. It has been shown in several studies that BATF3+ DC1 play important role in anti-tumor responses and the correlation between DC1 and favorable prognosis has been shown in different tumors, while the roles of the DC2 and other subsets are less clear. In this study the authors were attempting to touch this issue and call their “secretary DCs” DC2 (or DC2 derived), however, their experimental design and the results are quite confusing.

R1.1. The authors should make it clear whether they are analyzing whole CD11c+HLA-DR+ cells or specifically focus on CD11c+CD14- DC2, because they behave differently. The DCs dichotomy in this manuscript is established on cancer or blood CD11c+HLA-DR+ cells (Fig 1-3). The authors thereafter claimed that secretary DCs are DC2, based on the argument that “they are the major group of DCs”, without providing detail analysis. This argument is problematic. Based on their gating strategy, it is very likely that CD14+ monocyte, monocyte-derived DC or macrophages are all included in the analysis. Therefore the secretary DC could come from multi sources. It can also be seen in their HNSCC DCs that the majority of CD11c+HLA-DR+ cells are MMAC (>60%, defined by the authors, and similar results can be seen in their previous publications) and in Fig1D that PD-L1+ DC1e and PD-L1+ MMAC but not PD-L1+ DC2 are dominant in CD3hi tumor. As the consequence, the scRNAseq analysis (Fig 6B) shows that a substantial proportion of MMAC express PD-L1 (another major source of PD-L1 in tumor), and the gene expression pattern of these PD-L1+ mmDC are quite different from DC2 (Fig S11), rather closer to DC1. Therefore it is confusing why the authors sorted only blood and tumor DC2 for their “secretary DCs signature” (Fig4). The authors should either include other subsets in their “secretary DC signature” or prove there is no difference between blood CD11c+HLA-DR+ cells from DC2 (what exactly the cells were used

in Fig2-3 and how many of them are CD1c+CD14- DC2?), or provide more evidences to prove that “secretary DCs” are indeed only derived from DC2.

Response:

R1.1a. Clarification on DC subsets and validation of the Secretary/Helper dichotomy on pure cDC2

This concern was also raised by Reviewers 2 & 3 and is partly redundant with R2.1 and R3.1.

We acknowledge that the CD11c⁺HLA-DR⁺ populations presented in Fig 1 versus Fig 2 and Fig 3 did not refer to the same cell populations, which was confusing. The label “cDC1_e” for “cDC1 enriched” population was also a confusing term.

We have proceeded to changes in cell population labels and names as detailed below.

In HNSCC (Fig 1), CD11c⁺HLA-DR⁺ cells contain DC subsets and MMAC and this name was kept. The “cDC1_e” population that contained CD11c⁺HLA-DR⁺CD14⁻CD1c⁻ was renamed “double negative DC and MMAC” and labelled “DN DC/MMAC”. This population contains a minority of cDC1 (median < 5%) identified as CD141^{high} as shown in Fig S2B (6 independent HNSCC samples) and other cells that most likely are non-classical CD14⁻ macrophages.

In Blood (Fig 2; Fig3), CD11c⁺HLA-DR⁺ cells referred to primary blood cDC1 and cDC2. We have removed this term and now use “blood cDC”. These cells were gated as CD3⁻, CD14⁻, CD16⁻, CD19⁻, HLA-DR⁺, CD4⁺, CD11c⁺, as explained in the resource paper Grandclaoudon et al., Cell, 2019, and the methods paper Alculumbre and Pattarini, Methos Mol, Biol, 2016. These blood cDC are composed of a majority CD1c⁺CD141⁻ cDC2 (around 80%) and a minority of CD1c-CD141+ cDC1 (less than 10%) and other CD11⁺CD4⁺CD1c⁻CD141⁻ cells as shown in Figure 2f of Alculumbre and Pattarini, 2016, that is reproduced at the end of this document (Appendix 1). This information was added clearly in the methods section.

Besides name changes, we have validated the concept of Secretary and Helper DC with “pure” blood cDC2 by performing supplementary experiments. Pure cDC2 were gated as CD3⁻CD14⁻CD16⁻CD19⁻CD20⁻CD56⁻CD123⁻CD11c⁺CD1c⁺ (new Fig S5A). These results are now presented in new supplementary Figure S5B.

R1.1b. PD-L1 expression on DC subsets and MMAC

The expression of PD-L1 and ICOSL was analyzed independently on each of the four cell populations: MMAC, DN DC/MMAC, cDC2 and CD14⁺DC. Previously, we had separated the 22 samples per tertile (high, int, low) of CD3 infiltration, and performed an ANOVA. The increased proportion of PD-L1-expressing cDC2 did not appear in the initial Fig 1D because this parameter did not reach statistical significance. However, the expression of PD-L1 within the four cell populations was highly correlated, and PD-L1 cDC2 were enriched in CD3 high tumors, as previously shown in supplementary Fig S3C (this figure S3C has now been deleted).

Because of other reviewers 2 & 3 comments, we have modified the analysis performed in Fig 1. We have performed a Spearman correlation between CD3 and all other non-redundant parameters to eliminate the arbitrary division per tertile. This is shown in the new Fig 1D, in which the correlation of PD-L1 expression in all DC and MMAC subsets and its association with CD3 infiltration level is more clearly shown.

We agree that an important fraction of PD-L1⁺ cells in the tumor microenvironment are macrophages. However, the presence of PD-L1⁺ macrophages does not change the fact that PD-L1⁺ mmDC more closely resemble *in vitro*-defined Secretory cDC than Helper cDC. We agree that the gene expression pattern of cDC2 (cluster 7) is different from PD-L1⁺ mmDC (cluster 20) and from cDC1 (cluster 23), hence the three different clusters well separated visible in the UMAP in Fig 6B. As mentioned in the results section “ADT CD1c confirmed that these cells are at least in part cDC2 (Fig 6B); cDC1 may also contribute to this population of mmDC as previously shown.”. Therefore, our data show that cDC2 in tumors mature with a Secretory phenotype to become mmDC. Because functional assays with blood DC contained only a minority of cDC1 and because RNAseq analysis of sorted tumor infiltrating cDC1 could not be performed due to the very low cell number limitation, we do not claim (nor exclude) that the Secretory versus Helper dichotomy also applies to cDC1. We show at the single-cell level the signature and gene expression of PD-L1⁺mmDC, that is composed for sure of matured cDC2 (as shown by the ADT CD1c), and most likely also by matured cDC1 (literature data, Maier et al. (10)), that converge at the transcriptomic level. We have now better addressed this point in the discussion.

R1.2. The authors have demonstrated the presence of “secretory DCs” in different pathological tissues. However, the fundamental question remains to answer: what are the roles of “secretory DCs” and “helper DCs”?

and are “secretory DCs” superior to “helper DCs” in terms of anti-tumor responses, for example?

The authors have demonstrated in Fig5 that the infiltration of secretory DCs has a beneficial effect. One could therefore ask whether this is because they attract more CD8 or CD4 T cells or other immune cells to infiltrate to tumors?

and is there OS or PFS difference between CD3hi and CD3lo patients?

and how about patients with more “helper DCs” (in Fig1 for example)? Do they have worse prognosis?

Response:

R1.2a. Roles of secretory and helper cDC in tissues.

In Fig 1, DC infiltrating CD3 low tumors presented with an intermediate expression of ICOSL that was close to the one observed in blood cDC and stronger than the one observed on cDC in CD3 high tumors. However, that level of ICOSL expression remained clearly lower than the very strong upregulation of ICOSL observed *in vitro* on Helper DC. Therefore, cDC in CD3 low tumors (Fig 1) as well as blood cDC (Fig 1 and Fig 4) were not matching the *in vitro* Helper phenotype but were rather corresponding to medium-DC in Fig 2.

Fig 8 presents cDC from the various anatomical locations and contexts, confirming that Helper DC were not identified *in vivo*, and that in all datasets mature migratory DC converged towards the Secretory signature, regardless of the context, even rich in TSLP such as Atopic Dermatitis.

Altogether, *in vitro* Secretory cDC corresponded to mature migratory cDC in tissues in various contexts, and we did not identify a tissue counterpart of the *in vitro* Helper cDC. The question of the existence or the inducibility (e.g. by high doses of Helper inducers) of Helper

cDC *in vivo* remains to be addressed. To increase the clarity of this point we have modified the text accordingly in the Results and Discussion sections.

R1.2b. Prognostic / predictive impact of secretory DC and relationship to CD3 infiltration.

This concern was also raised by Reviewers 2 & 3 and our answer below is partly redundant with R2.6 and R3.8.

The beneficial effect of a higher infiltration with Secretory DC as shown in Fig 5 can indeed be due to their ability to efficiently attract CD8 and CD4 T cells, which is in line with the positive association found between PD-L1⁺ DC and CD3 High tumors shown in Fig 1 and their increased expression of T cell-attracting chemokines as shown in Fig 3 (*in vitro*), Fig 4 (*ex vivo* HNSCC RNAseq), Fig 7C and Fig S13A (*ex vivo* HNSCC ScRNAseq), Fig S15A (*ex vivo* various contexts ScRNAseq).

We confirmed the strong relationship between Secretory DC and T cell inflammation by performing correlations of the metagenes of each cell population now displayed in additional graphs in Fig 5 and new Fig S9. Pearson correlations were higher than 0.6 in all the 5 datasets (HNSCC, triple negative breast cancer (TNBC), luminal breast cancer (LumBC) and the 2 datasets studying the response to immunotherapy).

Moreover, the Secretory DC signature had a positive prognostic impact in HNSCC and TNBC, an absence of prognostic impact in LumBC, and a positive predictive impact for response to immunotherapy. We performed the same analysis with a T cell metagene and obtained similar results. Altogether, the data support the existence of a multi-cellular inflamed immune archetype enriched, among other, in T cells and Secretory cDC. This archetype ensures this partially efficient spontaneous anti-tumor immune response that provides patients with improved outcomes.

R1.3. The authors have demonstrated the difference between “secretory DCs” and “helper DCs” in term of CD4 activation. However, CD8 T cells might be more critical than CD4 T cells for anti-tumor responses. Is there any difference between these 2 types of DCs in term of CD8 T cells activation?

Response R1.3:

This concern was also raised by Reviewers 2 & 3 and the answer below is redundant with R2.10 and R3.3.

To address this comment, we have performed new experiments with cDC2 co-cultured with naïve CD8 T cells. cDC2 were activated by the Secretory inducer R848 or by the Helper inducer TSLP. As with CD4 T cells, Helper TSLP-cDC2 and not Secretory R848-cDC2 induced high levels of CD8 T cell cytokine secretion. These results are displayed in new Fig S6B.

R1.4. It is curious how could “secretory DC” that express suppressive molecule PD-L1 and do not induce activation of CD4 T cells (few cytokine secretion), do not induce more T cell proliferation than “helper DC”, could be beneficial.

Response R1.4:

In HNSCC, mmDC that align with the *in vitro* Secretory DC are associated to good prognosis and response to immunotherapy as shown in Fig 5. These cells are part of a spontaneous multicellular anti-tumor response as explained in response R1.2b above (Fig 1, new graphs in Fig 5, and new Fig S9) that altogether provides the advantage over the patients with non-inflamed tumors. Despite this advantage, this immune response is still insufficient to achieve tumor clearance without additional treatment. The transcriptome of mmDC and the functional assays on Secretory DC show that these cells seem efficient for T cell attraction, but indeed express several inhibitory molecules *ex vivo* and have a limited effect on CD4 T cell activation *in vitro*.

Therefore, in non-inflamed TME, it may be interesting to use the compounds identified as Secretory inducers to try to induce the infiltration by mmDC and initiate this immune response. Then - or as a first step in inflamed tumors already infiltrated by mmDC - we may want to block the inhibitory molecules and/or activate the stimulatory molecules of mmDC described here to achieve better outcomes. This point is commented in the discussion in our manuscript.

Among the inhibitory molecules expressed by Secretory cDC, we have tested the effect of a combined blocking of PD-1, TIGIT, IL-10R/IL10, and IL-4R during CD4 T cell – Secretory cDC2 co-culture. We observed that this multiple blocking increased the CD4 cytokine production of IL-3, IL-5, IL-13, and IFN γ (new Fig 3C, new Fig S6A) but not of the other cytokines measured (IL-2, IL-4, IL-6, IL-9, IL-10, GM-CSF, TNF α). For the cytokines that were increased, the levels remained low as compared to the Helper TSLP-cDC2 condition. No individual blocking was able to induce the increased CD4 cytokine production observed with the multiple blocking. Altogether, the combined expression of PD-L1, PD-L2, PVR, and secretion of IL-10 and IL-4 by Secretory cDC2 played a significant role in the inhibition of CD4 Th cytokine production.

Minor comments:

R1.m.1. Please verify carefully the manuscript. There are several typos at line 203, line 479, legend of Fig 1E (left vs right) and Fig 8 (C vs D).

We have corrected these errors, thank you. However, the error line 479 was not found.

R1.m.2. The abbreviation should be mentioned when using for the first time: HKLM, HKSA, cGAS...etc.

We have added the corresponding full names, thank you.

Reviewer #2 (Remarks to the Author):

In their manuscript, Hoffmann et al define two different subsets of activated dendritic cells (DC) according to the expression of different markers identified initially in tumor samples cancer patients. By using a combination of in vivo and in vitro approaches and flow cytometry, protein secretion and bulk and single cell-RNAseq techniques, they confirm the data not only in several groups of cancer patients, but also in other inflammatory settings. According to this, they define secretory DC (PD-L1high, ICOSLlow) and helper DC (PD-L1low, ICOSLhigh). Secretory DC with a mature migratory profile are found in cancer (where they are associated with a better prognosis) and in inflammatory diseases like atopic dermatitis and lupus. Analysis of the main features of these cells, together with the role they may play in inflammatory conditions, the characterization of suitable markers for their identification and its prognostic value, led authors to propose important implications of these data in immunotherapy. The identification of DC features that characterize them as immunogenic or tolerogenic is an important goal to understand the biology of these cells and to apply this knowledge for immunotherapeutic purposes. In this regard, their data are of relevance, since they define precise markers to identify DC subsets with different functional properties, which are associated with several inflammatory conditions. There are several comments regarding the manuscript:

- A general comment relates to the particular DC subset used for most of the analysis. At the beginning of the manuscript, they refer to DC (in general), but later, they jump to cDC2. When authors stimulate blood DC in vitro and they analyze the resulting PD-L1/ICOSL phenotype they present a subset of "PD-L1high and ICOSLlow/neg, " indicating that these are "like ex vivo cDC2 from inflamed tumors". Although true, in this sense, they are like most DC subsets found on inflamed tumors, not only cDC2, which specially for cDC2 PDL1high, they are not predominant in the tumor. In the same line, when testing DC in vitro, they use blood DC, where they indicate a higher proportion cDC2 cells. However, according to Figure 1B, in the tumor cDC2 account for ~8% of CD11c+HLA-DR+ DC, whereas cDC1e correspond to 21%.

R2.1. In this regard, there are some issues that would need some clarification, such as they precise method of DC purification. If the DC purification protocol has been based on lin-CD11c+HLA-DR+ DC, results are more associated with cDC1e in the tumor. Moreover, these cells are not the predominant ones, but MMC1 cells. On the other side, in Methods, authors refer a previous paper (ref 21), where the purification protocol relies on Lin-, CDC11c+, CD4+, presumably related to cDC2. These discrepancies need to be clarified, since depending on the purification method, the DC subset analyzed, and the corresponding results may be biased. Finally, poly(I:C) is a ligand for cDC1 cells. How do they explain their effects on their DC, if they have purified cDC2? Are there contaminating cells responsible for what they define as a "unique profile"?

Response R2.1:

This concern was also raised by Reviewers 1 & 3, and our answer below is partly redundant with R1.1a.

We acknowledge that the CD11c⁺HLA-DR⁺ populations presented in Fig 1 versus Fig 2 and Fig 3 did not refer to the same cell populations, which was confusing. The label “cDC1_e” for “cDC1 enriched” population was also a confusing term.

We have proceeded to changes in cell population labels and names as detailed below.

In HNSCC (Fig 1), CD11c⁺HLA-DR⁺ cells contain DC subsets and MMAC and this name was kept. The “cDC1_e” population that contained CD11c⁺HLA-DR⁺CD14⁻CD1c⁻ was renamed “double negative DC and MMAC” and labelled “DN DC/MMAC”. This population contains a minority of cDC1 (median < 5%) identified as CD141 high as shown in Fig S2B (6 independent HNSCC samples) and other cells that most likely are non-classical CD14⁻ macrophages. Because cDC1 were only a minor fraction of this heterogeneous population in the tumor, we could not perform bulk RNAseq on that subset. The cDC subsets proportions in the tumor are also shown in Fig 6 after single-cell RNAseq: PD-L1^{neg/low} cDC1 formed the smallest subset (cluster #23; 55 cells), PD-L1^{neg/low} cDC2 were most abundant (cluster #7; 473 cells) and finally PD-L1^{high} cDC also named here mmDC formed a small population (cluster #20; 111 cells) but more abundant than PD-L1^{neg/low} cDC1. ADT CD1c data in this study and literature data (Maier et al.) suggest that these PD-L1⁺ mmDC are composed of both cDC1 and cDC2 that have a convergent transcriptomic program upon tumor-induced maturation. Information on this point has been added in the discussion.

In Blood (Fig 2; Fig 3), CD11c⁺HLA-DR⁺ cells referred to primary blood cDC1 and cDC2. We have removed this term and now use “blood cDC”. These cells were gated as CD3⁻, CD14⁻, CD16⁻, CD19⁻, HLA-DR⁺, CD4⁺, CD11c⁺, as explained in the resource paper Grandclaudon et al., Cell, 2019, and the methods paper Alculumbre and Pattarini, Methods Mol Biol, 2016. These blood cDC are composed of a majority CD1c⁺CD141⁻ cDC2 (around 80%) and a minority of CD1c⁻CD141⁺ cDC1 (less than 10%) and other CD11⁺CD4⁺CD1c⁻CD141⁻ cells as shown in Figure 2f of Alculumbre and Pattarini, 2016, that is reproduced at the end of this document (Appendix 1). This information was added clearly in the methods section. We therefore conclude that the dichotomy of Secretory versus Helper DC is valid on cDC2. We acknowledge that the presence of a low number of cDC1 may have partially influenced the data, as it is visible for the samples stimulated by the TLR3 agonist Poly I:C that have a specific cytokine secretion profile, as mentioned in the results section. Therefore, we have validated the concept of Secretory and Helper DC with “pure” blood cDC2 by performing supplementary experiments. Pure cDC2 were gated as CD3⁻CD14⁻CD16⁻CD19⁻CD20⁻CD56⁻CD123⁻CD11c⁺CD1c⁺ (new Fig S5A). These results are now presented in new supplementary Fig S5B.

R2.2 For figure 1, they indicate "We extracted a total of 434 parameters" "To avoid bias in the subsequent analyses, we used a sub-list of 81 non-redundant parameters among the 434 measured" Please explain better this point in order to understand how this selection was carried out. Moreover, these parameters are checked in 3 subgroups of patients with low, medium and high T cell infiltration. It would be also helpful to indicate how this division has been carried out.

Response

R2.2a Clarification on "We extracted a total of 434 parameters":

Regarding the parameters excluded from cytometry analysis in Fig 1, we simply excluded all redundant data, meaning any parameter that was including partly the information contained in another one. With flow cytometry data obtained with a gating strategy, non-redundant parameters are the expression of each population in its direct parental gate. To the contrary, the redundant parameters were made of the same populations but expressed in their grand-parental, grand-grand-parental, etc..., gates. For example, if the gating is Live cells → CD3 T cells → CD8 T cells, then non-redundant parameters are the percentages of CD3⁺ among live cells and the percentages of CD8⁺ T cells among CD3, whereas the percentage of CD8 T cells among live cells is a redundant parameter that contains the information of both previously mentioned non-redundant parameters. We performed multicolor flow cytometry with the gating strategies including several subsequent gates as shown in Fig 1B, S1 and S2A, and recorded all parameters in each subsequent gate, which lead to a high proportion of redundant parameters (353 out of 434 including 52 cell ratios). Similarly, all 52 cell ratios measured here were considered redundant parameters too and excluded from the analysis in Fig 1. We have modified the sentence referring to this process in the Methods - "Analysis of Flow cytometry data" section to ensure clarity.

R2.2b Changes regarding the 3 subgroups of patients in Fig 1:

Regarding the 3 subgroups of patients with low, medium, and high T cell infiltration, we simply divided the 22 patients in groups per tertile, as mentioned in the results section. A cut-off at median did not seem appropriate when observing the distribution of CD3/Live in Fig 1C.

To address your and other reviewers' comment on this point, we have changed our strategy in Fig 1 and performed a Spearman correlation analysis between CD3 T cell infiltration and all other 81 non-redundant parameters, now shown in the new Fig 1D. We also provide the readers with the full correlation matrix in the new Table 2. However, this approach was not compatible with the analysis of the other redundant parameters, so we kept the division per tertile for the elastic net model shown in Fig S3B. This point also addresses a concern by reviewer 1 and 3 (see our reply R1.1b, R3.2).

R2.3. In Figure 4, authors compare genes upregulated in tumor DC vs those upregulated in blood DC. In order to ascribe a secretory vs helper signature to those DC, they next define these signatures based on differences between DC stimulated with secretory- and helper-inducing stimuli, and finally they compare all lists. Although they show differences between tumor and blood DC, features of blood DC in a cancer patient may mask differences that could be observed when using blood DC from non-cancer individuals. This underestimation should be considered when interpreting data.

Response R2.3:

Our goal was to determine the changes occurring after cDC2 maturation in the tumor tissue. To minimize inter-patient variability, we have favored the pairing of blood and tumor samples, when possible, hence the use of cDC2 from cancer patients. We acknowledge that blood cDC2 in cancer patient may present with some level of pre-existing activation. However, in Fig 8, we compared at the single-cell transcriptomic level the healthy donor cDC "HD_PBMC" with cancer patient cDC "HNSCC_PBMC_HPVneg"; "HNSCC_PBMC_HPVpos";

“NSCLC_PBMC”, and we did not observe any important differences between cDC2 subsets within these PBMC. Most cells were found in the same clusters #0 and #2 regardless of the presence or absence of cancer. We only observed that 1% of the cells within “HNSCC_PBMC_HPVneg” were mature migratory DC with a Secretory phenotype, suggesting only a minor effect of the presence of cancer on circulating cDC2.

R2.4. Moreover, when comparing tumor and blood DC they use DC from highly infiltrated tumors (due to logical methodological reasons). However, this may bias the type of DC found in their experiments, mainly considering initial data shown in figure 1, where infiltrated tumors have more secretory DC.

Response R2.4:

Indeed, we agree that the signature obtained by bulk RNAseq of cDC2 from inflamed tumor were enriched in mmDC that aligned with the *in vitro*-defined Secretory DC. The Single-cell RNAseq experiment was designed and performed to answer that question. We confirmed that the bulk RNAseq signature matching Secretory DC was driven by the fraction of PD-L1^{high} mmDC included in this population of sorted cDC2. The other fraction of sorted cDC2 were the PD-L1neg/low cDC2 (cluster #7 in the HNSCC ScRNAseq data in Fig 6) and their signature is available in Table 18 and Table 19. The flow cytometry data presented in Fig 1 supports that non-inflamed tumors are infiltrated by less cDC2, and that among them the proportion of PD-L1⁺ mmDC is also reduced.

R2.5. Supplementary Figure 7B-D. Which are the comparisons carried out to extract the DEGs in Fig S7D, every DC subset against the others?

Response R2.5:

Absolutely, this information was added to the figure legend. For clarity during this revision process, please note that this figure is now supplementary Fig S8D.

R2.6. Authors show that secretory DC are associated with a favorable microenvironment for antitumor immunity, as well as with response to immunotherapy based on immune checkpoint blockers. Considering that secretory DC are predominant in tumors with high CD3 infiltrate, to what extent can be these parameters (secretory DC and CD3 infiltrate) considered as independent factors or are they two inter-related factors?

Response R2.6:

This concern was also raised by Reviewers 1 & 3 and our answer below is partly redundant with R1.2b and R3.8.

The beneficial effect of a higher infiltration with Secretory DC as shown in Fig 5 can indeed be due to their ability to efficiently attract CD3 T cells, which is in line with the positive association found between PD-L1⁺ DC and CD3 High tumors shown in Fig 1 and their increased expression of T cell attracting chemokines as shown in Fig 3 (*in vitro*), Fig4 (*ex vivo*)

HNSCC RNAseq), Fig 7C and Fig S13A (*ex vivo* HNSCC ScRNAseq), Fig S15A (*ex vivo* various contexts ScRNAseq).

We confirmed the strong relationship between Secretory DC and T cell inflammation by performing correlations of the metagenes of each cell population now displayed in additional graphs in Fig 5 and new Fig S9. Pearson correlations were higher than 0.6 in all the 5 datasets (HNSCC, triple negative breast cancer (TNBC), luminal breast cancer (LumBC) and the 2 datasets studying the response to immunotherapy).

Moreover, the Secretory DC signature had a positive prognostic impact in HNSCC and TNBC, an absence of prognostic impact in LumBC, and a positive predictive impact for response to immunotherapy. We performed the same analysis with a T cell metagene and obtained similar results. Altogether, the data support the existence of a multi-cellular inflamed immune archetype enriched, among other, in T cells and Secretory cDC. This archetype ensures this partially efficient spontaneous anti-tumor immune response that provides patients with improved outcomes.

R2.7. Experiments of scRNAseq show that, as opposed to bulk RNAseq, there are different profiles of cytokine secretion between mmDC, cDC1, cDC2 and MMAC. Therefore, profiles of cytokine secretion by DC shown in Figure 3 may represent the overall secretion by all DC groups. Authors should clarify this issue when interpreting their data, since "secretory" phenotype may encompass different secretion profiles leading to different outcomes. Indeed, CCL19 and IL27 are now shown to be produced by cDC2, IL12 by mmDC and IL10, TNF and IL1B by MMAC.

Response R2.7:

As explained in R2.1 the cells analyzed in Fig 3 were cDC2 (>80%) and cDC1 (<10%), but did not include MMAC, and we have confirmed our findings with pure cDC2 (new Fig S5 A-B).

IL12, CCL19, IL27

In the ScRNA experiment, *IL12* was indeed preferentially expressed by mmDC (cluster 20), but it was also the case for the other two Secretory cytokines *CCL19* (Fig 6B and Fig S13A) and *IL27* (Fig S13B).

TNF, IL1B, IL10

At the RNAseq level (Fig 4E), Secretory DC (HNSCC-cDC2) expressed more *TNF* and *IL1B* than blood cDC2.

Regarding IL-10, it was also a feature of Secretory DC *in vitro* as shown at the protein level (Fig 3) and the RNA level (pRNA-DC, Table 10). However, there was a discrepancy with *ex vivo* data that we have highlighted in the results and the discussion, since *IL10* was not among the differentially expressed genes expressed by tumor vs blood cDC2 at the RNAseq level. At the ScRNAseq level, Secretory mmDC expressed some IL10 and were involved in an IL10/IL10R autoloop, as shown with interaction data, but the scores were weak (as compared to MMAC) and poorly increased upon maturation.

In the HNSCC ScRNAseq experiment (Fig 6), we observed that in the TME, the main producers of *TNF*, *IL1B*, and *IL10* were the MMAC, as already stated in the results section. However, mmDC may not interact with the same cell types than MMAC in the TME, and mmDC, and not MMAC, are aimed at migrating to the lymph node and interact with T cells.

Therefore, the fact that Secretory mmDC (cluster #20) express *TNF*, *IL1B* and *IL10* may still be of importance for the understanding of the anti-tumor immune response.

R2.8. cDC2 and Tregs in tumors have been correlated with the presence of differentiated CD4 T cells and with CD8 T cell responses (their ref#6). These results were obtained using the global cDC2 subset. Since according to the present manuscript mmcDC seem as those precise cells with the highest interaction capacity, would authors see a better association with Tregs and CD8 T cells when considering only mmcDC?

Response R2.8:

In the reference Binnewies et al., Cell, 2019, the authors find a positive correlation between the proportion of cDC2 among HLA-DR⁺ cells and the proportion of Treg among CD3. We confirmed this finding with a Spearman correlation coefficient of 0.47 for cDC2 among CD11c⁺HLA-DR⁺ cells.

Conversely, Treg in CD3 had a negative correlation of -0.36 with the proportion of PD-L1⁺ cDC2 among cDC2.

An opposed pattern was seen with CD8 T cells. PD-L1⁺ cDC2 were positively associated to CD8⁺ T cells among CD3 ($r = 0.24$) and PD-1⁺ cells among CD8 ($r = 0.33$), whereas cDC2 had no correlation with CD8 T cells ($r = -0.06$) and a negative correlation with PD-1⁺ cells among CD8 ($r = -0.33$).

In summary, the percentage of PD-L1⁺ cDC2 corresponding to mmDC among total CD1c⁺ cDC2 was moderately and positively associated with CD8 T cells and PD1⁺ CD8 T cells, respectively, and negatively associated with CD4 T cells and Treg.

Please find below the corresponding graphs. We did not to include this supplementary figure in the manuscript, but we provide the readers with the new Table 2 that presents the full correlation matrix of the non-redundant flow cytometry data, including cDC2 and PD-L1⁺cDC2.

R2.9. In the Discussion, authors mention about the potential effect of “other stimuli, dose, duration, etc”. Results from those perturbators that have been tested at different doses (e.g. PAM3, HKSA or HKLM) show that the chosen doses dictates important features. In some cases, increase/decrease in the dose may simply modify the "potency" of the stimulus (HKSA) without changing the profile of DC generated. In other cases (PAM3, HKLM) changes in the doses shifts the profile from a purely helper DC or purely secretory to a mixed one. Therefore, in addition to the corresponding signaling pathway, the dose should be considered when classifying these compounds, mainly for translational purposes.

Response R2.9:

Absolutely, we have highlighted this comment in the revised discussion.

R2.10. Finally, a secretory DC profile is associated with a T cell inflamed TME and with good prognosis in two of their cancer types. However, this DC subset does not promote cytokine secretion by CD4 cells in vitro, a property of CD4 T cells that would suggest a higher chance of activated T cells. Although in vivo vs in vitro differences may account for these discrepancies, there seem to be other factors involved which are not considered in this model? Would this classification apply to cDC1 cells (which, as they show, predominate in the tumor) and are responsible for activating CD8 T cells? Would it similarly affect to effector functions displayed by these CD8 T cells, with a proven role in antitumor immunity?

Response R2.10:

cDC1 do not predominate in the tumor, this confusion was related to our choice of an unclear term that has now been changed, as mentioned above (see responses R1.1a & R2.1). The low percentage of cDC1 in the blood prevented us from being able to perform such large assay as in Fig 3, so that we cannot conclude, nor exclude, that cDC1 may also present with the same Secretory/Helper dichotomy under appropriate stimuli. That is now clearly stated in the revised discussion.

Regarding the beneficial prognostic effect of Secretory DC, indeed this cDC2 activation state did not promote CD4 T cell cytokine secretion but lead to a strong secretion of T cell attracting chemokines which is key for the anti-tumor immune response and may be part of the explanation.

Regarding CD8 activation, your concern was also raised by Reviewers 1 & 3 and the answer below is redundant with response R1.3 and R3.3.

We have performed new experiments with cDC2 co-cultured with naïve CD8 T cells. cDC2 were activated by the Secretory inducer R848 or by the Helper inducer TSLP. As with CD4 T cells, Helper TSLP-cDC2 and not Secretory R848-cDC2 induced high levels of CD8 T cell cytokine secretion. These results are displayed in the new Fig S6B.

Minor comments

R2.m.1 - Reference 21 should be updated in the reference list section

R2.m.2 - Line 203: "IL-10, IL-13, IL-31 and GM-CSF as compared to both PD-L1low ICOSLhigh DC" Should this be PD-L1high ICOSL low DC?

R2.m.3 - References shown in the Y-axis in figure 5C are 25 and 26 whereas in the main text (Results and Reference sections) correspond to 27 and 28.

Response:

We have corrected these errors, thank you.

Reviewer #3 (Remarks to the Author):

In this paper, the authors seek to correlate T-cell infiltration with dendritic cell phenotype and function in human head and neck squamous cell carcinoma (HNSCC) samples and also other diseased tissues. They conclude that PD-L1 expression on DCs positively correlates with CD3+ T cell infiltration, whereas ICOS expression correlates negatively. The myeloid cells were initially categorized as monocytes and macrophages (MMAC), CD14+ DC, cDC2 and cDC1e based on flow cytometry staining. Next, the classification of DCs by PD-L1 and ICOS expression was extended into blood DCs and a database generated previously, and the analysis led the authors to conclude that PD-L1 and ICOS expression are mutually exclusive. By coculturing the 5 blood DC populations matured in vitro and analyzing the cytokines secreted, the authors conclude that PD-L1 and ICOS define “secretory” and “helper” DCs, respectively. In separate studies, based on RNA-seq, they conclude that T- inflamed HNSCCs are

enriched with secretory DC2s, and further that tumor secretory DC2s are associated with a better prognosis and response to immune checkpoint antibody therapy. Tumor secretory DC2s are characterized by mature migratory and cell-cell interaction associated genes. The secretory DC signature also exists in other chronic inflammatory and autoimmune disease sites.

Critique:

R3.1) In Figure 1, the gating of DCs appears incorrect. DCs in humans are generally gated as CD45+ Lin neg (CD3-CD19-, CD20-, CD335-) and HLA-DR+CD14-CD16-. Among these cells, pDCs are BDCA2+ CD123+, while cDC1s are CD141+ and cDC2s are CD1c+. Unfortunately, none of the DC subsets shown in figure 1B appear to be correctly gated. Problems related to the gating of DCs and T cells are present throughout the paper, which make it difficult to interpret the data.

Response R3.1:

This concern was also raised by Reviewers 1 & 2, and our answer below is partly redundant with R1.1a and R2.1.

We acknowledge that the CD11c⁺HLA-DR⁺ populations presented in Fig 1 versus Fig 2 and Fig 3 did not refer to the same cell populations which was confusing. The label “cDC1_e” for “cDC1 enriched” population was also a confusing term. We have proceeded to changes in cell population labels and names as detailed below.

In HNSCC, in Fig 1B, all cells were previously gated as CD45⁺, CD3⁻, CD19⁻, CD56⁻, as shown in Fig S2A and mentioned in the legend.

As compared to your recommendation, we have used only CD19 as a universally accepted B cell marker and not the double staining CD19⁻CD20⁻, and we have used CD56 to exclude NK cells instead of CD335.

CD11c⁺HLA-DR⁺ cells contained cDC subsets and MMAC and this name was kept.

The “cDC1_e” population that contained CD11c⁺HLA-DR⁺CD14⁻CD1c⁻ was renamed “double negative DC and MMAC” abbreviated “DN DC/MMAC”. This population contained a minority of cDC1 (median < 5%) identified as CD141^{high} as shown in Fig S2B in six independent HNSCC

samples, and other cells that most likely were non-classical CD14⁻ macrophages (possibly CD16⁺). Because of the limited number of channels available for multicolor flow cytometry, and because we had observed that cDC1 were very rare in HNSCC, we decided not to include CD141 staining in the antibody panel used in Fig 1B for our 22-patient cohort and preferred to analyze other molecules such as ICOSL and PD-L1 instead.

cDC2 were gated as HLA-DR⁺, CD14⁻, CD11c⁺, CD1c⁺, and pDC as HLA-DR⁺, CD11c⁻, CD123⁺. Therefore, as compared to your recommendations, we did not use BDCA2 and CD16 antibodies but CD11c. We used a similar strategy for cDC2 and pDC sorting for the RNAseq experiment in Fig 4, and each subset signature identified matched the corresponding subset at the ScRNAseq level.

In Blood (Fig 2; Fig 3), CD11c⁺HLA-DR⁺ cells referred to primary blood cDC1 and cDC2. We have removed this term and now use “blood cDC”. These cells were gated as CD3⁻, CD14⁻, CD16⁻, CD19⁻, HLA-DR⁺, CD4⁺, CD11c⁺, as explained in the resource paper Grandclaude et al., Cell, 2019, and the methods paper Alculumbre and Pattarini, Methods Mol Biol, 2016. These blood cDC are composed of a majority CD1c⁺CD141⁻ cDC2 (around 80%) and a minority of CD1c⁻CD141⁺ cDC1 (less than 10%) and other CD11⁺CD4⁺CD1c⁻CD141⁻ cells as shown in Figure 2f of Alculumbre and Pattarini, 2016, that is reproduced at the end of this document (Appendix 1). This information was added clearly in the methods section.

Besides name changes, we have validated the concept of Secretory and Helper DC with “pure” blood cDC2 by performing supplementary experiments. Pure cDC2 were gated as CD3⁻CD14⁻CD16⁻CD19⁻CD20⁻CD56⁻CD123⁻CD11c⁺CD1c⁺ (new Fig S5A). These results are now presented in new Fig S5B.

R3.2) Unrelated to the gating issues, in Fig. 1c it is unclear why the authors decided to divide the tumor samples obtained from the 24 patients into groups of “equivalent size” based on relative T cell frequency. In the first place, a visual examination of the dot plot suggests that rather than 3 “clusters”, there are only 2. However, there is no a priori reason to divide the samples into arbitrary groups when there are standard unbiased statistical methods that can be used to determine if there is a correlation between relative T cell frequency and other parameters. If the authors’ objective is to correlate CD3+ T cell infiltration with DC subsets, they should consider performing RNA-seq on an adequate cell sample and classify DC subsets using an unbiased algorithm.

Response R3.2:

Our initial objective was not to directly correlate DC subsets to CD3+ T cell infiltration, but rather to identify in an unsupervised manner the parameters that were part of the inflamed and non-inflamed immune archetype among 31 T cell parameters, and 50 myeloid cell parameters. This approach provided us with the discovery of the PD-L1 and ICOSL anti-correlation, that lead us to analyze its functional impact on cDC2, the main cDC subset infiltrating HNSCC.

Regarding the 3 subgroups of patients with low, medium, and high T cell infiltration, we simply divided the 22 patients in groups per tertile. To address your comment (that was also raised by reviewer 1 and 2 (see our reply R1.1b, R2.2b), we have changed our strategy in Fig 1 and performed a Spearman correlation analysis between CD3 T cell infiltration and all

other 81 non-redundant parameters, now shown in new Fig 1D. We also provide the readers with the full correlation matrix in the new Table 2. However, this approach was not compatible with the analysis of the other redundant parameters, so we kept the division per tertile for the elastic net model shown in Fig S3B.

R3.3) Different DCs preferentially stimulate CD4+ and CD8+ T cells, respectively, so it is difficult to understand the rationale for using percentages of total CD3+ T cells to correlate with DC subsets and function.

Response R3.3:

CD8 and CD4 were among the 31 T cells parameters analyzed in Fig 1. Although there were variations in the proportions of CD4 and CD8 T cells among CD3, the main driver of the total amount of CD4 and CD8 T cells in the tumor was the total amount of CD3 T cells, justifying our approach by non-redundant parameters: CD3 in Live; CD4 in CD3; CD8 in CD3.

The inflamed immune archetype characterized on the left part of the new Fig 1D shows that beyond DC subsets and PD-L1 expression, the proportion of CD8 T cells among CD3 T cells was also increased. The new Table 2 provides the full correlation matrix of the non-redundant flow cytometry parameters measured. Therefore, the reader may consult the association of each population quantified with either CD4 or CD8 T cells.

Regarding function, we focused on cDC2 and CD4 T cells.

We did not focus on cDC1 because they were so rare in the tumor (limitation of the importance and technical limitation by the number of cells for bulk RNAseq). cDC1 is also a minor subset in blood and the amount of the cells that can be collected by a single blood donor did not allow us to perform broad functional assays such as the ones performed in Fig 2 & 3.

To address your concern regarding CD8 T cells, we performed new experiments with cDC2 co-cultured with naïve CD8 T cells. cDC2 were activated by the Secretory inducer R848 or by the Helper inducer TSLP. As with CD4 T cells, Helper TSLP-cDC2 and not Secretory R848-cDC2 induced high levels of CD8 T cell cytokine secretion. These results are displayed in the new Fig S6B. This concern was also raised by Reviewers 1 & 2 and the answer above is redundant with R1.3 and R2.10.

R3.4) The authors do not explain why PD-L1 and ICOS expression are expressed in a mutually exclusive manner on the secretory and helper DCs, respectively. If this is truly the case, then shouldn't these molecules contribute to the different functions of the DCs? The details underlying Figures 1E and 1F are not provided.

The data are based on analysis of DCs in tumors, but no studies are performed to directly investigate whether or how either DC population affects anti-tumor immunity?

Response:

R3.4a

In cancer patient data, our approach to investigate the effect on anti-tumor immunity of our subset of interest, namely PD-L1⁺ mmDC that align with Secretory DC was performed by: (1)

identifying their transcriptomic signature; (2) analyzing the expression of cytokines, chemokines, checkpoint and maturation markers, and innate receptors at the bulk and single-cell transcriptomic level; (3) identifying their beneficial effect on prognosis and predictive value for response to immunotherapy as a part of the multicellular anti-tumor immune response.

Further *in vitro* functional analysis of tumor cDC would require a high number of cells that is almost impossible to obtain from primary fresh tumor samples. For this reason, functional characterization was performed with our large-scale *in vitro* database, with a specific focus on Secretory DC (e.g. R848-DC) that resemble the most the tumor infiltrating activated cDC2.

To address the role of PD-L1^{high} and ICOSL^{low} cDC2 on T cells, we have performed new blocking experiments, now presented in Fig 3C and S6A. This was also asked by Reviewer 1 (see R1.4). We wanted to determine if the absence of stimulation of CD4 cytokine secretion by Secretory DC was induced by the main immunosuppressive molecules they expressed. We performed a multiple blocking of the PD-1 / PD-L1/2; IL-10 / IL-10 receptor (IL-10R); IL-4 / IL-4 receptor (IL-4R); TIGIT / PVR axes in a naïve CD4 T cell / Secretory R848-cDC2 co-culture. IL-4 was not among the Secretory cDC molecules identified above, but its inhibitory role had been previously shown (Maier et al.). We observed that this multiple blocking increased the CD4 cytokine production of IL-3, IL-5, IL-13, and IFN γ (new Fig 3C, new Fig S6A) but not of the other cytokines measured (IL-2, IL-4, IL-6, IL-9, IL-10, GM-CSF, TNF α). For the cytokines that were increased, the levels remained low as compared to the Helper TSLP-cDC2 condition. No individual blocking was able to induce the increased CD4 cytokine production observed with the multiple blocking. Altogether, the combined expression of PD-L1, PD-L2, PVR, and secretion of IL-10 and IL-4 by Secretory cDC2 played a significant role in the inhibition of CD4 Th cytokine production.

R3.4b. About “The details underlying Figures 1E and 1F are not provided”:

The legend of Fig 1E was enriched. Figure 1F was removed since the grouping per tertile was removed from the main figure following your comment in R3.2.

R3.5) Figure S3, since all the four DC subsets express CD141, it is difficult to understand the intent of the data.

Response R3.5:

To address your comment, we have redesigned this Figure S3A (now Fig S2B). cDC1 were gated as CD141^{high}, meaning a CD141 level superior to the one observed on all other DC and MMAC subsets, among CD11c⁺HLA-DR⁺CD1c⁻CD14⁻ cells, and not on the isotype. Indeed, it is well known that CD141 is specific of cDC1 only in the blood of healthy donors but can be induced at low levels on other myeloid subsets in inflammatory conditions.

This strategy allowed us to eliminate the non-classical MMAC also present in the parental gate, as explained in R3.1.

R3.6) The rationale for performing the extensive experiments described in Figure 3 on blood DCs using diverse stimuli, *in vitro*, is unclear. By definition, all of these conditions are distinct

from those in HNSCC. Importantly, also, the studies described in reference 21 used moDCs, which are distinct from blood CD11c+ DCs.

Response R3.6:

For the analyses presented in Fig 2 and Fig 3, all data presented were obtained with primary blood cDC only, as explained in R3.1 and we have excluded data obtained with monocyte-derived DC. We have clarified this point in the results section.

Regarding the stimuli, they may indeed be different from those in HNSCC, which are mostly uncharacterized, since a tumor is a very complex inflammatory environment. For *in vitro* studies, we had to use well identified stimuli representative of a diversity of inflammatory settings. The objectives were (i) to confirm the anti-correlation of PD-L1 and ICOSL observed in 22 HNSCC in a larger *in vitro* dataset with many different stimuli; (ii) to identify the functional impact associated with the PD-L1 / ICOSL profiles of activated DC using this *in vitro* model. Then, to validate the relevance of the *in vitro* model in HNSCC, we used bulk RNAseq (Fig 4) and ScRNAseq in HNSCC (Fig 6) and confirmed that tumor infiltrating cDC2 aligned with the *in vitro*-defined Secretory phenotype. Finally, the ScRNAseq analysis performed in Fig 8 confirmed that the mature migratory tissue cDC present in multiple pathologies share functional similarities with the *in vitro* Secretory cDC, showing that in all these different contexts, tissue stimuli converged with *in vitro* Secretory inducers (e.g R848, Zymosan) and not Helper inducers (e.g Flu, GM-CSF, TSLP).

R3.7) The raw data for Figure S5 C and D are not provided, so it is not possible to match the data with the figures.

Response R3.7:

Sorry we do not understand this comment. The Figures S5C (now S4C) representing T cell expansion and S5D (now S4D) representing cDC survival, are dot plots presenting the full raw data, like the other plots in Fig 3 and the Fig S5A and S5B (now S4A-B).

R3.8) The rationale for the studies of DC2s described in Figures 4 and 5 is not well explained, and therefore difficult to connect to the initial and main focus of the report.

Response R3.8:

In Fig 1, we had discovered the anti-correlation of PD-L1 and ICOSL and shown that PD-L1^{high} and ICOSL^{low} cDC2 were enriched in CD3 inflamed HNSCC.

In Fig 2 and Fig 3 we observed that the PD-L1 and ICOSL anti-correlation was reproducible by various stimuli engaging different receptors and had a functional impact that defined *in vitro*-Secretory DC.

In Fig 4, we confirmed that the Secretory phenotype applied to tumor infiltrating cDC2 using the similarities in the expression at the RNA level of the molecules relevant to Secretory DC (cytokines, chemokines, maturation markers, checkpoints).

In Fig 5, we used the transcriptomic signature of Tumor Secretory cDC2 to show that the abundance of Secretory cDC is associated to good prognosis and to response to immunotherapy. We have now added new graphs that confirm at the transcriptomic level the observation made by flow cytometry in Fig 1 of the positive correlation between T cell

infiltration and Secretory cDC2 (Fig 5 and new Fig S9A-D). We therefore propose that cDC2 in inflamed HNSCC mature and acquire a Secretory phenotype, promote T cell infiltration and are 2 major components of the inflamed immune archetype. This inflamed environment translates the presence of a spontaneous immune response, that although insufficient for tumor clearance, still confers an advantage on outcomes to patients with or without immunotherapy, as compared to patients that have non-inflamed tumors. This concern on prognosis was also raised by Reviewers 1 & 2 and our answer above on Fig 5 is partly redundant with R1.2b and R2.6.

R3.9) In the discussion, the authors describe the “Secretory DC signature as the combination of maturation markers (CD40, CD70, CD80, LAMP3), migratory marker CCR7, chemokines (CCL19, CXCL9), cytokines (IL12B, IL23A, EBI3), the receptor LY75 (DEC-205), checkpoints CD274 (PD-L1), PDCD1LG2 (PD-L2), PVR, IDO, CD200, TNFRSF9 (4-1BB), and the transcription factors ETV3/ETV3L and REL/RELB.” Since DEC205 is a DC1 marker, while IDO, CD200, PD-L1 are generally expressed by regulatory DCs, the authors’ definition is at odds with current understanding of DC biology.

Response 3.9:

The concept of regulatory DC that would preferential induce TReg polarization as opposed to other Th profiles is not universally accepted. The functional dichotomy presented in this study was obtained in an unbiased and data-driven manner. The large amount of data and diversity of the stimuli used here allowed us to identify one DC maturation profile preferentially secreting cytokines and chemokines (Secretory DC) and a second one preferentially inducing Th cytokine production without a clear polarization towards a TReg or specific Th profile (Helper DC).

The Bulk RNAseq data presented in Fig 4 was obtained by sorting cDC2 and confirmed that DEC-205 (*LY75*) was upregulated upon tumor-induced maturation in this subset as this gene is part of the cDC2 signature (Table 15).

In the ScRNAseq data presented in Fig 6, consistently with the literature, we observe 3 cDC subsets: cDC1, cDC2 and mature migratory cDC (“mmDC”) that originated from cDC1 or cDC2, and that others have called “LAMP3+DC”, “DC3”, or “mregDC”. Here, we observed that DEC-205 (*LY75*) was selectively expressed by mmDC, and not by non-migratory cDC1 and cDC2, confirming that DEC-205 (*LY75*) is upregulated upon tumor-induced maturation in both subsets.

Beyond DEC-205, the set of genes expressed by Secretory DC in vitro and mmDC ex vivo are consistent with the recent literature on tumor-infiltrating DC, and indeed characterized by the co-expression of immune-stimulatory and immune-suppressive molecules.

To address your comment, we have clarified this point and the relevant literature in the Discussion section.

Appendix 1

For the understanding of cell subsets analyzed in the *in vitro* data presented in Fig 2 and Fig 3 in the present manuscript, please find the gating strategy used in the Methods article from Alculumbre and Pattarini, 2016, PMID: 27142015.

[REDACTED]

REVIEWER COMMENTS

Reviewer #1 (Remarks to the Author):

The authors have addressed most of my concerns in this revision and the revised manuscript is much clear.

There is however some ambiguities from line 131 to 134: how could MMAC enrich in both CD3^{hi} tumor and CD3^{lo} tumor? Please clarify it.

Reviewer #2 (Remarks to the Author):

All my comments and suggestions have been properly addressed. The manuscript has gained in clarity in the new version.

Minor point:

Discussion, line 582, please correct:

"A recent study in lung cancer proposed that IL-12 secretion IS WAS in part..."

Reviewer #3 (Remarks to the Author):

Summary of key concerns: In this revised paper, the authors have addressed a number of the issues raised in the initial review, and the data are interesting. However, there remain several problems that limit the significance of the findings. The dichotomy of 'secretory' and 'helper' DCs is based on studies mainly of cDC2 cells (not total DCs or DC1s), and the analysis is based on in vitro cytokine secretion in response to certain exogenous stimuli, along with cell surface expression of PD-L1 and ICOSL. Since DC2s were the main cells studied, the term 'DCs' in the title, abstract, key findings, and discussion is overly broad and should be corrected. Beyond this issue, the use of PD-L1 and ICOSL to define the two DC2 populations appears to have varied among experiments. For example, ICOSL expression but not PD-L1 expression is used in Fig1E and FigS3B without explanation. "Secretory DCs" are defined on the basis of cytokine secretion in response to potent exogenous stimuli, in vitro, but it's not clear whether a subpopulation of DCs secretes these cytokines in vivo. The fact that they are preferentially present in inflamed tumors, atopic dermatitis and lupus nephritis is interesting, but if and how this population contributes to disease development or progression is not clear.

Additional specific comments are as follows:

In their description of Fig. 1, the authors state: 'Using correlation analysis, we observed that tumors high in CD3 were enriched in cDC2, DN DC/MMAC, pDC, PD-L1+ DC and MMAC (all 132 subsets), and in CD8 T cells (Fig 1D; full correlation matrix in Table 2).' This is confusing. Why is PD-L1 expression described within DC, but not in cDC2, DN, pDC, or MMAC?

In their description of Fig2B, the authors specified the population as 'PD-L1high and ICOSLlow/neg, like ex vivo cDC2 from inflamed tumors.' Why in figure1 the term DC used, but in Fig2 the cells are described as cDC2?

In FigS3B, the ICOSL+ percentage was analyzed within CD1c-HLA-DRhigh, CD11+HLA-DR+, MMAC+, MMAC HLA-DRhigh cells, but where is the analysis of PD-L1+ cells in DCs from high CD3+ T infiltration samples? Moreover, it seems unlikely that the extremely potent activating stimuli used in in vitro reflect the situations in cancers, autoimmune and allergic diseases.

Line 173, the authors use the term 'cDC', which is not accurate: 'The cDC perturbators inducing a majority of PD-L1high ICOSLlow cDC were R848, Zymosan, heat-killed Staphylococcus aureus (HKSA) and heat-killed Listeria monocytogenes (HKLM).' Moreover, since none of these stimuli are present in significant quantities in tumors, what is their relevance of the findings to human tumors in vivo? Could the data explain the correlation of PD-L1high DCs with CD3+ T cell high infiltration?

The authors used the profiles obtained from their in vitro experiments to characterize blood cDCs. However, they used sorted cDC2s to validate the dichotomy by co-culturing these cells with naïve CD4+ T cells in the presence of either R848 or TSLP. They also analyzed the impact of these two populations on CD8+ T cells. Since cDC1s are the main population responsible for cross-presentation, which is critical in anti-tumor immunity, and their functions cannot be recapitulated with cDC2 cytokines, it is difficult to understand how the in vitro stimulation studies relate to the role of CD1s in tumors.

REVIEWER COMMENTS

Reviewer #1 (Remarks to the Author):

The authors have addressed most of my concerns in this revision and the revised manuscript is much clear.

There is however some ambiguities from line 131 to 134: how could MMAC enrich in both CD3hi tumor and CD3lo tumor? Please clarify it.

Response:

CD3hi tumors were enriched in PD-L1+ MMAC (expressed in % of total MMAC), whereas CD3lo tumors were enriched in MMAC (expressed in % of CD11c+HLADR- cells, their parental population).

That means that CD3hi tumors contained less MMAC in total than CD3lo tumors, but these MMAC in CD3hi tumors expressed more PD-L1.

Reviewer 3 also raised the concern about the ambiguity of this sentence.

We have modified the text in lines 131 to 134 to make this difference PDL1+MMAC versus MMAC (i.e. all MMAC) more clear, thank you.

Reviewer #2 (Remarks to the Author):

All may comments and suggestions have been properly addressed. The manuscript has gained in clarity in the new version.

Minor point:

Discussion, line 582, please correct:

“A recent study in lung cancer proposed that IL-12 secretion IS WAS in part...”

Response: We have corrected this error, thank you.

Reviewer #3 (Remarks to the Author):

Summary of key concerns: In this revised paper, the authors have addressed a number of the issues raised in the initial review, and the data are interesting. However, there remain several problems that limit the significance of the findings. The dichotomy of ‘secretory’ and ‘helper’ DCs is based on studies mainly of cDC2 cells (not total DCs or DC1s), and the analysis is based on in vitro cytokine secretion in response to certain exogenous stimuli, along with cell surface expression of PD-L1 and ICOSL.

3.1/ Since DC2s were the main cells studied, the term ‘DCs’ in the title, abstract, key findings, and discussion is overly broad and should be corrected.

Response:

We acknowledge that the present maturation dichotomy was obtained in vitro in a database studying blood cDC that contained a majority of cDC2 and a minority of cDC1, and that

further validation was made on pure cDC2 in vitro. Therefore, we have added precision on this point in the abstract and key findings section.

However, our single-cell experiments (Fig6; Fig8) showed that Secretory cDC in vivo in cancer and other pathological contexts aligned with mature migratory cDC. In the literature, these mature migratory cDC have been named alternatively “LAMP3+cDC” (Cheng et al., Cell 2021, PMID:333545035); “IDO+DC” (Cillo et al., Immunity 2020, PMID: 31924475); “CCR7+cDC” (Brown et al., Cell 2019, PMID: 31668803); “mregDC” (Maier et al., Nature 2020, PMID: 32499658); or even “mDC3” (Zilionis et al., Immunity 2019, PMID: 30979687). Therefore, we consider that changing “dendritic cells” in the title into “type 2 dendritic cells” would be confusing with the final findings of the paper. In addition, the title describes the new concept that we have established, which may apply to a larger set of DC types. We hope that follow up studies in the field will address this topic, as was the case for previous DC biology concepts, such as the DC1/DC2 paradigm (Rissoan et al, Science, 1999), or the concept of regulatory DC.

3.2/ Beyond this issue, the use of PD-L1 and ICOSL to define the two DC2 populations appears to have varied among experiments. For example, ICOSL expression but not PD-L1 expression is used in Fig1E and FigS3B without explanation.

Response:

Fig 1E and Fig S3B correspond to data collected within the same experiments, the immune monitoring by flow cytometry of 22 fresh human head and neck cancer samples.

Fig 1E are representative FACS plots and PD-L1 is represented in the 3 graphs left.

Fig S3B is an elastic net model performing statistics on the three supervised groups of CD3low, CD3Int and CD3high. This model contains all the data collected by flow cytometry (434 parameters), including redundant variables and cell/cell ratios and 14 clinical parameters. Therefore, PD-L1 related data were present in the analysis in Fig S3B. The graph only represents the statistically significant variables. It appears that ICOSL but not PD-L1 was significant with this specific approach, that explains why PD-L1 is not visible. The objective of this model was not to see again the anti-correlation of ICOSL and PD-L1, but to determine whether another flow cytometry parameter, cell-to-cell ratio, or clinical parameter was also significantly associated to CD3 levels, which eventually was not the case.

3.3/ “Secretory DCs” are defined on the basis of cytokine secretion in response to potent exogenous stimuli, in vitro, but it’s not clear whether a subpopulation of DCs secretes these cytokines in vivo.

Response:

Indeed “Secretory DCs” were defined by cytokine secretion in vitro. In vivo, we have analyzed and confirmed the enriched cytokine gene expression by RNAseq in cDC2 from HNSCC (Fig 4E) and by ScRNAseq in CCR7+ LAMP3+ mature migratory cDC from HNSCC (Fig S13) and other pathologies (Fig S15). It was not feasible to address cytokine secretion by ex vivo sorted cDC2 from tumors at the protein level due to obvious limitations in the number of cells required for such experiments.

3.4/ The fact that they are preferentially present in inflamed tumors, atopic dermatitis and lupus nephritis is interesting, but if and how this population contributes to disease development or progression is not clear.

Response:

The purpose of the paper is to define for the first time the concept of Secretary/Helper dichotomy in DC maturation, and to show that mature migratory DC in various anatomical locations align with each other, and with the Secretary phenotype defined in vitro. The gene signature of these cells and the cell-cell communication network analysis raise hypothesis on their role in disease development or progression, but the demonstration of their contribution is beyond the scope of the present work.

Additional specific comments are as follows:

3.5/ In their description of Fig. 1, the authors state: 'Using correlation analysis, we observed that tumors high in CD3 were enriched in cDC2, DN DC/MMAC, pDC, PD-L1+ DC and MMAC (all subsets), and in CD8 T cells (Fig 1D; full correlation matrix in Table 2).' This is confusing. Why is PD-L1 expression described within DC, but not in cDC2, DN, pDC, or MMAC?

Response:

Yes, the PD-L1+ enrichment is indeed in cDC2, DN, pDC, and MMAC in the CD3hi tumors, that is what we meant with "PD-L1+ DC and MMAC (all subsets)". A similar concern on the ambiguity of this sentence was raised by reviewer 1.

For clarity, we have changed this sentence to completely detail the subsets.

3.6/ In their description of Fig2B, the authors specified the population as 'PD-L1high and ICOSLlow/neg, like ex vivo cDC2 from inflamed tumors.' Why in figure1 the term DC used, but in Fig2 the cells are described as cDC2?

Response:

As explained in response 3.5, in Figure 1 the term DC was used only in a generic manner as "DC and MMAC (all subsets)" to avoid to repeat several times the names of the 4 subsets defined above in the first part of the paragraph and in Fig 1B (upper right panel): cDC2, CD14+, DN DC/MMAC and MMAC.

For clarity, we have removed "DC and MMAC (all subsets)" and replaced by the detailed names of each subset in this paragraph of the results section.

Therefore, both Figure 1 and 2 contain data on cDC2.

3.7/ In FigS3B, the ICOSL+ percentage was analyzed within CD1c-HLA-DRhigh, CD11+HLA-DR+, MMAC+, MMAC HLA-DRhigh cells, but where is the analysis of PD-L1+ cells in DCs from high CD3+ T infiltration samples?

Response:

Please refer to the response provided in point 3.2.

3.8/ Moreover, it seems unlikely that the extremely potent activating stimuli used in in vitro reflect the situations in cancers, autoimmune and allergic diseases.

Line 173, the authors use the term 'cDC', which is not accurate: 'The cDC perturbators inducing a majority of PD-L1^{high} ICOSL^{low} cDC were R848, Zymosan, heat-killed Staphylococcus aureus (HKSA) and heat-killed Listeria monocytogenes (HKLM).

Response:

In the first part of the paragraph (lines 153-154), we defined the term blood cDC: "We used the existing data on primary blood cDC composed of a majority of cDC2 and a minority of cDC1 (we excluded monocyte-derived DC and pDC), [...]."

The term "blood cDC" or more concisely "cDC" is used throughout this paragraph to refer to the cells used in the large database containing all the stimuli.

In lines 211-213 we present the validation experiments on sorted pure cDC2 (Fig S5A, S5B) and only there use this term "cDC2".

3.9/ Moreover, since none of these stimuli are present in significant quantities in tumors, what is their relevance of the findings to human tumors in vivo?

Response:

This project was developed in three main steps. First, flow cytometry data lead to the observation of PD-L1 and ICOSL anti-correlation on several myeloid cell subsets including cDC2. Secondly, we used a large database of blood cDC (large majority of cDC2, minority of cDC1) stimulated by a broad spectrum of molecules. These molecules were used at doses defined as optimal DC activation concentration, which allow comparability and meets the standard of field. These many different stimulation types allowed us to validate in vitro the observation of PD-L1 and ICOSL anti-correlation and to decipher its functional impact. Thirdly, we went back to human tissue samples and confirmed that in vivo mature migratory cDC, alternatively labelled in the literature as "LAMP3+cDC"; "IDO+DC"; "CCR7+cDC"; "cDC3"; "mregDC", aligned with the in vitro defined PD-L1^{high} ICOSL^{low} cDC2 Secretary DC.

Therefore, we identified in vitro the Secretary DC maturation as a common maturation type that was not specific of a single stimulus, nor a common receptor pathway (R848, Zymosan, heat-killed Staphylococcus aureus (HKSA) and heat-killed Listeria monocytogenes (HKLM); Table 4 for the corresponding receptors).

Similarly, in vivo, we observed that mature migratory cDC aligned together in the different tissues, anatomical locations, and pathologies studied in Figure 8 (three different cancers; atopic dermatitis; lupus). This further confirms that a common maturation phenotype can be obtained in different environments that differ by the type, doses, and number of stimuli.

3.10/ Could the data explain the correlation of PD-L1^{high} DCs with CD3⁺ T cell high infiltration?

Response:

Yes, PD-L1^{high}DCs are mature migratory DC in tumors, have a Secretory phenotype, and secrete many T cell attracting chemokines that at least in part explain the high CD3+ T cell infiltration. It is also possible that the damage-associated molecular patterns and other cDC stimuli present in inflamed tumors, and that lead to cDC maturation and Secretory phenotype induction, are associated with molecules that may have a direct effect in T cell attraction.

3.11/ The authors used the profiles obtained from their in vitro experiments to characterize blood cDCs. However, they used sorted cDC2s to validate the dichotomy by co-culturing these cells with naïve CD4+ T cells in the presence of either R848 or TSLP. They also analyzed the impact of these two populations on CD8+ T cells. Since cDC1s are the main population responsible for cross-presentation, which is critical in anti-tumor immunity, and their functions cannot be recapitulated with cDC2 cytokines, it is difficult to understand how the in vitro stimulation studies relate to the role of CD1s in tumors.

Response:

cDC1 are the main population for CD8 T cell cross-presentation in mice, but (i) cDC2 are similarly capable of cross-presentation in human (Segura et al., J Exp Med, 2013, PMID: 23569327) and (ii) cDC2 largely outnumber cDC1 in tumors as shown in Fig S2B and Fig 6A, supporting that cDC2-CD8 T cell interaction in tumors is of physiological importance. Besides, our T cell functional readout was cytokine secretion and not cross-presentation.

REVIEWER COMMENTS

Reviewer #1 (Remarks to the Author):

The authors have addressed my concerns. I have no further question

Reviewer #2 (Remarks to the Author):

Authors have addressed all my comments.

Reviewer #3 (Remarks to the Author):

At the most fundamental level, DCs are defined on their ability to induce naïve T cells to respond to specific antigens, and the distinction between cDC2s and cDC1s is based on their distinct ontogeny and functions. The authors' concept of distinct secretory and helper DCs is novel and intriguing. However, since they performed no direct studies of the ontogeny or functions of these cells (including any measure of how secretory DCs influence tumor infiltrating T cells), and the phenotypic profiling and cellular sources are inconsistent throughout, it is impossible to know the true significance of the findings or place "secretory" and "helper" DC into our current understanding of DC biology.

Examples of experimental inconsistency are as follows. In Figure 6, the sc-seq was performed on 2 HNSCC tumor samples and 1 juxtatumor sample. Three samples are not sufficient to draw any conclusions. Moreover, how can the authors suggest that cDC2s mature into mMDCs enriched with the secretory signature without correlating the frequency mMDCs with CD3 T cell infiltration? Tumor MMAC is a confusing term. Monocyte-derived DCs/macrophages are distinct from cDC2s, and this and other inconsistencies are reflected in several figures, as stated in the earlier reviews. For example, in Figure 1, the PD-L1 vs. ICOSL definition was made from different DC subsets, including MMAC and cDC2s. In Figure 2, the secretory signature was derived from a mixture of blood cDC2 and cDC1 cultures, although cDC2 was the major population. In Figure 6, the mature migratory DCs were derived from cDC2s, but the authors do not mention MMAC which also express CCR7 and CD274. These types of inconsistencies are both confusing and frustrating.

REVIEWER COMMENTS

Reviewer #1 (Remarks to the Author):

The authors have addressed my concerns. I have no further question

Reviewer #2 (Remarks to the Author):

Authors have addressed all my comments.

Reviewer #3 (Remarks to the Author):

3.1/ At the most fundamental level, DCs are defined on their ability to induce naïve T cells to respond to specific antigens, and the distinction between cDC2s and cDC1s is based on their distinct ontogeny and functions. The authors' concept of distinct secretory and helper DCs is novel and intriguing. However, since they performed no direct studies of the ontogeny or functions of these cells (including any measure of how secretory DCs influence tumor infiltrating T cells), and the phenotypic profiling and cellular sources are inconsistent throughout, it is impossible to know the true significance of the findings or place "secretory" and "helper" DC into our current understanding of DC biology.

Examples of experimental inconsistency are as follows. In Figure 6, the sc-seq was performed on 2 HNSCC tumor samples and 1 juxtatumor sample. Three samples are not sufficient to draw any conclusions.

Response:

We acknowledge that 3 samples are a limited number to draw definite conclusion, especially in the context of inter-patient variability. This is why we have performed the large analysis presented in Figure 8 that includes a total of 111 samples, among which 36 samples of intra-tumoral dendritic cells from HNSCC, breast and lung cancer.

We also performed a similar analysis restricted to tumor-infiltrating CD45+ cells from the 26 samples of the HNSCC dataset Cillo et al. (GSE139324), and confirmed that the mature migratory cDC from cluster 16 (signature available in Table 24) aligned with Tumor Secretory cDC2, similarly to what we showed in Figure 6C in our scRNAseq data in HNSCC. This result is shown in the graph below. Since these 26 samples were part of the analysis with 111 samples in Figure 8, we decided not to include this graph in the present revised version of the manuscript.

3.2/ Moreover, how can the authors suggest that cDC2s mature into mmDCs enriched with the secretory signature without correlating the frequency mmDCs with CD3 T cell infiltration?

Response:

In Figure 5, we showed the correlation between the Tumor Secretory cDC2 signature and the CD3 T cell signature. To address this additional concern, we performed new correlation analyses between the Mature Migratory DC signature obtained from scRNA data and the CD3 T cell signature. We used the same 5 datasets as in Figure 5 and obtained similar results. The new graphs are now presented in Figure 8E and S18 A-D. The Mature Migratory DC signature is provided in new Table 31 and is composed of the 29 genes in common between the signatures of cluster 20 in Figure 6 and cluster 4 in Figure 8, using the thresholds of log2 fold change > 1 and adjusted p-value < 0.05. This information has been added to the Methods section.

3.3/ Tumor MMAC is a confusing term. Monocyte-derived DCs/macrophages are distinct from cDC2s, and this and other inconsistencies are reflected in several figures, as stated in the earlier reviews. For example, in Figure 1, the PD-L1 vs. ICOSL definition was made from different DC subsets, including MMAC and cDC2s. In Figure 2, the secretory signature was derived from a mixture of blood cDC2 and cDC1 cultures, although cDC2 was the major population. In Figure 6, the mature migratory DCs were derived from cDC2s, but the authors do not mention MMAC which also express CCR7 and CD274. These types of inconsistencies are both confusing and frustrating.

Response:

As pointed by the reviewer, our study analyses different antigen-presenting cell subsets, in different figures and parts of the manuscript. What is called “inconsistencies” in fact serves very different purposes in the logical flow of the study. Below our replies to each specific point.

“Tumor MMAC is a confusing term”:

To avoid any confusion, MMAC were precisely defined in the manuscript as “CD14+ monocytes and macrophages” (see first paragraph of the results section). This is compatible with standard definitions. We have used this abbreviation in other publications without raising any issues (For example: Michea et al, Nat Immunol, 2018).

“Monocyte-derived DCs/macrophages are distinct from cDC2s”:

We perfectly agree with the reviewer. As a matter of fact, we have not used the term monocyte-derived DCs anywhere in the manuscript. We have used the term mono/macrophages (MMAC), which indeed are different from cDC2.

“[...] this and other inconsistencies are reflected in several figures, as stated in the earlier reviews. For example, in Figure 1, the PD-L1 vs. ICOSL definition was made from different DC subsets, including MMAC and cDC2s. In Figure 2, the secretory signature was derived from a mixture of blood cDC2 and cDC1 cultures, although cDC2 was the major population. In Figure 6, the mature migratory DCs were derived from cDC2s, but the authors do not mention MMAC which also express CCR7 and CD274. These types of inconsistencies are both confusing and frustrating.”:

As commented earlier, we have indeed analyzed different types of antigen-presented cells, for different purposes and to address different questions.

In Figure 1, our purpose was exploratory, and hypothesis-generating. That is why we studied bulk tumor cell suspensions, which included all antigen-presenting cell types. We could observe the differential expression of PD-L1 and ICOSL, and raised the hypothesis that this may correspond to various APC states.

To explore this hypothesis more in depth, we turned to DC as an APC type of interest, hence our Figure 2 and Figure 3, which include a mixture of cDC1 and cDC2. This established the concept of secretory versus helper DC. We have validated this concept in pure cDC2 (Fig 3C, S5, S6). MMAC were not studied here, that is why our conclusion and concept is on DC and not MMAC.

In the third step (Figure 4 to Figure 8), we wanted to validate the existence of secretory DC *ex vivo* in different physiopathological situations.

In the Figure 4, because of the rarity of cDC1, we decided to focus on cDC2 in blood and tumor and defined the Tumor Secretory cDC2 signature. Indeed, we did not mention MMAC because it was not our purpose.

In the Figure 6, we identify the cDC subset that expresses this Tumor Secretory cDC2 signature. Again, as expected, none of the three MMAC clusters expressed this signature (Fig 6C).

In the Figure 8, we validate our findings from Figure 6 in a more extensive set of samples, tissues, and clinical situations, still focusing on cDC subsets.

In conclusion, the apparent inconsistencies correspond to different steps of a logical flow to demonstrate our major conclusions.

REVIEWERS' COMMENTS

Reviewer #4

"I have two comments from my side to be considered:

The authors write in line 432 from cDC3 (citation 42). However, they are only called DC3 and not cDC3. This should be corrected.

Furthermore, the authors cite the manuscript of Brown et al., 2019 (citation 31) that has dealt with the distinction of the murine cDC2 subset into cDC2A and cDC2B cells. However, in the human system this classification cannot be made. In addition, the authors cite this manuscript due to the presence of the marker CLEC10A (cDC2 expressed the marker CLEC10A,...described in blood and spleen...lines 357-359). As said, the Brown paper solely focused on murine DCs. I would suggest to acknowledge the work of Heidkamp et al., who were the first to identify the marker CLEC10A as an identifier of human cDC2 cells in human blood, spleen, and thymus (Sci Immunol. 2016 Dec 16; 1(6):eaai7677. doi: 10.1126/sciimmunol.aai7677. Epub 2016 Dec 16. PMID: 28783692), followed by a study of Heger et al., 2018 (Front Immunol. 2018 Apr 27; 9:744. doi: 10.3389/fimmu.2018.00744. eCollection 2018. PMID: 29755453)."

Reviewer #5 (Remarks to the Author):

Please see comments to Editor. As our role was as mediator, we have addressed it to them, not to the authors. Please let us know if you need us to rephrase to address authors.

One small suggestion would be to show Figure 1 panel E as a dot plot PD-L1 vs ICOSL, as they claim that it is rather one or the other and then play with the different level of expression combination (Fig 2.B) for the tumor CD3 low as they just show the CD3 hi in the Sup2D. Not critical, however.

REVIEWER COMMENTS

Reviewer #4 (Remarks to the Author):

"I have two comments from my side to be considered:

4.1/ The authors write in line 432 from cDC3 (citation 42). However, they are only called DC3 and not cDC3. This should be corrected.

Response: This has been corrected, thank you

4.2/ Furthermore, the authors cite the manuscript of Brown et al., 2019 (citation 31) that has dealt with the distinction of the murine cDC2 subset into cDC2A and cDC2B cells. However, in the human system this classification cannot be made. In addition, the authors cite this manuscript due to the presence of the marker CLEC10A (cDC2 expressed the marker CLEC10A,...described in blood and spleen...lines 357-359). As said, the Brown paper solely focused on murine DCs. I would suggest to acknowledge the work of Heidkamp et al., who were the first to identify the marker CLEC10A as an identifier of human cDC2 cells in human blood, spleen, and thymus (Sci Immunol. 2016 Dec 16;1(6):eaai7677. doi: 10.1126/sciimmunol.aai7677. Epub 2016 Dec 16. PMID: 28783692), followed by a study of Heger et al., 2018 (Front Immunol. 2018 Apr 27;9:744. doi: 10.3389/fimmu.2018.00744. eCollection 2018. PMID: 29755453)."

Response: The manuscript of Brown et al., 2019 is mostly based on murine experiments but contains some human data. We have clarified this in the discussion.

Thank you for the other references. We have modified the discussion as follows:

"At the protein level, CLEC10A is known to be expressed in over 80% of cDC2 in blood, spleen and thymus (Heidkamp et al.), and downregulated upon TLR7/8 maturation (Heger et al.), consistent with the absence of expression on mmDC #20."

Reviewer #5 (Remarks to the Author):

Please see comments to Editor. As our role was as mediator, we have addressed it to them, not to the authors. Please let us know if you need us to rephrase to address authors.

One small suggestion would be to show Figure 1 panel E as a dot plot PD-L1 vs ICOSL, as they claim that it is rather one or the other and then play with the different level of expression combination (Fig 2.B) for the tumor CD3 low as they just show the CD3 hi in the Sup2D. Not critical, however.

Response: We have added the PD-L1 vs ICOSL dot plots graph to Fig 1E, thank you.